# Integrative proteomic characterization of adenocarcinoma of esophagogastric junction

Shengli Li [1,2,8], Li Yuan [1,3,4,8], Zhi-Yuan Xu[1,3,4], Jing-Li Xu[5], Gui-Ping Chen[6], Xiaoqing Guan[1], Guang-Zhao Pan[1], Can Hu[5], Jinyun Dong[1], Yi-An Du[1,3,4], Li-Tao Yang[1,3,4], Mao-Wei Ni[1], Rui-Bin Jiang[1], Xiu Zhu[1], Hang Lv [7], Han-Dong Xu[5], Sheng-Jie Zhang[1], Jiang-Jiang Qin [1,3,4] ✉ & Xiang-Dong Cheng [1,3,4] ✉

The incidence of adenocarcinoma of the esophagogastric junction (AEG) has been rapidly increasing in recent decades, but its molecular alterations and subtypes are still obscure. Here, we conduct proteomics and phosphoproteomics profiling of 103 AEG tumors with paired normal adjacent tissues (NATs), whole exome sequencing of 94 tumor-NAT pairs, and RNA sequencing in 83 tumor-NAT pairs. Our analysis reveals an extensively altered proteome and 252 potential druggable proteins in AEG tumors. We identify three proteomic subtypes with significant clinical and molecular differences. The S-II subtype signature protein, FBXO44, is demonstrated to promote tumor progression and metastasis in vitro and in vivo. Our comparative analyses reveal distinct genomic features in AEG subtypes. We find a specific decrease of fibroblasts in the S-III subtype. Further phosphoproteomic comparisons reveal different kinase-phosphosubstrate regulatory networks among AEG subtypes. Our proteogenomics dataset provides valuable resources for understanding molecular mechanisms and developing precision treatment strategies of AEG.

Adenocarcinoma of the esophagogastric junction (AEG) generally refers to the adenocarcinoma that occurs in the esophagogastric junction within the range of 5 cm in both directions[1,2]. More than 1.5 million patients suffer from AEG each year[3,4]. AEG tumors are anatomically classified into three types[5]: Siewert type I, tumors with an epicenter of 1–5 cm above the esophagogastric junction (EGJ); Siewert type II, tumors within 1 cm above and 2 cm below the EGJ; and Siewert type III, tumors within 2–5 cm below the EGJ. AEG is obviously different

from gastric cancer in epidemiology, etiology, and pathological characteristics. The incidence rate of AEG has increased year by year, while that of gastric antral carcinoma has decreased significantly[6,7]. According to the Lauren classification, the intestinal type was most common in AEG, and intestinal metaplasia led by gastroesophageal reflux disease (GERD) is the main risk factor for AEG[8,9]. However, there are more diffuse type cases of gastric antrum carcinoma, and chronic atrophic gastritis is an important precancerous lesion of gastric

[1]Department of Gastric Surgery, The Cancer Hospital of the University of Chinese Academy of Sciences (Zhejiang Cancer Hospital), Institutes of Basic Medicine and Cancer (IBMC), Chinese Academy of Sciences, Hangzhou 310022, China. [2]Precision Research Center for Refractory Diseases, Institute for Clinical Research, Shanghai General Hospital, Shanghai Jiao Tong University School of Medicine, Shanghai 201620, China. [3]Zhejiang Provincial Research Center for Upper Gastrointestinal Tract Cancer, Zhejiang Cancer Hospital, Hangzhou 310022, China. [4]Zhejiang Key Lab of Prevention, Diagnosis and Therapy of Upper Gastrointestinal Cancer, Zhejiang Cancer Hospital, Hangzhou 310022, China. [5]First Clinical Medical College, Zhejiang Chinese Medical University, Hangzhou 310053, China. [6]Department of Gastrointestinal Surgery, the First Affiliated Hospital of Zhejiang Chinese Medical University, Hangzhou 310006, China. [7]Biological Sample Bank, the First Affiliated Hospital of Zhejiang Chinese Medical University, Hangzhou 310006, China. [8]These authors contributed equally: Shengli Li, Li Yuan. ✉e-mail: jqin@ucas.ac.cn; chengxd@zjcc.org.cn

antrum carcinoma[9]. In addition, *Helicobacter pylori* (*H. pylori*) infection is a recognized carcinogenic factor of gastric antrum cancer. Cytotoxigenic associated gene A (CagA) in *H. pylori* may significantly increase the risk of atrophic gastritis and gastric antrum cancer, but its role in AEG is controversial[10]. Some studies have shown that *H. pylori* infection can prevent GERD, Barrett's esophagus and other reflux diseases, thus reducing the incidence of AEG to a certain extent[11]. Currently, comprehensive treatment, including surgical resection, chemotherapy, and immunotherapy, is the most effective treatment for AEG. However, most AEG patients have locally advanced tumors or distant metastasis at diagnosis and are ineligible for surgery[12]. Targeted therapies are only for patients with late-stage metastatic HER2-positive tumors, and the benefited population is very limited[12,13]. With the use of PD1/PD-L1 inhibitors, the immunotherapy of AEG has made significant progress. However, due to the heterogeneity and complexity of the immune microenvironment, immunotherapy still has many challenges, such as hyperprogression[14]. Therefore, it is necessary to better understand the molecular mechanisms underlying AEG carcinogenesis and to identify potential prognostic indicators and drug targets.

Genomic interrogations in AEG have revealed that most AEG tumors are characterized by focal copy number variations (CNVs)[15,16]. These focal CNVs are thought of as tumorigenic factors that promote chromosomal instability in AEG tumors. The TCGA Research Network analyzed 295 primary gastric adenocarcinomas using six molecular platforms, including array-based somatic copy number analysis, whole-exome sequencing, array-based DNA methylation profiling, messenger RNA sequencing, microRNA (miRNA) sequencing, and reverse-phase protein array (RPPAR)[15]. They classified gastric cancer into for subtypes: tumors positive for Epstein–Barr virus; microsatellite unstable tumors; genomically stable tumors; tumors with chromosomal instability, which was mainly dependent on genomics data. Cristescu et al. used transcriptomics data to describe four molecular subtypes of gastric cancer, including the mesenchymal-like type, microsatellite-unstable type, and the tumor protein 53 (TP53)-active and TP53-inactive types[17]. The subtyping was primarily based on gene expression signatures. Other studies related to AEG subtyping based on omics data mainly including genomics and transcriptomics data[15,16,18–21]. In addition to various post-translational modifications, genomic changes are supposed to be translated into protein-level alterations to affect phenotypes[22,23]. Increasing attention has been given to the application of proteomics and various modified proteomics approaches in the molecular typing of tumors. Multiple studies have included mass spectrometry (MS)-based proteomics analyses of various cancers, including brain cancer[24,25], gastrointestinal cancer[18,26,27], breast cancer[28], lung cancer[29–32], and liver cancer[33,34]. These studies have revealed that proteomic signatures can provide complementary information for patient stratification and can better identify potential drug targets and disease markers. Proteomic analysis, integrated with other types of omics data, may help advance our understanding of the molecular mechanism of AEG carcinogenesis and the development of therapeutic drugs for AEG patients.

In this work, we perform comprehensive genomic, transcriptomic, proteomic, and phosphoproteomic analyses of tumor tissues and paired normal adjacent tissues (NATs) derived from 103 AEG patients. We describe integrative proteogenomic analyses of a large cohort of AEG samples and focus particularly on the clinically actionable insights revealed in the proteome and phosphorylation modifications. Based on proteomic data, we identify three different AEG subtypes that exhibit clearly significant differences in clinical and molecular features. Our study may improve current knowledge about AEG and contribute to its diagnosis, prognosis evaluation, and drug development.

## Results

### Molecular landscape of AEG tumor samples

To characterize a comprehensive molecular landscape in AEG tumors, we applied multi-omics profiling to the paired tumor and NAT samples from 103 patients (Supplementary Data 1), including proteomics profiling, phosphoproteomics profiling, WES, and RNA-seq (Fig. 1a). In particular, proteomics and phosphoproteomics profiling were performed on 206 samples. Of these 206 samples, 188 had been analyzed for WES, and 166 had corresponding RNA-seq data. In total, 30,053 non-synonymous single-nucleotide variants (SNVs) were identified in 94 AEG patients (Supplementary Data 2). In the present AEG cohort, the most frequently mutated cancer-related genes (derived from COSMIC v95)[35] were *TP53* (62%), *MUC16* (31%), *FAT4* (22%), *LRP1B* (18%), *ARID1A* (16%), and *FAT3* (16%) (Fig. 1b). We reviewed the gastro-esophageal locations of cancer and retrieved 129 samples that were regarded as AEG in the TCGA esophageal and gastric carcinoma cohort[36]. The most frequent genomic alterations in the TCGA AEG cohort were captured in our cohort (Supplementary Fig. 1a). Of note, 9 of top 10 mutated genes in our cohort were among the top mutated genes of the TCGA cohort. Genes with top 20 frequent CNVs in the TCGA cohort were also found to be frequently altered in our cohort (Supplementary Fig. 1b). The most frequent nucleotide variant across 103 AEG patients was C > T (16.7%). AEG patients of older age were found to harbor higher tumor mutation burdens (TMB) ($P = 0.045$, Wilcoxon rank sum test), while other clinicopathological features showed no obvious association with the TMB (Supplementary Fig. 2). Proteomics and phosphoproteomics data showed consistent quality across 206 samples (Supplementary Fig. 3a, b). In addition, principal component analysis of 206 proteomes showed clear divergence between AEG tumor and NAT samples and also showed heterogeneity among tumor samples (Supplementary Fig. 3c). On average, 8885 proteins (Fig. 1c) and 8445 phosphorylation sites (Fig. 1d) were identified from the 206 proteomes and phosphoproteomes of 103 AEG patients. From the RNA-seq data, 23,131 genes were found to be expressed in 166 AEG tumor and NAT samples on average (Fig. 1e). Overall, significantly more proteins ($P = 3.8E{-}15$, Wilcoxon rank sum test), phosphorylation sites ($P = 1.6E{-}4$, Wilcoxon rank sum test), and genes ($P < 2.2E{-}16$, Wilcoxon rank sum test) were detected in AEG tumors than in NAT samples (Supplementary Fig. 4). This observation indicates that compared with NATs, AEG tumors might show abnormally higher molecular activity. In summary, our multi-omics profiling presented a comprehensive molecular atlas of AEG.

### Proteomic characteristics of AEG tumors

We next investigated the disturbance of proteins in AEG tumors. Differential protein analysis revealed 2,300 upregulated and 1667 downregulated proteins in AEG tumor samples compared to paired NAT samples (Fig. 2a and Supplementary Data 3). The upregulated proteins were significantly enriched in genome regulation and instability-related biological processes, such as "spliceosome" and "DNA replication", while downregulated proteins were more enriched in metabolism-related processes, such as "oxidative phosphorylation" and "carbon metabolism" (Fig. 2b). Furthermore, the overall protein-level integrated abundances of fifty hallmark biological processes were evaluated in each sample (see Methods). Most of the hallmarks (36 out of 50, 72%) showed significantly distinct integrated abundance between paired tumor and NAT samples (Fig. 2c). For example, the "apical junction" hallmark gene set was remarkably upregulated ($P = 2.40E{-}16$), whereas the "KRAS signaling up" hallmark gene set was significantly downregulated ($P = 1.1E{-}3$) in tumor samples (Fig. 2d). Higher integrated abundances of the "apical junction" hallmark gene set indicate a worse prognosis ($P = 0.016$, log-rank test), while the higher integrated abundance of "KRAS signaling up" indicated a longer overall survival time in AEG patients ($P = 0.0033$, log-rank test) (Fig. 2e). These results revealed extensive dysregulation of hallmark

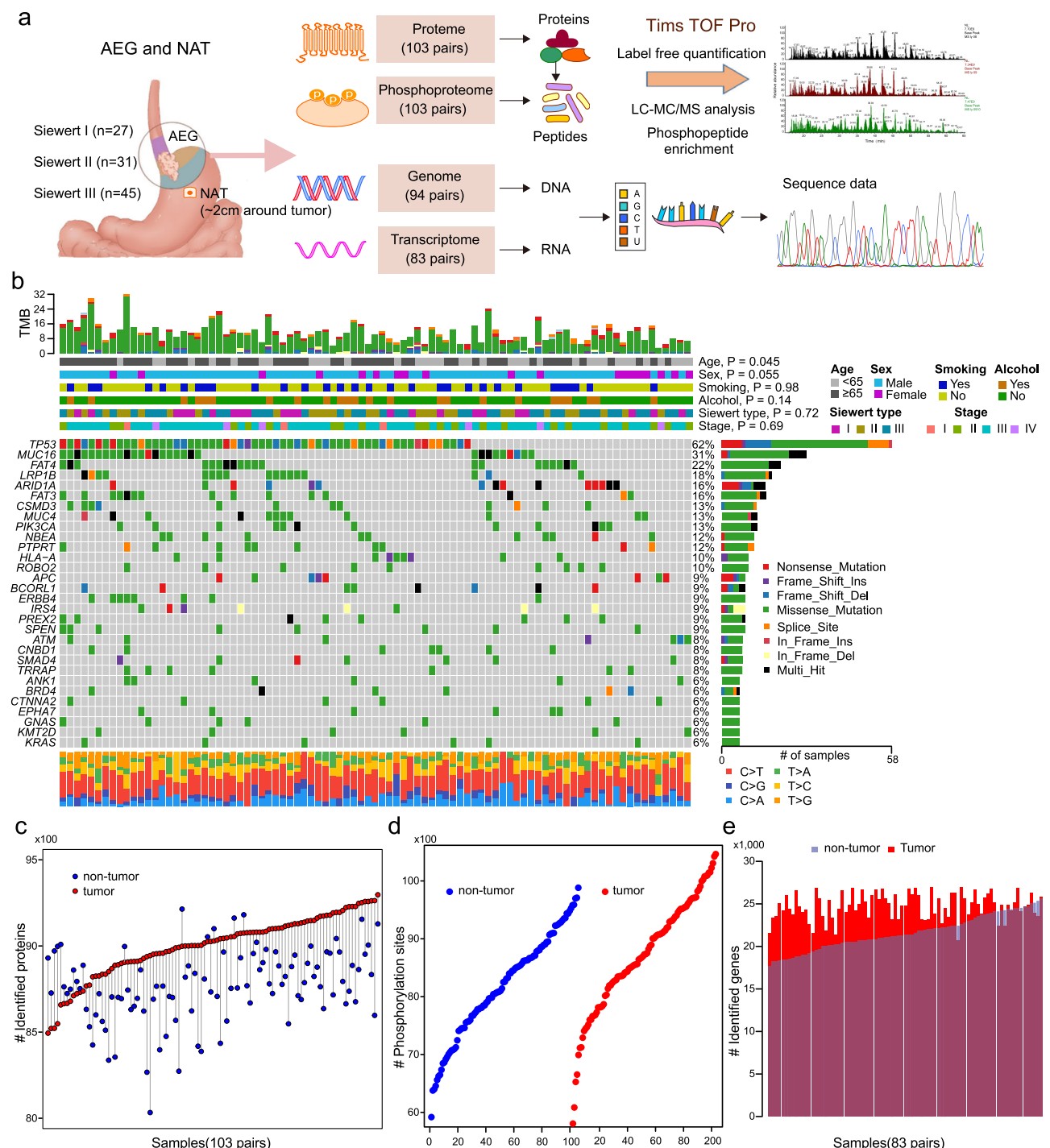

**Fig. 1 | Multi-omics landscape of adenocarcinoma of the esophagogastric junction (AEG). a** Schematic overview of the experimental design and data acquisition process for proteomics, phosphoproteomics, WES, and RNA-seq. NAT indicates normal adjacent tissue. **b** The genomic profiles of AEG patients. The top panel shows the tumor mutation burden (TMB) in each patient. The top bars show the clinicopathological features of AEG patients. The middle panel is the oncoplot generated with maftools depicting the top 30 mutated cancer-related genes in the present AEG cohort. The bottom panel shows the proportion of different types of nucleotide substitutions in each patient. The right panel represents mutation types and frequencies for each gene. *P*, two-sided Wilcoxon's rank test for age, sex, smoking, and alcohol, and Kruskal–Wallis rank sum test for Siewert type and tumor stages. **c** Overview of the proteomics profile in 103 AEG patients. **d** Overview of the phosphoproteomics profile of 206 samples from 103 AEG patients. **e** Overview of the RNA-seq profile of 83 tumors and paired NAT samples.

biological processes in AEG tumors, which also showed clinical significance. To examine whether these differentially expressed proteins (DEPs) were targeted by FDA-approved drugs or candidate anti-cancer compounds in clinical trials, we screened datasets of the Genomics of Drug Sensitivity in Cancer (GDSC)[37], Cancer Therapeutics Response Portal (CTRP)[38], and Broad Institute Drug Repurposing project[39]. Of these DEPs, 252 were found to be targeted by FDA-approved drugs or candidate drugs that are currently under clinical trials (Supplementary Data 4 and Supplementary Fig. 5a). For example, the AHR protein, which could be inhibited by flutamide, was significantly upregulated in tumor samples (Supplementary Fig. 5b). AEG patients with high AHR protein levels showed markedly shorter (*P* = 6.7E−3, log-rank test)

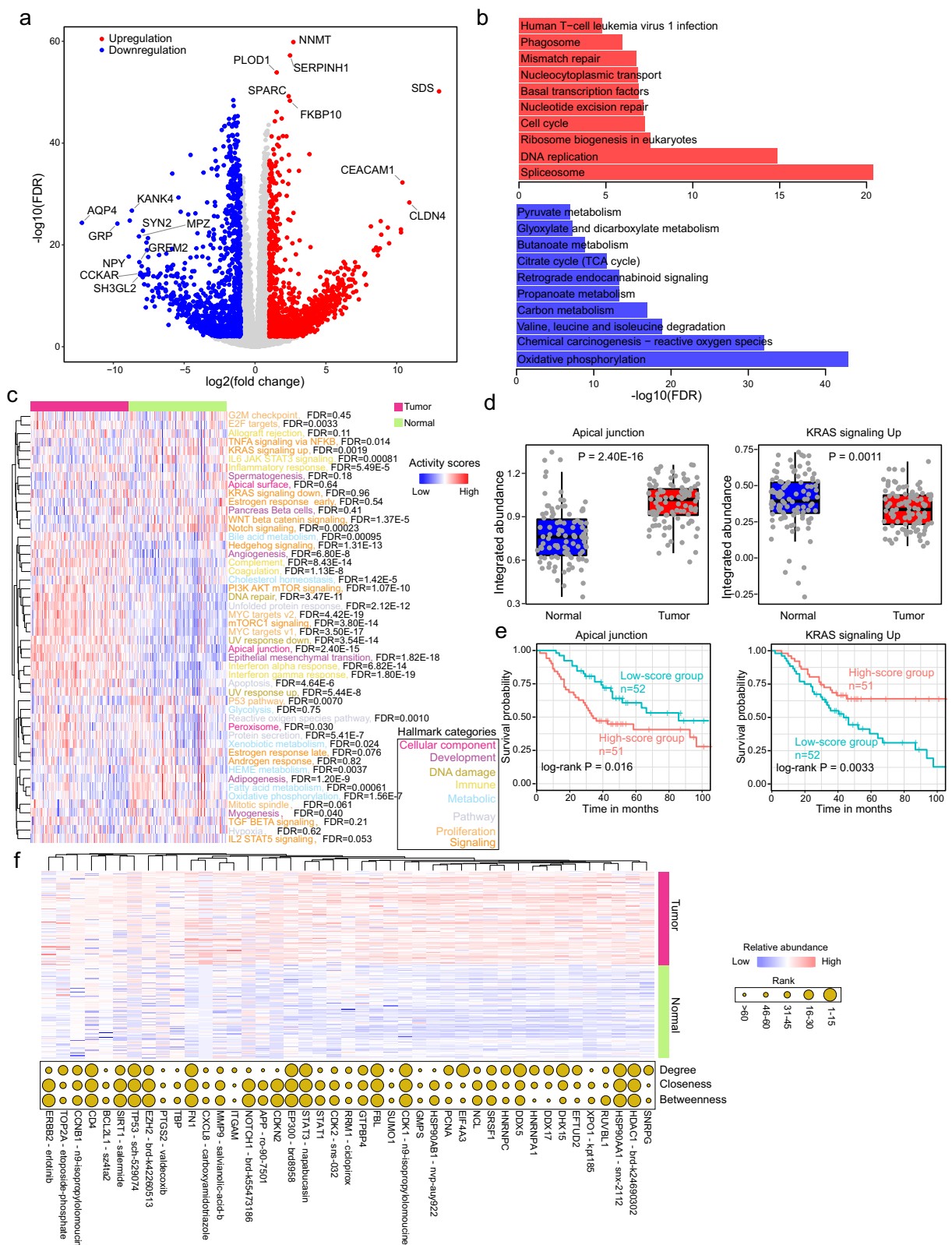

overall survival times than those with low levels (Supplementary Fig. 5c). To identify proteins that may play crucial roles in AEG, and can be potential drug targets, we constructed the protein-protein interactions (PPI) network of DEPs (see Methods). A PPI network of 3923 nodes and 79,088 edges was obtained (Supplementary Fig. 6a). The network topology was further analyzed to identify hub proteins, including the degree, closeness and betweenness (Supplementary

Fig. 6b–d and Supplementary Data 5). To further optimize the list of protein candidates, we mapped the top 50 DEPs with the top 50 proteins with the largest degree, closeness, or betweenness, some of which were also found to be targeted by known anti-cancer compounds, such as HDAC1, HSP90AA1, and TP53 (Fig. 2f). Our analysis presented a comprehensive view of proteomic alterations in AEG tumors, and further investigation on their functions and molecular

**Fig. 2 | Proteomic variations in AEG tumors. a** Volcano plot showing the difference in proteins between AEG tumor and paired NAT samples. Red circles represent upregulated proteins (FDR < 0.01 and log2(fold change) > 1), and blue circles indicate down-regulated proteins (FDR < 0.01 and log2(fold change) < −1). **b** Functional enrichment results of upregulated and downregulated proteins, respectively. **c** Heatmap showing the difference in the protein abundance of hallmarks between AEG tumor and paired NAT samples. Different font colors indicate different hallmark categories. FDR, adjusted $P$ from Wilcoxon's rank-sum test. **d** Comparisons of integrated abundances of "apical junction" and "KRAS signaling up" between AEG tumor and paired NAT samples ($n = 103$). Each box represents the IQR and median of the hallmark scores in each group, whiskers indicate 1.5 times IQR. $P$, two-sided Wilcoxon's rank-sum test. **e** Kaplan–Meier survival curves comparing groups with high ($n = 51$) and low ($n = 52$) abundance of "apical junction" and "KRAS signaling up" gene sets, respectively. **f** Heatmap showing the difference in the proteins that are included in at least two sets of top 50 DEPs, top 50 proteins with the largest degree, closeness, or betweenness. Bubble plot on the right shows the degree, closeness, or betweenness of the corresponding proteins in the PPI network. $^{*}P < 0.05$, $^{**}P < 0.01$, $^{***}P < 0.001$, two-sided Wilcoxon's rank-sum test.

mechanisms in AEG may provide promising drug targets for this disease.

## Proteomics-based subtyping of AEG tumors

The proteomic heterogeneity among tumor samples inspired us to explore AEG subtypes based on proteomics data. A NMF algorithm was employed to cluster AEG tumor samples by using proteomics data (see Methods). Three different subtypes were identified, with 40 samples in the S-I subtype, 23 samples in the S-II subtype, and 40 samples in the S-III subtype (Fig. 3a and Supplementary Data 6). Clinicopathological characteristics, including age, sex, smoking, alcohol, Siewert type and tumor stage, exhibited no significant differences between these three AEG subtypes except for age and Siewert type. The S-I subtype was significantly associated with older age (75% ≥65 years old, $P = 0.0093$, Fisher's exact test). The Siewert type II patients were more enriched in the S-I subtype, while the S-III subtype had many more Siewert type III patients ($P = 0.011$, Fisher's exact test). Patients in these three subtypes showed significantly distinct overall survival times ($P = 0.0011$, log-rank test), wherein S-III patients had the longest survival time and S-I patients had the shortest survival time (Fig. 3b). The proteomics-based AEG subtyping remained an independent prognostic factor when adjusted for other clinicopathological characteristics in multivariate Cox regression analysis ($P = 0.002$, Supplementary Fig. 7). The top mutated genes showed clear distinctions among these three subtypes (Supplementary Fig. 8a–c). We next compared gene mutation frequencies among these three subtypes and found 97, 143, and 29 specifically mutated genes in the S-I, S-II, and S-III subtypes, respectively (Fig. 3C and Supplementary Data 7). For example, *LEPR* mutation was most common in the S-I subtype (OR = 20.1, $P = 2.8E{-}4$, Fisher's exact test), *NCKAP1* mutation was most common in the S-II subtype (OR = 10.5, $P = 5.8E{-}3$, Fisher's exact test), and *WIZ* mutation was most common in the S-III subtype (OR = 10.0, $P = 7.5E{-}3$, Fisher's exact test) (Supplementary Fig. 8d). To further integrate the genomics and proteomics data, we examined how subtype-specific mutations influence proteins (Supplementary Fig. 9 and Supplementary Data 8). The consequence of mutation on protein was evaluated by comparing the T/N (tumor/normal) values between mutation and wild-type samples as described in a previous study[32]. For each mutated gene, we examined changes of all the possible proteins. We identified 65,184, 3900, and 1146 significant mutation-to-protein associations in the S-I subtype, S-II subtype, and S-III subtype, respectively (Supplementary Fig. 9a). In all three subtypes, over 60% are negative associations, i.e., most mutations directly or indirectly led to the decrease of protein levels. We showed the top five mutation-protein associations of the top five mutated genes in Supplementary Fig. 9b–d. Although tumor samples exhibited dysregulation of integrated protein abundance of hallmarks in all subtypes, samples in the S-II subtype showed a decreased degree of change (Fig. 3d). The S-III subtype not only displayed a higher degree of dysregulation in tumor samples but also showed a substantial difference in abundance than the S-I and S-II subtypes. For example, the integrated abundance of the "G2M checkpoint" hallmark in the S-III subtype was significantly greater than that in the other two subtypes ($P = 1.7E{-}3$ compared to S-I subtype, $P = 1.2E{-}4$ compared to S-II subtype, Student's $t$ test) (Fig. 3e), while "pancreas beta cells" showed markedly lower levels in the S-III subtype ($P = 1.7E{-}2$ compared

to S-I subtype, $P = 4.3E{-}2$ compared to S-II subtype, Student's $t$ test) (Fig. 3f). To further investigate the protein features in specific subtypes, we identified subtype signature proteins that showed subtype-specific high expression patterns (see "Methods"). Briefly, the expression levels of signature proteins in specific subtypes were significantly higher in tumor samples than in all NAT samples and tumor samples of the other subtypes. Our analysis found 36, 54, and 10 signature proteins in the S-I, S-II, and S-III subtypes, respectively. Of these, 12 signature proteins showed a significant association with patient survival time in the univariate Cox regression analysis (Fig. 3g). Seven of these 12 signature proteins showed significant prognostic associations in AEG patients (Supplementary Fig. 10). In the multivariate Cox regression analysis, FBXO44 was the most unfavorable risk factor according to the risk score, while PKD2 and CD3D were potent favorable factors. In summary, our proteomics analysis identified three different AEG subtypes that exhibited molecular and clinical distinctions.

## FBXO44 promotes AEG tumor progression and metastasis

In the multivariate Cox regression analysis above, FBXO44 showed a significantly high unfavorable risk score (Fig. 3g), which was a valuable candidate for further investigation. FBXO44 is a member of the F-box protein family that has been shown to play roles in human cancers[40]. The FBXO44 gene showed significant dysregulation in eight of 18 different tumor types from TCGA cohorts (Supplementary Fig. 11a). FBXO44 showed upregulation in colon cancer but showed no significant expression change in stomach cancer. The FBXO44 protein exhibited significantly higher abundance in S-II AEG tumor samples than in S-II normal samples ($P = 1.1E{-}4$, Student's $t$ test), S-I tumor samples ($P = 2.3E{-}3$, Student's $t$ test), and S-III tumor samples ($P = 5.3E{-}4$, Student's $t$ test) (Fig. 4a). The upregulation of FBXO44 protein in tumor samples was further validated in an independent clinical cohort of 251 AEG patients ($P = 1.55E{-}4$, Student's $t$ test) (Fig. 4b and Supplementary Fig. 11b). Our analysis in this cohort found that FBXO44 was significantly associated with distant metastasis ($\chi^2 = 6.19$, $P = 0.013$) and advanced TNM stage ($\chi^2 = 8.95$, $P = 0.030$) of AEG tumor (Supplementary Fig. 12). Furthermore, we also assessed the association between FBXO44 protein level and all other available clinicopathological features of AEG patients (Supplementary Data 9). In addition to distant metastasis and advanced TNM stage, FBXO44 was found to be highly associated with older age ($\chi^2 = 5.507$, $P = 0.019$) and high AFP level ($\chi^2 = 14.489$, $P < 2.00E{-}16$). AEG patients with high levels of FBXO44 showed significantly shorter survival times than those expressing low levels of FBXO44 in both the present cohort ($P = 1.5E{-}2$, log-rank test) (Fig. 4c) and the other independent clinical cohort of 251 AEG patients ($P = 7.0E{-}3$, log-rank test) (Fig. 4d). To further confirm the role of FBXO44 in AEG, we performed overexpression (OE) and knockdown (KD) of FBXO44 in two different AEG cell lines, OE19 and SK-GT-4. In OE19 and SK-GT-4 cells, FBXO44 OE promoted cell proliferation by 1.79-fold ($P = 0.031$) and 1.48-fold ($P = 0.029$) (Fig. 4e and Supplementary Fig. 11c), increased cell invasion by 1.68-fold ($P = 0.032$) and 2.18-fold ($P = 0.035$) (Fig. 4f and Supplementary Fig. 11d), and enhanced cell migration by 2.13-fold ($P = 0.004$) and 1.18-fold ($P = 0.018$) (Fig. 4g and Supplementary Fig. 11e), respectively, compared to control cells. In contrast, FBXO44 KD inhibited cell

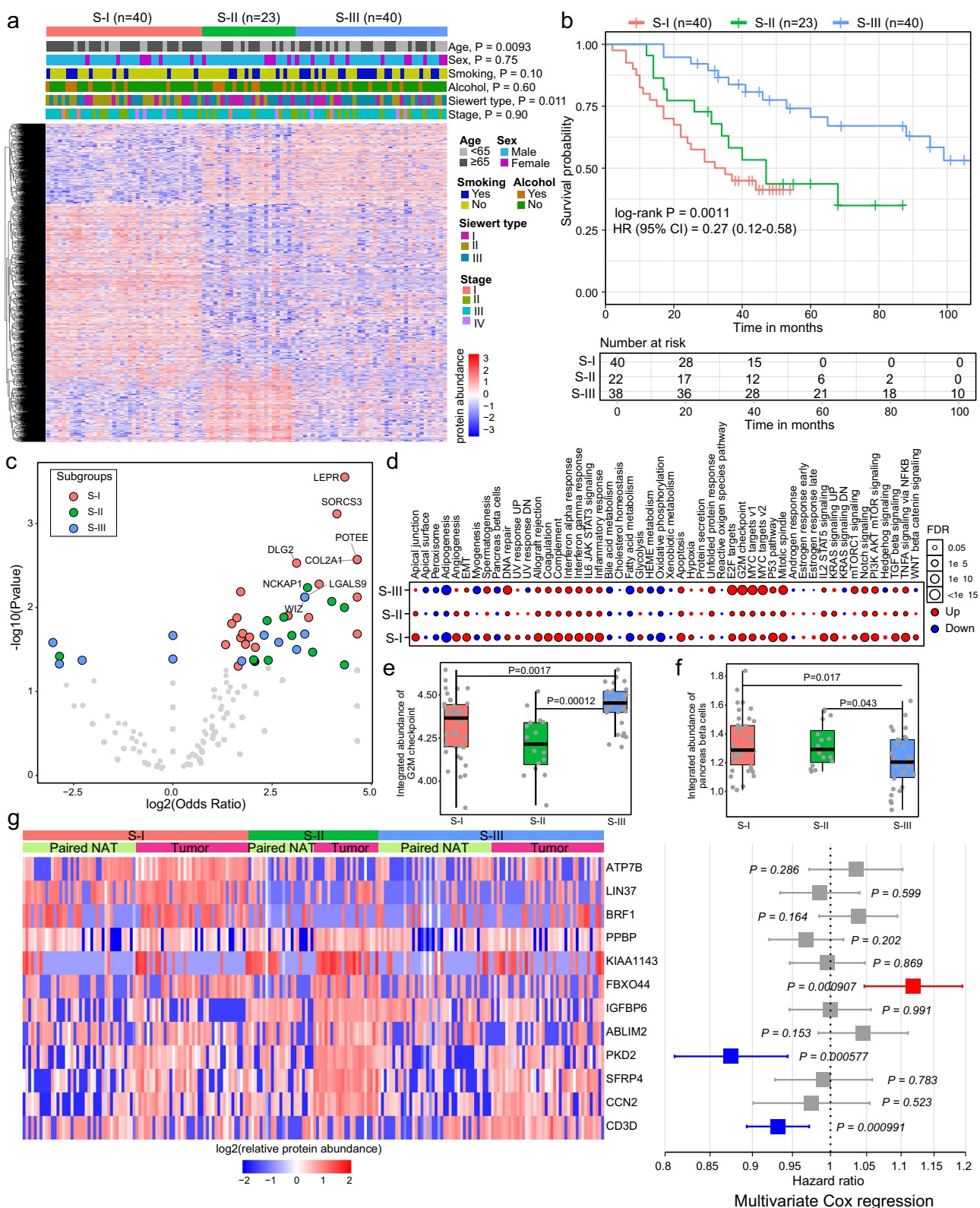

proliferation by 68.1% ($P = 0.002$) and by 49.1% ($P = 0.005$) (Fig. 4e and Supplementary Fig. 11c), decreased cell invasion by 79.3% ($P = 0.008$) and 70.9% ($P = 0.001$) (Fig. 4f and Supplementary Fig. 11d), and reduced cell migration by 71.8% ($P = 0.005$) and 54.7% ($P = 0.003$) (Fig. 4g and Supplementary Fig. 11e) in OE19 and SK-GT-4, respectively. The oncogenic role of FBXO44 in AEG was further validated in the OE19 xenograft mouse model. We observed that FBXO44 OE increased the growth of AEG xenograft tumors by 2.54-fold ($P = 0.004$), whereas

FBXO44 KD suppressed tumor growth by 67.17% ($P = 0.029$) in vivo (Fig. 4h and Supplementary Fig. 11f−h). Similar results were also observed in an OE19 orthotopic AEG mouse model (Fig. 4i, j, and Supplementary Fig. 11i, j). Of note, FBXO44 OE not only enhanced tumor growth but also increased the incidence of liver metastasis. In conclusion, our analysis revealed that a high level of FBXO44 expression is associated with a poor prognosis in AEG patients and promotes the growth and metastasis of AEG tumor cells in vitro and in vivo.

**Fig. 3 | Proteomic subtyping of AEG tumors. a** Heatmap showing the differentially expressed proteins among the three subtypes. Tiling bars above the heatmap show the distribution of different clinicopathological characteristics among the three subtypes. *P*, Fisher's exact test. **b** Kaplan–Meier survival curve comparing patients in different subtypes ($n = 40$ for S-I, $n = 23$ for S-II, $n = 40$ for S-III). The hazards ratio (HR) with 95% confidence interval (CI) is also shown. **c** Volcano plot showing the difference in subtype-specific mutated genes. *P*, Fisher's exact test. **d** The differences in integrated protein abundances of hallmarks comparing tumor and NAT samples in each subtype. **e** Comparison of the integrated abundance of "activity of G2M checkpoint" among three subtypes ($n = 40$ for S-I, $n = 23$ for S-II, $n = 40$ for S-III). *P*, two-sided Wilcoxon's rank-sum test. **f** Comparison of the integrated

abundance of "activity of MYC targets" gene set among the three subtypes ($n = 40$ for S-I, $n = 23$ for S-II, $n = 40$ for S-III). *P*, two-sided Wilcoxon's rank-sum test. **g** Twelve signature proteins that are significantly associated with patient survival. The heatmap on the left shows the relative abundance of signature proteins in tumor and paired NAT samples of each subtype. The forest plot on the right shows the prognostic score for each protein in multivariate Cox regression analysis. The middle points indicate hazard ratios. The endpoints represent lower or upper 95% confidence intervals. Red indicates unfavorable proteins, while blue indicates favorable proteins. *P*, multivariate Cox proportional-hazards. In **e** and **f**, each box represents the IQR and median of integrated abundance in each subtype, whiskers indicate 1.5 times IQR.

## Genomic differences among different AEG subtypes

We further examined the genomic alterations between different AEG proteomics subtypes. Mutation signatures were separately extracted in AEG subtypes (see Methods). These three subtypes showed shared and specific mutation signatures (Fig. 5a–c). In particular, S-I and S-II shared the SBS3 signature (Fig. 5a, c), which indicates defects in DNA double-strand break (DSB) repair by homologous recombination (HR). Both the S-II and S-III subtypes exhibited SBS6 mutation signatures that represent defective DNA mismatch repair (Figs. 5b and 5c). The SBS17b mutation signature was shared by the S-I and S-III subtypes (Fig. 5a, c), which displayed an exclusively high frequency of T > G nucleotide substitution. The SBS1 signature was specifically identified in the S-I subtype, which showed spontaneous or enzymatic deamination of 5-methylcytosine (Fig. 5a). The S-II subtype exclusively exhibited the mutation signature of APOBEC cytidine deaminase (the SBS2 signature) (Fig. 5b). The mutation signature of "deficiency in base excision repair due to inactivating mutations in NTHL1" (the SBS30 signature) was specifically detected in the S-III subtype (Fig. 5c). To further characterize subtype-specific genomic features, we separately conducted somatic interaction analyses in different AEG subtypes. We identified 21, 12, and 19 co-occurrence mutated gene pairs in the S-I, S-II, and S-III subtypes, respectively (Fig. 5d–f). Moreover, 2 and 4 mutually exclusive mutated gene pairs were found in the S-II and S-III subtypes, respectively. In particular, *CSMD1* and *ANKRD36C* genes showed significant mutation co-occurrence across patients in the S-I AEG tumor subtype (Fig. 5d and Supplementary Fig. 13a). Co-occurring mutations of the *MUC4* and *CPED1* genes were specifically identified in the S-II subtype (Fig. 5e and Supplementary Fig. 13b). Mutations in *FAT4* and *PRKDC* genes showed significant co-occurrence across AEG patients in the S-III subtype (Fig. 5f and Supplementary Fig. 13c). In addition, *RYR2* and *TTN* were found to be exclusively mutated in the S-III AEG subtype (Fig. 5f and Supplementary Fig. 13d). Apart from being distinctive features among different AEG subtypes, co-occurring or exclusive mutations also implicate potential therapeutic strategies that pharmacologically target both genes or either gene of the related gene pair. Furthermore, known oncogenic pathways were examined in AEG tumors. The most frequently mutated oncogenic pathways in all subtypes were the "TP53", "RTK-RAS", and "Hippo" pathways (Fig. 5g). Although gene mutations in the "RTK-RAS" pathway were found in over half of the samples for individual subtypes, remarkably different sets of genes were affected in distinct subtypes (Fig. 5h). In conclusion, AEG subtypes showed clearly distinguishable genomic characteristics that might suggest different etiologic mechanisms and precision treatments for individual subtypes.

## Immune infiltration in AEG tumors

To investigate the heterogeneity of the tumor microenvironment in AEG tumors, we performed cell type deconvolution analysis based on RNA-seq data. The xCell algorithm was employed to infer the relative cell abundance of 41 different cell types (see Methods). The infiltration of some cell types showed significant differences between the three AEG subtypes, such as regulatory T cells and fibroblasts, but none of them have associations with clinicopathological features of AEG

patients (Fig. 6a). The S-II AEG tumor samples showed lower abundance of gamma delta T cells, regulatory T cells, and plasmacytoid dendritic cells, whereas they had higher infiltration of fibroblasts, lymphatic endothelial cells, and microvascular endothelial cells, compared to those of the S-I and S-III subtype (Supplementary Fig. 14). Comparisons of cell abundances between tumor and NAT samples in each subtype revealed pervasive changes in cell abundances across various cell types (Fig. 6b). Compared to the corresponding NAT samples, tumors in the S-II subtype had the least number of cell types, while the S-III subtype had the most cell types that showed alterations in cell abundance, especially the increase in lymphoid and myeloid cells. Some types of cells exhibited dysregulated abundances in all AEG subtypes. For example, the abundance of activated dendritic cells (aDCs) showed a significant increase in tumor samples of all three AEG subtypes (Fig. 6c). The abundance of fibroblasts was significantly decreased in the S-III subtype (FDR = 2.6E−4, Student's *t* test) but showed no obvious changes in tumor samples from the S-I (FDR = 0.48, Student's *t* test) and S-II (FDR = 0.98, Student's *t* test) subtypes (Fig. 6d). Compared to samples in the S-I and S-II subtypes, our H&E analysis also revealed a decrease in fibroblast abundance of the S-III subtype (Fig. 6e). Given that fibroblasts may limit the immune cell infiltration to exert the immunosuppressive role in cancer[41], this observation may partly explain that AEG patients in the S-I and S-II subtype had worse prognosis than those in the S-III subtype. Furthermore, we examined the expression changes in immune checkpoint genes, which were retrieved from a previous study[42]. Some immune checkpoints, such as *CEACAM1*, *CD276*, *PLEC*, *HLA-DRB1*, and *LAIR1*, were consistently up-regulated in all three subtypes (Fig. 6f). Subtype-specific dysregulation of immune checkpoints, such as the upregulation of *CD200* and downregulation of *TNFSF14* in the S-II subtype, was also observed. We further evaluated the associations between FBXO44 and immune cells or checkpoints. The high expression of FBXO44 was found associated with the low infiltration of Th2 cells and CD4⁺ Tem cells (Supplementary Fig. 15a, b), and also correlated with the high expression of immune checkpoints *TNFRSF14*, *TNFRSF25*, *CD40*, and *VTCN1* (Supplementary Fig. 15c, d). Our analysis revealed the heterogeneity of tumor microenvironment infiltration and immune checkpoints, which suggested potential common and subtype-specific immunotherapy strategies for AEG patients.

## Phosphoproteomic characterization of AEG tumors

We next investigated the alterations of phosphorylation modifications and kinase activity in AEG tumors. Differential phosphorylation analysis identified 4932 sites with increased phosphorylation (fold change > 1.5 and FDR < 0.05) and 3146 sites with decreased phosphorylation (fold change < 0.67 and FDR < 0.05) in tumor samples (Fig. 7a and Supplementary Data 10). Furthermore, sites with differential phosphorylation were identified in each subtype, revealing 1930, 601, and 2111 sites with increased phosphorylation and 1472, 645, and 1580 sites with decreased phosphorylation in the S-I, S-II, and S-III subtypes, respectively (Supplementary Data 11). The differentially phosphorylated proteins in the S-I and S-II subtypes were enriched in nuclear transport and cell organization,

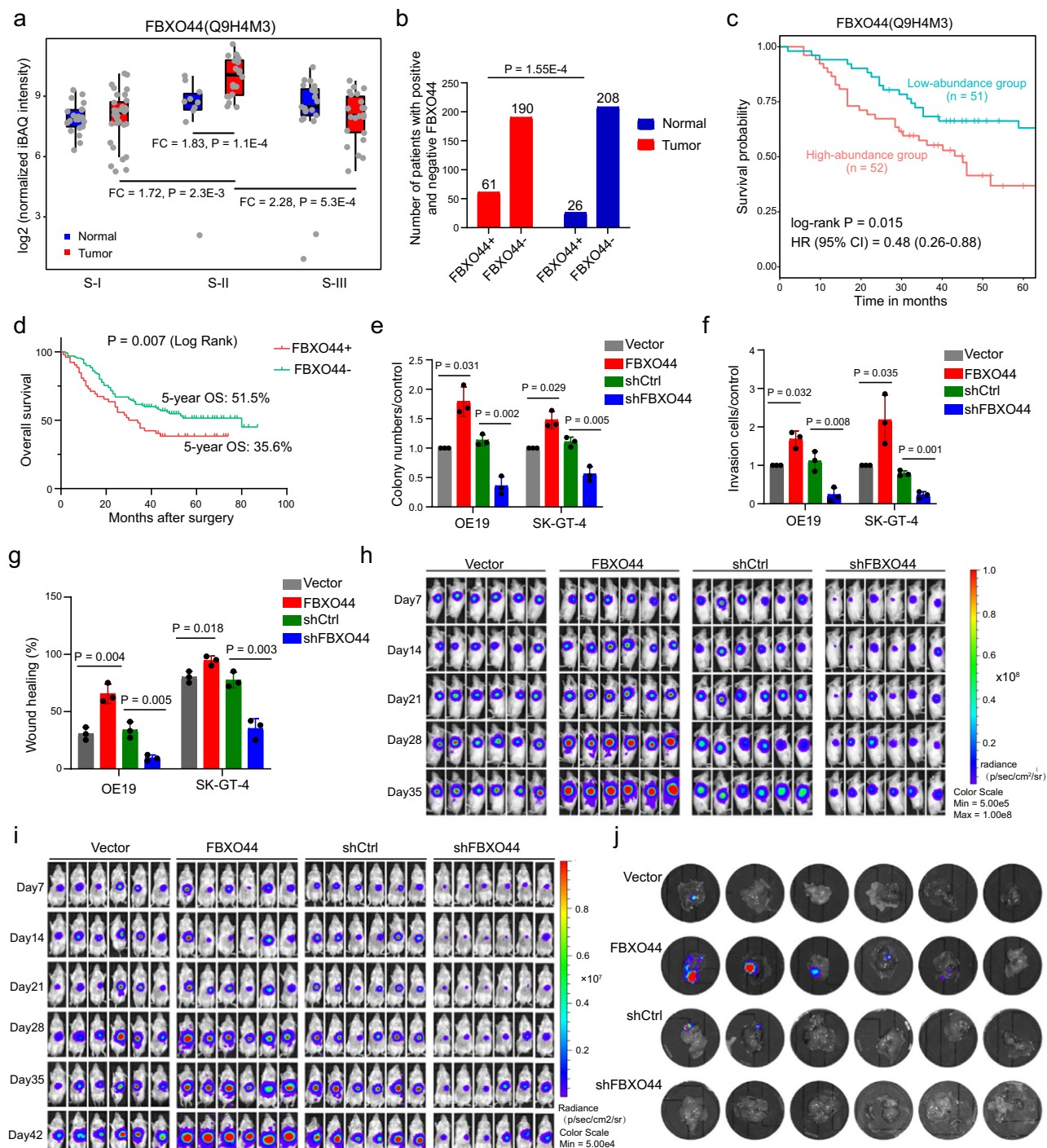

**Fig. 4 | Clinical relevance and biological functions of FBXO44. a** Comparison of FBXO44 protein abundance between tumor and NAT samples in each subtype and tumors from different subtypes. FC indicates fold change ($n$ = 40 for S-I, $n$ = 23 for S-II, $n$ = 40 for S-III). Each box represents the IQR and median of normalized protein abundance in normal or tumor samples of each subtype, whiskers indicate 1.5 times IQR. $P$, two-sided Wilcoxon's rank-sum test. **b** The distribution of FBXO44 in an independent cohort of tumor and NAT samples from 251 AEG patients. $P$, chi-squared test. **c** Kaplan–Meier survival curve comparing FBXO44-high ($n$ = 52) and -low ($n$ = 51) abundance patients. $P$, log-rank test. **d** Kaplan–Meier survival curve of FBXO44 in an independent clinical cohort ($n$ = 61 for FBXO44+, $n$ = 190 for FBXO44−). $P$, log-rank test. **e** Cell proliferation assays of FBXO44 OE or KD OE19 and

SK-GT-4 cell lines ($n$ = 3 biological replicates). $P$, two-sided Student's $t$ test. **f** Transwell invasion assays of FBXO44 OE or KD OE19 and SK-GT-4 cell lines ($n$ = 3 biological replicates). $P$, two-sided Student's $t$ test. **g** Cell migration assays of FBXO44 OE or KD OE19 and SK-GT-4 cell lines ($n$ = 3 biological replicates). $P$, two-sided Student's $t$ test. **h** Representative bioluminescent images of mice bearing OE19 xenograft tumors harboring FBXO44 OE or FBXO44 KD or their corresponding controls at different time points after injection. **i** Representative bioluminescent images of mice bearing orthotopic OE19 tumors harboring FBXO44 OE or FBXO44 KD or their corresponding controls at different time points post implantation. **j** Representative images of liver metastasis in mice bearing orthotopic OE19 tumors. In **e**–**g**, error bars represent mean ± SDs.

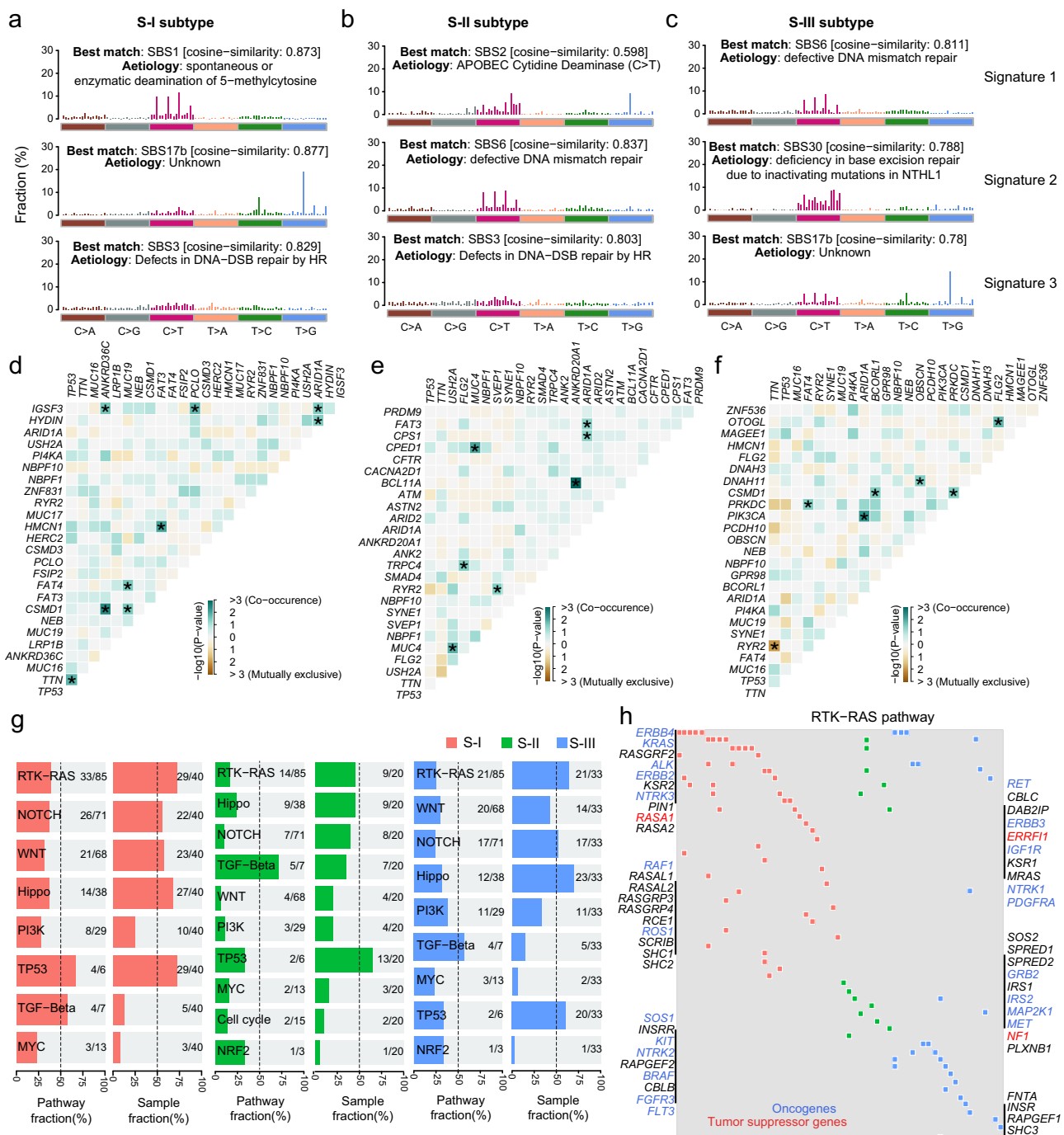

**Fig. 5 | Comparisons of genomic features among the three proteomic subtypes.** Mutation signatures and the best match SBS signatures in the S-I (**a**), S-II (**b**), and S-III (**c**) subtypes. Somatic interactions of the top mutated genes in the S-I (**d**), S-II (**e**), and S-III (**f**) subtypes. Asterisks indicate statistical significance (Fisher's exact test). **g** Fractions of affected pathways and samples by gene mutations in major oncogenic pathways. **h** Oncoplot of the RTK-RAS pathway in the three subtypes. Red font indicates tumor suppressor genes and blue font represents oncogenes.

whereas those in the S-III subtype were enriched in chromatin modification and organization (Fig. 7b). A large proportion (35.5%) of differentially phosphorylated sites that were identified in all AEG tumor samples showed no obvious dysregulation in all subtypes (Supplementary Fig. 16a). Specifically, 2040 sites with increased phosphorylation and 1078 sites with decreased phosphorylation exhibited no significant changes in all three subtypes (Supplementary Fig. 16b). The kinase activities were then interpreted based on the differentially phosphorylated sites in each AEG subtype. Kinase-substrate enrichment analysis was performed to detect

enriched kinases in different subtypes. Different AEG subtypes were enriched for distinct lists of kinases, and the same kinases showed different levels of activities in the S-I, S-II, or S-III subtypes (Fig. 7c). CDK2 and CDK7 were highly enriched in all three subtypes. The S-I subtype specifically showed enrichment of IKBKB and PRKDC. HIPK2 kinase was exclusively enriched in the S-II subtype, while CHEK2 and AURKB were specifically enriched in the S-III AEG subtype. Based on the correlations of known kinase-phosphosubstrate pairs (see Methods), we separately constructed the kinase-phosphosubstrate regulatory networks in three AEG

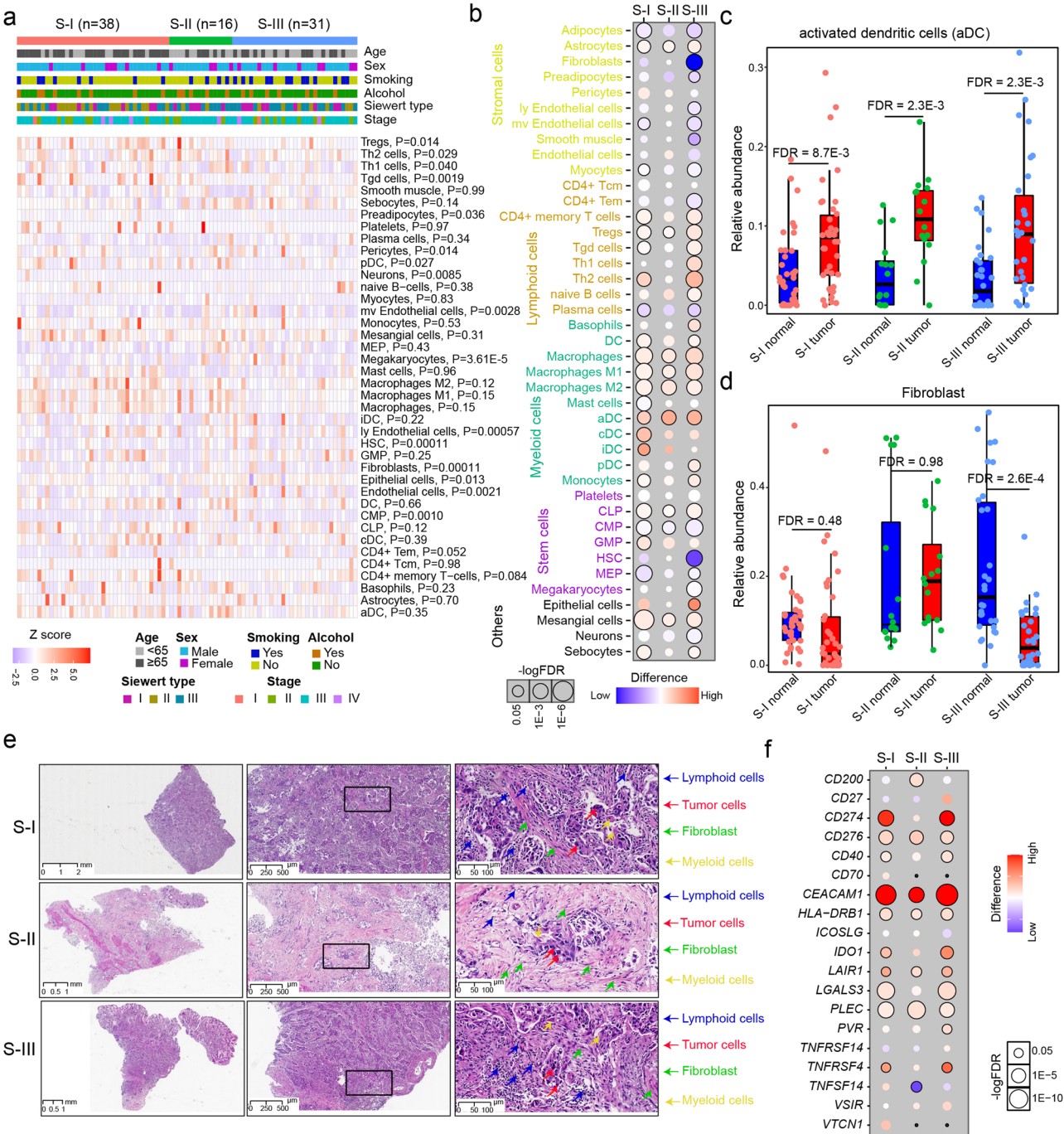

**Fig. 6 | Immune infiltration across different proteomic subtypes. a** Heatmap showing the relative abundance of different cells across samples of the three AEG subtypes. The Kruskal−Wallis Rank Sum test was used to compare the differences between subtypes. *P*, Kruskal−Wallis rank sum test. **b** The difference in the relative abundance of different infiltrating cells in the three AEG subtypes. FDR, Wilcoxon's rank-sum test. **c** Comparisons of aDC abundance between AEG tumor and NAT samples in the S-I, S-II, and S-III subtypes (*n* = 40 for S-I, *n* = 23 for S-II, *n* = 40 for S-III). FDR, Wilcoxon's rank-sum test. **d** Comparisons of fibroblast abundance

between AEG tumor and NAT samples in the S-I, S-II, and S-III subtypes (*n* = 40 for S-I, *n* = 23 for S-II, *n* = 40 for S-III). FDR, Wilcoxon's rank-sum test. **e** H&E analysis of tumor cells, lymphoid cells, myeloid cells and fibroblasts across three AEG subtypes. Scale bars used for 0×, 40×, and 200× magnification were 1, 500, and 100 μm, respectively. **f** The differential significance of the protein expression of immune checkpoints across the three AEG subtypes. In **c** and **d**, each box represents the IQR and median of the relative cell abundance in normal or tumor samples of each subtype, whiskers indicate 1.5 times IQR.

---

proteomic subtypes (Fig. 7d−f and Supplementary Data 12). In both S-I and S-III subtypes, CDK1 exhibited the most significant correlations with its phosphosubstrates (Fig. 7d, f). CDK2 showed significantly positive correlations with 18 and 28 phosphosubstrates in the S-I and S-III subtypes, while no remarkable correlations were detected in the S-II subtype (Fig. 7e). We observed only one remarkable kinase-phosphosubstrate pair in the S-II subtype,

wherein CSNK2A1 was significantly associated with the phosphorylation of Occludin S408 (*P* = 3.5E−2). Significant correlations of some kinases were found in specific subtypes, such as ATR in the S-I subtype and MAPK3 in the S-III subtype. Conclusively, our analysis revealed differences in kinase-phosphosubstrate regulatory networks between different subtypes and suggested potential personalized responses to clinical therapeutics for AEG patients.

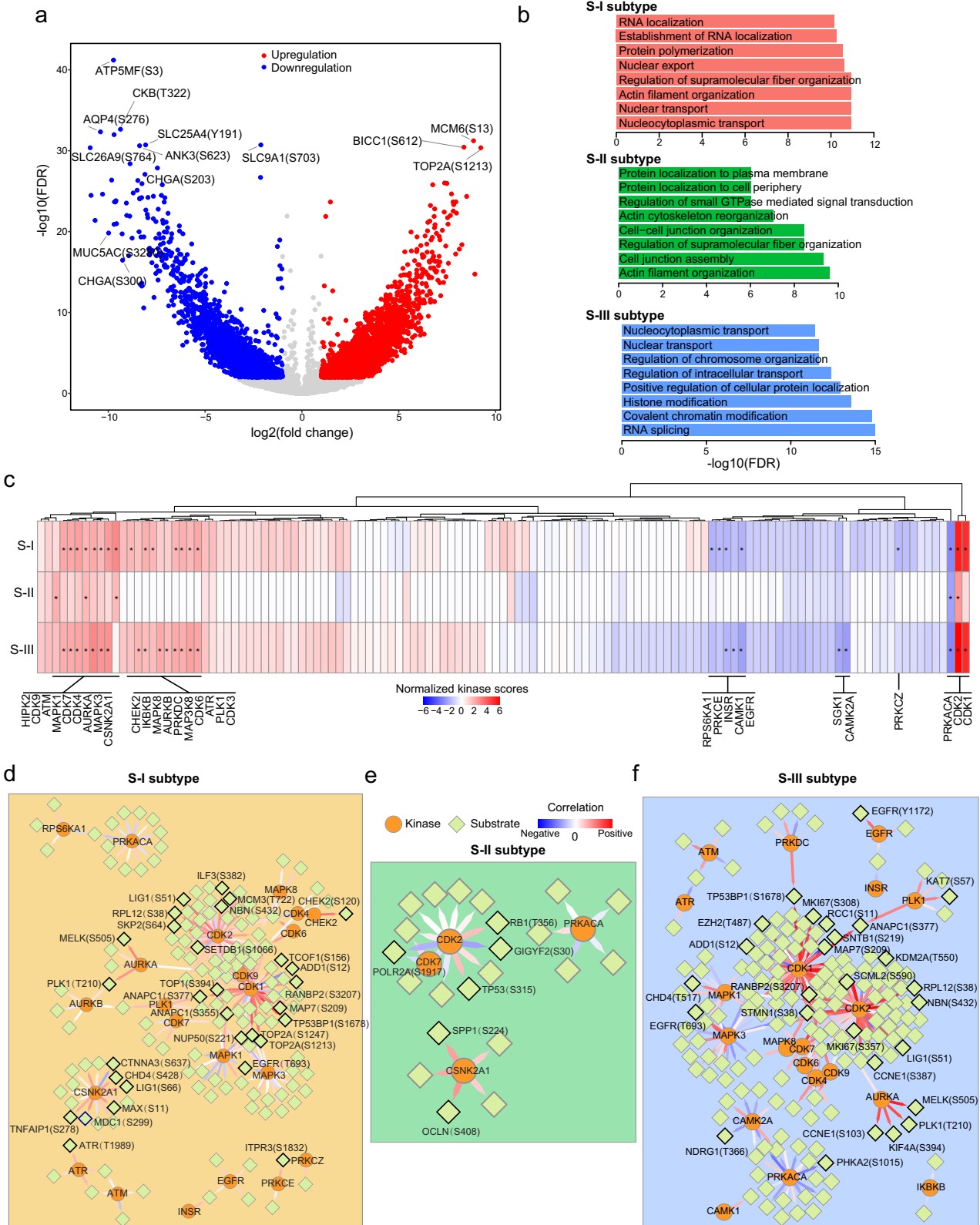

**Fig. 7 | Phosphoproteomic analyses in three AEG subtypes. a** Volcano plot showing the differential significance of phosphorylation sites. Red circles represent sites with increased phosphorylation (FDR < 0.01 and log2(fold change) > 1) and blue circles indicate downregulated phosphorylation sites (FDR < 0.01 and log2(fold change) < −1). **b** Enriched biological processes of differentially phosphorylated sites in each subtype. **c** Kinase enrichment of differentially phosphorylated sites in each AEG tumor subtype. *P*, Fisher's exact test. Asterisks (*) represent statistical significance (*P* < 0.05). Kinase-phosphosubstrate regulatory networks in tumors of the S-I (**d**), S-II (**e**), and S-III (**f**) subtypes.

## Discussion

AEG is a gastroesophageal cancer whose incidence has notably risen in recent decades. However, there has been a lack of molecular classification and systematic characterization for AEG, which prevents the development of effective therapeutic strategies[2,13]. Our study represents the attempt at proteomics-based multi-omics profiling for AEG tumors, including genomics, transcriptomics, proteomics, and phosphoproteomics. We presented the proteogenomic alterations in AEG tumors and classified AEG into three different subtypes based on proteomics data. These three AEG subtypes significantly differ in terms of clinical prognosis and molecular alterations.

It is well recognized that molecular subtyping has greatly improved our understanding of inter- and intra-tumor heterogeneity and promoted the development of personalized oncotherapy[15,43,44]. Based on proteomics data, three different AEG subtypes were identified in our study. Patients with the S-I subtype have the shortest survival, whereas those with the S-III subtype have the longest survival. Stratification of patients based on survival time will help with precise clinical management and intervention strategies. Furthermore, we compared molecular features among these AEG subtypes. We identified signature proteins that exhibited exclusive high expression in specific subtypes, which could be used for subtype differential diagnosis and as potential targets of personalized treatments. Of these signature proteins, some were found to be significantly associated with AEG tumor progression. For example, FBXO44 was specifically upregulated in the S-II subtype, and its high expression is closely related to a poor prognosis in AEG patients. We experimentally validated that FBXO44 could promote the proliferation and metastasis of AEG tumor cells in vitro and in vivo. A recent study demonstrated that FBXO44 is an essential repressor of DNA replication-coupled repetitive elements in human cancer[40]. The same study also showed that FBXO44 inhibition could enhance the response to anti-PD-1 therapy in immunocompetent mice bearing 4T1 cell-derived tumors. Combining our observations in AEG tumors, these results suggest that FBXO44 inhibition might overcome anti-PD-1 resistance in AEG tumors, especially for patients with the S-II subtype.

It is known that molecular alterations occurred frequently in tumor samples, but the specific alterations of proteome in AEG tumor have not yet systematically investigated. Pairwise comparisons of tumor and NAT around tumor sites are common in many multi-omics studies in gastric or colon cancer[18,26,27]. By comparing to the normal samples, we identified differentially expressed proteins and altered biological processes in AEG tumor. Our analysis presented a comprehensive view of proteomic alterations in AEG tumors, and further investigation on their functions and molecular mechanisms in AEG may provide promising drug targets for this disease. The normal samples were also used to identify subtype-specific alterations. In our study, all NAT samples were collected from regions within ~2 cm around the corresponding AEG tumor sites. Paired tumor-NAT samples were derived from the same patients. To reduce the effect of inter-patient heterogeneity and identify subtype-specific tumor differences, we separately compared tumor with NAT samples in each AEG subtype.

In the hallmark gene set analysis, the "pancreas beta cell" gene set showed a significant decrease in AEG tumor samples, especially in the S-III subtype. A large number of adult stem or progenitor cells residue in the epithelium of gastrointestinal organs, which is a source of renewable insulin[+] cells[45,46]. The pancreas and gastrointestinal organs are developed from adjacent embryonic domains[47]. Moreover, native antral endocrine cells and pancreatic β cells share high molecular similarity, and Ariyachet et al. demonstrated that antral stomach cells could be reprogrammed into pancreatic β cells in vivo[48]. Therefore, the changes of "pancreas beta cell" gene set observed in our study might reflect changes in the epithelium. Further investigations are needed to examine our conjecture.

We examined the expression changes in immune checkpoint genes to screen potential immunotherapy targets of different AEG subtypes, which were not necessarily associated with prognosis. We observed that some of the markers may be related to the prognosis, indicating that patients of the S-III subtype may have a better response rate and treatment effect to tumor immunotherapy. Specifically, the expression of CD27 in the S-III subtype was significantly higher than that in the other types, while the expression of VTCN1 in the S-III subtype was significantly lower than that in the other types. CD27, which belongs to the tumor necrosis factor receptors, is a co-stimulatory immune checkpoint. CD27 has been demonstrated to participate in the regulation of generating and maintaining T cell immunity. Evidences have shown that CD27 was able to promote T cell function or dysfunction by regulating the production of IL-2[49,50]. VTCN1, also known as B7-H4, is an immune checkpoint molecule that negatively regulates immune responses and is known to be over-expressed in many human cancers[51]. VTCN1 negatively regulates T cell immune response and promotes immune escape by inhibiting the proliferation, cytokine secretion, and cell cycle of T cells[52]. However, further studies are needed to confirm the specific role of these markers in the immune microenvironment of AEG.

Protein kinases have been developed as operable drug targets in the treatment of cancer[53,54]. We identified hundreds of differentially phosphorylated sites in each AEG subtype, which could be utilized as possible subtype-specific drug targets. Kinase enrichment and kinase-phosphosubstrate relations were also evaluated in all AEG subtypes. Our analysis revealed shared and subtype-specific kinase enrichment and kinase-phosphosubstrate regulatory networks. These results suggest that drugs targeting different kinases might be effective in distinct AEG subtypes (for example, casein kinase II subunit alpha (CSNK2A1) could be a target in the S-II subtype). We hope that these target candidates could be experimentally and clinically explored to benefit patients with AEG tumors in the near future.

In conclusion, the multidimensional analysis in this study represents an advancement in the understanding of the molecular alterations and possible oncological mechanisms of AEG tumors. Although some of our findings need further biological and clinical validation, as the proteomics-based multi-omics characterization of AEG, these data and observations open prospective paths for biological interrogation and therapeutic exploration. Our study may also serve as a valuable resource for future drug discovery and precision clinical practice for patients with AEG tumors.

## Methods

### Collections and preparation of clinical specimen

This study included samples derived from 103 patients from the Cancer Hospital of the University of Chinese Academy of Sciences (Zhejiang Cancer Hospital) from April 2009 to April 2018. The Research Ethics Committees of Zhejiang Cancer Hospital approved the study (No. IRB-2021-288) and all patients provided written informed consent. The informed consent form clearly informs the patients that all clinical information such as age, sex, and TNM staging will be used for academic research and publication. These patients were all newly diagnosed patients with AEG who underwent surgical resection and had received no prior treatment for this disease, including chemotherapy, radiotherapy, targeted therapy, or biological therapy. Patients who were found to have two or more malignancies were excluded.

Patients in this cohort ranged from 40 to 87 years old; the cohort included 81 males and 22 females, 4 cases in stage I, 24 cases in stage II, 69 cases in stage III, and 6 cases in stage IV. We included 27 Siewert type I, 31 Siewert type II, and 45 Siewert type III AEG patients. More detailed clinical information of individual patients, including age, sex, smoking, and drinking status, date of surgery, Lauren type, Borrmann classification, grade of differentiation, tumor size, tumor-node-metastasis (TNM) staging, and survival status and time, are listed in the Supplementary Data 1. Pathological staging was based on the eighth edition of the American Joint Committee on Cancer's Staging

System. Tumor tissues and paired NATs were collected from the same patients at surgical resection. Of note, NAT samples were collected from regions within ~2 cm around the corresponding tumor sites. The sample size was approximately 0.5 × 0.5 cm, and four to five tumor specimens and NATs were collected for most cases. For genomic, proteomic, and phosphoproteomic analyses, tissue specimens endured cold ischemia for less than 30 min prior to freezing in −80 °C refrigerators. For transcriptomic analysis, tissue specimens were soaked in RNA protective solution at 4 °C overnight, and then frozen in −80 °C refrigerators. Histologic sections obtained from the top and bottom portions of each specimen were reviewed by a senior board-certified pathologist to confirm the tissues as tumors or NATs. The top and bottom sections had to contain an average of 60% tumor cell nuclei with less than 20% necrosis to be deemed acceptable for this study.

### Protein extraction and tryptic digestion
A total of 103 AEG tumor tissues and paired NATs were analyzed by proteomics and phosphoproteomics profiling. The samples were taken out from the −80 °C freezers and total proteins were extracted from each sample. In particular, approximately 20–60 mg of tissue sample was placed into a mortar that was pre-cooled with liquid nitrogen and fully ground to a powder under liquid nitrogen. Four volumes of lysis buffer (1% Triton X-100, 1% protease inhibitor, 1% phosphatase inhibitor) were added to the sample powder of each group for ultrasonic lysis. The debris was removed by centrifugation at 12,000 × g at 4 °C for 10 min. Finally, the supernatant was collected and transferred to a new centrifuge tube and the protein concentration was determined using the BCA protein assay (BCA Protein Assay Kit, Pierce). For digestion, the same amount of protein was extracted from each sample, and the volume of each group was adjusted with lysate. Then, 20% trichloroacetic acid was added slowly and precipitated at 4 °C for 2 h. The samples were centrifuged at 4500 × g for 5 min, the supernatant was discarded, and the precipitate was washed with pre-cooled acetone three times. After drying the protein pellets, triethylammonium bicarbonate buffer was added at a concentration of 200 mM, and the pellet was ultrasonically dispersed. Then, trypsin was added at a ratio of 1:50 (protease:protein; m/m) to hydrolyze the proteins at 37 °C overnight. Dithiothreitol (DTT, 5 mM) was added as the reducing agent at 56 °C for 30 min. Finally, iodoacetamide (IAA, 11 mM) was added and incubated at room temperature in the dark for 15 min.

### Phosphorylation modification enrichment
The peptides were dissolved in an enrichment buffer solution (50% acetonitrile/0.5% acetic acid). The supernatant was transferred to the pre-washed immobilized metal affinity capture (IMAC) material, placed on a rotating shaker, and incubated by gentle shaking. The IMAC microspheres with enriched phosphopeptides were collected by centrifugation, and the supernatant was removed. To remove non-specifically adsorbed peptides, the IMAC microspheres were sequentially washed with 50% acetonitrile/6% trifluoroacetic acid and 30% acetonitrile/0.1% trifluoroacetic acid. To elute the enriched phosphopeptides from the IMAC microspheres, an elution buffer containing 10% NH$_4$OH was added, and the enriched phosphopeptides were eluted with vibration. The supernatant containing phosphopeptides was collected and lyophilized for LC-MS/MS analysis.

### Liquid chromatography-mass spectrometry (LC-MS) analysis
The tryptic peptides were dissolved in solvent A (0.1% formic acid, 2% acetonitrile in water) and directly loaded onto a homemade reversed-phase analytical column (25-cm length, 100 μm i.d.). Liquid gradient settings for proteomic analysis: Peptides were separated with a gradient from 6% to 24% solvent B (0.1% formic acid in acetonitrile) over 70 min, 24% to 35% in 14 min, further climbing to 80% in 3 min, and

then holding at 80% for the last 3 min, all at a constant flow rate of 450 nL/min on a NanoElute UHPLC system (Bruker Daltonics). Liquid gradient settings for phosphoproteomic analysis: Peptides were separated with a gradient from 2% to 22% solvent B (0.1% formic acid in acetonitrile) over 50 min, 22% to 35% over 2 min, further climbing to 80% over 4 min, and then holding at 80% for the last 4 min, all at a constant flow rate of 450 nL/min on a nanoElute UHPLC system (Bruker Daltonics). Then, the peptides were subjected to a capillary source followed by timsTOF Pro (Bruker Daltonics) mass spectrometry. The electrospray voltage applied was 1.7 kV. Precursors and fragments were analyzed at the time-of-flight (TOF) detector, with an MS/MS scan range from 100 to 1700 m/z. The timsTOF Pro was operated in parallel accumulation serial fragmentation (PASEF) mode. Precursors with charge states of 0–5 were selected for fragmentation, and 10 PASEF-MS/MS scans were acquired per cycle. The dynamic exclusion was set to 30 s/24 s (proteomic analysis/phosphoproteomic analysis).

### Protein database searching
The resulting tandem mass spectrometry data were processed using the MaxQuant search engine (v.1.6.6.0)[55]. Tandem mass spectra were searched against the human UniProt database[56] (20,366 entries, downloaded on May 9, 2020) concatenated with a reverse decoy database. Trypsin/P was specified as a cleavage enzyme allowing up to 2 missing cleavages. The mass tolerance for precursor ions was set as 20 ppm in the first search and 20 ppm in the main search, and the mass tolerance for fragment ions was set as 20 ppm. Carbamidomethyl on Cys was specified as a fixed modification, and acetylation on the protein N-terminal, oxidation on Met, and phosphorylation on Ser, Thr, and Tyr were specified as variable modifications. The quantitative method was set as label free quantitative (LFQ), and the FDR threshold for protein identification and peptide-spectrum match (PSM) identification was set as 1%. The protein group intensities are provided in Supplementary Data 13.

### Normalization of proteomic and phosphoproteomic data
The iBAQ intensities for proteomics and phosphoproteomics data of 206 samples (103 paired tumors and NATs) were extracted from the MaxQuant result files. A 10,148 × 206 matrix was generated to represent the expression of particular proteins across samples, and a 37,773 × 206 expression matrix was obtained for particular phosphorylation sites. The proteomics and phosphoproteomics data were normalized following a previous study[34]. More specifically, expression matrixes were then subjected to quantile normalization by using the normalized quantile functions implemented in the limma R package (version 3.46.0, R version 4.0.2)[57]. Next, log2-transformation of the normalized iBAQ intensities was calculated for the following quantitative analyses. In addition, all missing values were imputed with the minimum values across individual expression matrixes. The limma package was also adopted to compute the difference in protein and phosphorylation abundances between tumor and paired NAT samples. Specifically, the difference was statistically evaluated by employing a simple linear model and moderated t-statistics by the empirical Bayes shrinkage method.

### Whole-exome sequencing
WES was performed for paired tumor tissues and NATs of 94 AEG cases. Genomic DNA was isolated from tumor tissues and NATs using a DNeasy tissue kit (Qiagen, Hilden, Germany). The concentrations of genomic DNA samples were determined by using the Qubit dsDNA BR Assay (Thermo Fisher Scientific). The DNA integrity was determined by 1% agarose gel electrophoresis. Genomic DNA samples of 1-3 μg were sheared by a Bioruptor® Pico Sonication System (Diagenode SA, Belgium), and an Agilent 2100 Bioanalyzer (Agilent Technologies) was used to assess DNA fragment sizes of approximately 250 bp. These whole-genomic libraries were subsequently prepared by the

SureSelectXT Target Enrichment System for Illumina Paired-End Multiplexed Sequencing Library kit (Agilent Technologies). The whole-exome sequence was captured by SureSelectXT Human All Exon V6 (Agilent Technologies) and quantified by Qubit, Agilent 2100 Bioanalyzer, and qPCR (KAPA Library Quantification Kit KR0405). The final libraries were sequenced for paired-end 150 bp using the Illumina NovaSeq 6000 Sequencing System (Illumina Inc., San Diego, CA, USA) at LC-Bio Technology Co., Ltd. Adapters and low-quality reads ($q$ quality score < 20) were removed from raw WES reads by using fastp software (version 0.21.0)[58]. Then, BWA software (version 0.7.17)[59] was utilized to align filtered reads to the human reference genome (GRCh38). Alignments were subjected to Picard tools (http://broadinstitute.github.io/picard/) to identify and mark duplicate reads. Next, local realignment was performed to correct potential alignment errors around indels. Prior to variant calling, base quality score recalibration was performed to reduce systematic biases. Then, somatic SNVs and InDels were jointly called by Mutect2 (version 4.1.9.0)[60] and Strelka2 (version 2.9.10)[61]. Only variants that passed both quality filtering steps were used in the follow-up analysis. The Variant Effect Predictor (VEP) tool[62] was utilized to fetch biological information of the variant set. Called mutations with annotation information are supplied in Supplementary Data 2.

## mRNA sequencing

mRNA sequencing (RNA-seq) was performed in paired tumor tissues and NATs of 83 AEG cases. Total RNA was isolated from the tumor tissues and NATs in RNA protective solution using TRIzol reagent (Invitrogen, Carlsbad, CA, USA) following the manufacturer's procedure. The RNA amount and purity of each sample were quantified by using a NanoDrop ND-1000 (NanoDrop, Wilmington, DE, USA). The RNA integrity was assessed by an Agilent 2100 with RIN > 7.0. For mRNA sequencing, the library was prepared on 1 μg of DNase I-treated total RNA using a TruSeq kit (Illumina), and 150-bp paired-end sequencing was performed on an Illumina HiSeq X Ten machine at LC-Bio Technology Co., Ltd. (Hangzhou, China) following the vendor's recommended protocol. Raw sequencing RNA reads were first trimmed to remove low-quality bases and reads by using Trimmomatic software (version 0.39)[63] with default parameters. The filtered reads were then aligned to the human reference genome (GRCh38) by using the splice-aware aligner HISAT2 (version 2.2.1)[64]. Alignment results were subjected to gene quantification with gene annotation from GENCODE (version 35)[65] by adopting StringTie software (version 2.14)[66]. Gene expression levels were normalized in the unit of transcripts per million mapped reads (TPM). Genes with expression levels higher than 0.1 TPM in at least one sample remained for downstream analysis. Raw gene counts are provided in Supplementary Data 14.

## Hallmark gene set analysis

The hallmark gene sets were retrieved from the Molecular Signatures Database (MSigDB)[67]. These fifty gene sets were refined from a wide range of biological processes by reducing both variation and redundancy. The integrated abundance of proteins in these hallmarks was then calculated in each sample by utilizing the GSVA R package (version 1.38.2)[68]. A normalized protein expression matrix was used in the calculation.

## Proteomic subtype identification in AEG tumor samples

The non-negative matrix factorization (NMF) algorithm, which is a popular approach to effectively distinguish groups with different molecular features[15,34,69], was employed to identify AEG subtypes from the protein expression profiles of 103 AEG tumor samples. In particular, the consensus cluster method implemented in the NMF R package (version 0.23.0)[70] was utilized to identify the distinct proteomics patterns among individual samples. First, the proteomics profile was filtered before NMF analysis to remove proteins that were

detected in less than 25% of the samples, leaving 9783 proteins. Then, the variation coefficient of each protein across all samples was calculated, and the top 25% of most variable proteins (2445) were used for unsupervised consensus clustering. Next, the NMF algorithm was performed to estimate the optimal rank in a given range from 2 to 5 using 200 interactions. A rank of 3 was selected to run the NMF clustering in 200 interactions. Missing values were imputed with the minimum value in our proteomic dataset.

## Identification of signature proteins in each subtype

To identify the specific molecular alterations in our proteomic subtypes, we compared the protein abundances between tumor samples in individual subtypes with those in tumor and NAT samples of the other subtypes. The statistical significance was estimated by the empirical Bayes shrinkage method implemented in the limma package as described above. In each subtype, a protein that showed remarkably higher abundances than all NAT samples and tumor samples in the other subtypes was considered a signature protein.

## TCGA gene expression analysis

The gene expression profiles of 18 different cancer types in the TCGA cohort with paired tumor and normal adjacent samples were retrieved from the Genomic Data Commons data portal[71] (GDC). In each cancer type, the normalized expression matrix (in TPM unit) was adopted to perform differential expression analysis by using paired Student's $t$ test (as implemented in the R software). Genes with |fold change| ≥ 1.5 and FDR < 0.05 were regarded as statistically significant.

## Survival analysis

The overall survival time was compared between different groups by using the log-rank test implemented in the survival package (version 3.2.3, https://CRAN.R-project.org/package=survival). The survival curves were generated by using the Kaplan–Meier method in the survminer R package (version 0.4.9, https://CRAN.R-project.org/package=survminer). Except for the analysis of subtypes, tumor patients were divided into high- and low-abundance groups by using the median abundances of individual proteins, phosphorylation sites or genes. Hazard ratios with 95% confidence intervals were calculated from the Cox proportional hazards regression analysis. Clinical variables, including age, sex, smoking history, alcohol history, Siewert type, and tumor stage, were used in the Cox regression multivariate analysis.

## Protein-protein interaction network analysis

The human protein–protein interactions (PPIs) were obtained from the STRING database (v11.5)[72]. Differentially expressed proteins (DEPs) were mapped to PPIs to generate the DEP PPI network in AEG. Single nodes were removed from the network. We obtained a PPI network of 3,923 nodes and 79,088 edges. The Cytoscape (version 3.9.0) software[73] was used to visualize the network. The Cytoscape plugin cytoHubba[74] was utilized to calculate the degree, closeness, and betweenness of all nodes in the PPI network.

## Tissue microarray (TMA) construction and immunohistochemistry analysis

A total of 251 formalin-fixed, paraffin-embedded AEG tissues and corresponding NATs from Jan 1, 2009 to Dec 31, 2017 were collected in Zhejiang Cancer Hospital. Two pathologists independently selected the most representative tumors and paired NATs, and TMAs were produced as previously described[75]. Immunohistochemical staining of serial TMAs was carried out as previously described[75]. After treating with 3% $H_2O_2$/methyl alcohol solution for 10 min at room temperature, 5% normal goat serum buffer was used to block the tissue at 37 °C for 30 min. Slides were then incubated with primary antibodies at 4 °C overnight. After washing, the slides were incubated with biotin labeled goat anti-rabbit IgG and HRP-conjugated streptavidin at 37 °C for 1 h. Immunoreaction

was visualized by diaminobenzidine (DAB) (Cat#ZLI-9065, ZSGB-BIO Corp., Shanghai, China). After DAB staining, all tissues were counterstained with hematoxylin (Cat#ZLI-9609 ZSGB-BIO Corp., Shanghai, China) dehydrated and then blocked. The FBXO44 (1:300) antibody was purchased from Proteintech (Chicago, USA). Two experienced pathologists independently evaluated the slides. Brown-stained cells were considered positive. The expression of FBXO44 was assessed using the *H*-score system. The formula for the *H*-score was as follows:

$$H_{score} = \sum(IS \times AP) \qquad (1)$$

where IS represents the staining intensity and AP represents the percentage of positively stained tumor cells. The *H*-score ranged between 0 and 12. An IS between 0 and 3 was assigned for the intensity of tumor cell staining (0 for no staining; 1 for weak staining; 2 for intermediate staining; 3 for strong staining). AP depended on the percentage of positively stained cells as follows: 0 (0%), 1 (1–25%), 2 (26–50%), 3 (51–75%), and 4 (76–100%). The score was assigned using the estimated proportion of positively stained tumor cells. A score ≥6 was considered positive, and <6 was considered negative.

### Cell lines and cell culture
Human AEG cell lines, including OE19 (Cat#CBP60495, OE19 was established in 1993 from a 72-year-old male patient with gastric cardia adenocarcinoma[76]) and SK-GT-4 (Cat#CBP60462, SK-GT-4 was established in 1989 from the primary tumor of an 89-year-old Caucasian male with an adenocarcinoma of the distal esphagus[77,78]), were obtained from Cobioer Biosciences Co., Ltd. (Nanjing, China). OE19 and SK-GT-4 cells were cultured in RPMI 1640 medium (Kino Biological and Pharmaceutical Technology Co., Ltd, Hangzhou, China) containing 10% fetal bovine serum (FBS, Gibco, Grand Island, USA) and 1% penicillin/streptomycin (Kino Co., Ltd., Hangzhou, China) at 37 °C under 5% $CO_2$ in a cell culture incubator. These two cell lines were identified by Short Tandem Repeat, and bacterial and fungi contamination test were negative.

### Colony formation assays
FBXO44 knockdown (shFBXO44) and corresponding negative control (shCtrl) cells and FBXO44 overexpression (FBXO44) and corresponding vector cells were seeded in 6-well plates (500 cells/well) and cultured for 14 days with fresh medium. Thereafter, the cells were subjected to fixation and crystal violet (Solarbio, China) staining. Visible colonies (with >50 cells) were counted to determine the clonogenic potential of these cells.

### Transwell invasion assays
For invasion assays, the upper surface of the membrane was covered by a layer of Matrigel (BD Biosciences, USA). Then, approximately $5 \times 10^4$ OE19 and SK-GT-4 (Vector, FBXO44, shCtrl, and shFBXO44) cells were suspended in 200 μL serum-free medium and inoculated in the upper compartment of the transwell chamber (Corning, USA). Furthermore, 500 μL of complete medium containing 20% FBS was added to the lower chamber. After incubation for 48 h, the cells on the upper surface of the cell membrane were removed with cotton swabs, and the remaining cells were washed with PBS, stained with crystal violet (Solarbio, China), and photographed and analyzed under a microscope at 200× magnification (ix71, Olympus, Japan).

### Wound healing assays
For wound healing assays, approximately $2 \times 10^6$ OE19 and SK-GT-4 (Vector, FBXO44, shCtrl and shFBXO44) cells were seeded onto 6-well plates. Then, three fields of vision were randomly selected for each group and photos were taken at 200× magnification under an optical microscope (ix71, Olympus, Japan) at 0 h and 12 or 24 h after wound induction.

### Mutation signature analysis
To characterize the patterns of nucleotide substitutions, the trinucleotideMatrix function implemented in the maftools R package (version 2.6.05)[79] was used to extract the matrix of nucleotide substitutions in each AEG proteomic subtype. The nucleotide substitution matrix was then decomposed to generate mutation signatures by classifying the immediate bases surrounding mutated bases into 96 substitution classes. Each identified mutation signature was compared to the COSMIC SBS signatures[80] by calculating the cosine similarity.

### Identification of somatic interactions
Some genes were mutually or concomitantly mutated in individual samples. The somaticInteractions function implemented in the maftools R package was employed to detect the mutually exclusive or co-occurring gene pairs. In particular, the pair-wise Fisher's exact test was used to identify significant gene pairs with mutual or co-occurring mutations.

### Bioluminescence imaging
In vivo bioluminescence imaging was carried out by using a cooled CCD camera system (IVIS Imaging System, PerkinElmer, CA, USA) to observe tumor growth. Briefly, normal saline containing 15 mg/mL D-luciferin (Art.No.40901ES03, Yeasen Corp., Shanghai, China) was intraperitoneally injected into mice at 150 mg/kg body weight. These mice were placed in the light-tight chamber of the CCD camera system accompanying 2% isoflurane anesthesia. For luminescent image acquisition, an integration time of 1 to 60 sand binning factors of 4 was used. Signal intensity was measured according to the flux of all detected photon counts from the region tumor area using the Living Image software package (Xenogen Corp., Alameda, CA, USA).

### Hematoxylin–eosin staining and immunohistochemistry
Paraformaldehyde-fixation, ethanol dehydration, transparency with xylene, and paraffin-embedding were carried out for all tissues. A hematoxylin-eosin (H&E) staining kit (Art. ZLI-9609 ZSGB-BIO Corp., Shanghai, China) was used to stain the tissue slices. The histological changes in the tumor tissues were observed with a microscope at 200× magnification. For immunohistochemistry staining, 4-μm tissue slides were treated with 1 mM EDTA buffer (pH = 9.0) for antigen retrieval. The samples were incubated with the anti-FBXO44 antibody (Cat. No. 10626-1-AP) from Proteintech (Chicago, IL, USA). They were then incubated with biotin-labeled goat-rabbit IgG and horseradish peroxidase-conjugated streptavidin for 1 h. They were then photographed with an inverted microscope at 200× magnification.

### Estimation of infiltrating cell abundance
The abundances of different infiltrating cell types were calculated by using xCell (https://xcell.ucsf.edu/)[81] based on transcriptomic data. In the 64 cell types curated by the xCell method, we removed those that were not relevant in AEG tissues, such as hepatocytes, keratinocytes, and osteoblast. We then removed those cell types that had a xCell score of 0 across all samples. A total of 41 cell types were involved in subsequent analysis, including 10 stromal cell types (adipocytes, astrocytes, fibroblasts, preadipocytes, pericytes, lymphatic [ly] endothelial cells, microvascular [mv] endothelial cells, smooth muscle, endothelial cells, and myocytes), 9 lymphoid cell types (central memory CD4+ T cells [CD4+ Tcm], effector memory CD4+ T cells [CD4+ Tem], CD4+ memory T cells, regulatory T cells [Treg], gamma delta T cells [Tgd], T helper type 1 cells [Th1], T helper type 2 cells [Th2], naïve B cells, and plasma cells), 11 myeloid cell types (basophils, dendritic cells [DC], macrophages, M1 macrophages, M2 macrophages, mast cells, activated dendritic cells [aDC], conventional dendritic cells [cDC], immature dendritic cells [iDC], plasmacytoid dendritic cells [pDC], and monocytes), 7 stem cell types (platelets, common

lymphoid progenitor [CLP], common myeloid progenitor [CMP], granulocyte-macrophage progenitor [GMP], hematopoietic stem cells [HSC], megakaryocyte-erythroid progenitor [MEP], and mega-karyocytes) and 4 other cell types (epithelial cells, mesangial cells, neurons, and sebocytes). Briefly, xCell inferred cell types based on gene signatures that were extracted from 1822 pure human cell type transcriptomes by a curve fitting approach. The xCell scores (relative abundances) were calculated in each sample (Supplementary Data 15) and were compared between different groups by using Student's t test.

## Kinase-substrate enrichment analysis

The kinase-substrate enrichment analysis (KSEA) was conducted by using the KSEAapp R package (version 0.99.0)[82] with known kinase-substrate pairs derived from PhosphoSitePlus® (PSP)[83] and NetworKIN 3.0[84]. Kinase-substrate pairs with a score of more than 1 were used in the enrichment analysis. In each subtype, Spearman correlation coefficients between different kinase proteins and paired phosphosubstrates were calculated to build the kinase-phosphosubstrate network.

## Establishment of stable FBXO44 overexpression and knock-down cell lines

Human FBXO44-shRNA and FBXO44-overexpression lentiviral vectors were constructed, validated, and supplied by Shanghai Genechem Chemical Technology Co., Ltd. (Genechem, Shanghai, China). For FBXO44 silencing, among the three designed FBXO44 siRNA target sequences tested, the target sequence with the best silencing efficiency was: CCAGCAGAAGAGCGATGCCAA. After annealing, oligonu-cleotides were cloned into the AgeI/EcoRI sites of Luc-tagged GV344 lentivirus vectors (Genechem, Shanghai, China). After identification of the correct sequence and lentivirus packaging, OE19 and SK-GT-4 cells were infected at a multiplicity of infection (MOI) of 10 for 24 h. For FBXO44 overexpression, the cDNA of FBXO44 was sub-cloned using Taq DNA polymerase (SinoBio Biltech Co. Ltd., Shanghai, China) and inserted into the BamHI/AgeI sites of Luc-tagged GV260 lentivirus vectors (Genechem, Shanghai, China). Forward primer: AGGTCGACTC TAGAGGATCCCGCCACCATGGCTGTGGGGAACATCAAC, reverse pri-mer: CTTCCATGGTGGCGACCGGTACGGGCAGCGGGGGCCCGATGGT GATG. After identification of the correct sequence and lentivirus packaging, OE19 and SK-GT-4 cells were infected at an MOI of 10 for 24 h.

## Xenograft and orthotopic mouse models of AEG

In accordance with the protocols for experimentation on animals (National Institutes of Health Publication No. 85-23, revised 1996), the animal experiments conducted were approved by the Institutional Animal Care and Use Committee of Zhejiang Chinese Medical University (The Ethics Committee stipulates that the xenograft and orthotopic tumor volume of mice should not exceed 2000 mm³, and our experiments meet the ethical requirements.). The nude mice (male, 4 weeks old) were raised in the laboratory for a week before the experiment. Mice were fed in the Specific Pathogen Free (SPF) barrier center at the animal experimental center of Zhejiang Chinese Medical University, under standard conditions of temperature ($25 \pm 2\,°C$) and humility ($50 \pm 5\%$) in a 12 h light/12 h dark cycle with normal drink and food. A total of $5 \times 10^6$ OE19 (Vector, FBXO44, shCtrl and shFBXO44) cells were injected subcutaneously to establish subcutaneous xeno-graft tumor models in nude mice, 6 mice in each group. The body weight, living status, and tumor size of nude mice were recorded. After 5 weeks of observation, the mice were put into the carbon dioxide anesthesia box, the carbon dioxide valve was then open, and when the animal gradually loses consciousness, the carbon dioxide concentra-tion was increased to 100% for 2 min, and then followed by cervical dislocation. The nude mice were sacrificed, and tumors were frozen at −80 °C until use. For the orthotopic mouse model, subcutaneous tumors grown in nude mice were harvested and resected under aseptic conditions. Necrotic tissues were removed, and viable tissues were cut with scissors and minced into 1–2 mm³ fragments. Before implanta-tion, the mice were anesthetized by an intraperitoneal injection of 0.3% pentobarbital sodium (25 μl/g body weight) (Sigma, Steinheim, Germany). A 10–15 mm midline incision was made in the upper abdomen, and the stomach was carefully exposed. Part of the serosal membrane, approximately 2 mm in diameter, in the middle of the greater curva-ture of the stomach was mechanically injured with a scalpel. A tumor piece was then fixed onto the injured site of the serosal surface with medical OB glue. The stomach was then returned to the peritoneal cavity, and the abdominal wall and the skin were closed with sutures. The remaining steps were the same as those in the xenograft mouse model experiments.

## Statistical analysis

Statistical analysis and data visualization in this study were performed by using R software (R Foundation for Statistical Computing, Vienna, Austria; http://www.r-project.org). Unless otherwise specified, all tests were two-tailed, and a P value or FDR < 0.05 was considered to indicate statistical significance.

## Reporting summary

Further information on research design is available in the Nature Portfolio Reporting Summary linked to this article.

## Data availability

The proteomics and phosphoproteomics data were deposited in the ProteomeXchange database[85] with dataset identifiers PXD030667 and PXD030725, respectively. The WES and RNA-seq data were deposited in the Sequence Read Archive (SRA) database under the accession number PRJNA788008. The gene expression profiles, mutation, and CNV datasets of TCGA cohorts were retrieved from the Genomic Data Commons (GDC) data portal (https://portal.gdc.cancer.gov/). Soft-ware and publicly available resources used in this study were described in the Methods section. Other results generated in this study can be found in the Supplementary data. Source data are provided with this paper.

## Code availability

Scripts and code that were used for data analysis and visualization were deposited in https://github.com/lishenglilab/AEG_Proteomics.

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

## Acknowledgements

This study was supported by The National Key Research and Development Program of China (2021YFA0910100 to X.C.), Zhejiang Provincial Research Center for Upper Gastrointestinal Tract Cancer (JBZX-202006 to X.C.), Medical Science and Technology Project of Zhejiang Province (WKJ-ZJ-2202 to J.Q., WKJ-ZJ-2104 to X.C.), National Natural Science Foundation of China (82074245 to X.C., 81973634 to Z.X., 81903842 to J.Q.), Natural Science Foundation of Zhejiang Province (LR21H280001 to J.Q.), Science and Technology Projects of Zhejiang Province (2019C03049 to X.C.), and Program of Zhejiang Provincial TCM Sci-tech Plan (2018ZY006 to X.C., 2020ZZ005 to J.Q.). We thank the staff of the follow-up room of Zhejiang Cancer Hospital for their support to this work.

## Author contributions

X.C. and J.Q. designed and supervised the project; S.L. designed and performed omics data analysis and visualization; J.Q. and X.C. designed and supervised the experiments; L.Y., Z.X., and J.X. collected clinical specimens. L.Y., G.C., X.G., and G.P. conducted in vitro experiments; L.Y., C.H., and H.X. conducted in vivo experiments. L.Y. performed experimental data analysis. J.D., Y.D., L.Y., M.N., R.J., X.Z., H.L., and S.Z. interpreted the results and commented on the paper; S.L., J.Q., and X.C. wrote the paper from comments of other authors. All listed authors discussed the results and reviewed the paper.

## Competing interests

The authors declare no competing interests.
