## [Peer Review File · Nature Communications]

Integrative Proteomic Characterization of Adenocarcinoma of Esophagogastric JunctionReviewers' Comments:

Reviewer #1:

Remarks to the Author:

Li et al. Integrative Characterization of Adenocarcinoma of Esophagogastric Junction

Li and colleagues perform a multi-omics analysis of junctional cancer from more than 100 cases that includes whole exome, RNAseq, and mass spectrometry of proteins and the phosphorylation sites. They show that the cases can be roughly divided into three subclasses (S-1, SII, and SIII) based on the aggregate molecular parameters they identify, and that there is clinical significance for each with respect to survival times, potential for certain therapeutic regimens, and perhaps immunotherapy. This work adds important dimensions to the earlier studies out of the Broad and Sanger that were largely focused on genomics and expression profiles of EAC, and suggests that much will be learned from expanding and correlating the multiple datasets.

Comments:

1. In the introduction there needs to be a clearer presentation of "gastric cancer" as distinct from the junctional cancer. Earlier genomic studies of upper GI cancer showed what many expected that so-called "intestinal gastric cancer", the H. pylori-associated adenocarcinoma that initiates with GIM, is in fact closely related to EAC or junctional cancer. In contrast, diffuse gastric cancer is very different. As intestinal gastric cancer is so prevalent, especially in Asia, it needs to be highlighted in the introduction.
2. From an informatics standpoint, one is concerned for the "pairwise" analysis used here though these concerns may be unfounded. Regardless, the source of "normal" tissue in the figures is said to be the gastric mucosa. However, the gastric mucosa is topologically (regionally) diverse suggesting that the reference point in these studies could be variable for each case. Has the team considered developing an amalgam 'normal' from all normals to be generically compared with each tumor sample to limit such variability?
3. With the intense interest in immunotherapy for these difficult tumors, it was intriguing to see that the SIII subclass, with the longest survival time, was also the class that, by cell type deconvolution analysis, had the most lymphoid and myeloid cells and the fewest fibroblasts. Given the thoughts that fibroblasts, and particularly myofibroblasts, may limit access by immune cells, was this correlation evident in the H&E analysis across the subtypes.ss
4. The focus on FOXO44 is interesting, and yet it seems to be a property of S-II class which apparently has a more complex survival profile than S-I and S-III. May have missed this, but given the link proposed with metastasis, is S-II cases, especially those with high FOXO44, more metastatic?

Reviewer #2:

Remarks to the Author:

Please see attachment.

Li *et al.* present a multi-omic characterization of adenocarcinoma of the esophagogastric junction (AEG). The results presented are impressive and represent a comprehensive overview of these tumors from a variety of omics standpoints. Unlike many multi-omic survey manuscripts, the authors included follow up experiments and mouse model work to help validate their findings. The reviewer believes these results are important, but some work is needed before these findings are suitable for publication. In particular, the results describing FBXO44 seem out of place. Next, the authors need to go beyond simply listing feature and pathway differences between their multi-omic subtypes. Each results section reads like its own long list of differences, seemingly without much thought given to how these differences relate to the other results. This work would be greatly strengthened if the authors defined the key phenotypic features of these multi-omic subtypes and how they may be clinically relevant. Further, too much time is spent on comparisons with normal samples. Not surprisingly the normal samples are very different from tumors, and nearly every comparison made by the authors resulted in many hits. The corresponding results have too many genes and/or pathways to easily interpret. A better use of the normal data may be to use it in a more targeted manner. For example, to help screen for cancer-specific targets that were enriched in a particular multi-omic subtype. Finally, the wording throughout the manuscript needs to be more specific, and the statistical tests utilized along with fold changes and p-values need to be provided. In addition to this, please find the major and minor comments to address below.

Major comments

- Page 6
 - 'Protein Database Searching'
 - **All of the proteomic findings seem to refer to single protein accessions/names. What happened to the protein group assignments from MaxQuant? The reviewer expects at least some of these protein groups to contain more than one protein accession. Protein group abundances as rows and samples as columns should be included in supplemental.**
 - **The reviewer appreciates the data being deposited into various repositories, but as much as possible the data used to generate the figures presented in the manuscript should be included as supplemental materials. This includes raw gene counts, protein group intensities, called mutations, multi-omic subtyping, etc. at the sample level. Presently, the supplemental data only seems to contain summarized results and fold changes across conditions.**
- Page 7
 - 'Whole Exome Sequencing' ... 'mRNA Sequencing'
 - **The description of computational methods for WES and RNA-seq processing are severely deficient. Please provide detailed processing of the raw sequencing data including the alignment, reference genome, QC steps, variant callers, read counting, etc. to allow for reproduction of the presented results. Sample level RNA-seq counts should be included in supplemental.**
- Page 13
 - 'From the RNA-seq data, 23,131 genes were found to be expressed in 166 AEG tumor and NAT samples...'
 - **See the above comment on the RNA-seq processing. Include the expression/missingness thresholds used to filter the data (if any). Were there a large number of genes expressed in only the normal/only the tumor?**
 - '...including proteomics profiling, phosphoproteomics profiling, WES, and RNA-seq (Figure 1A).'
 - **The reviewer had a hard time with the figures. The text in most of the figures is very small, and many, many protein names and/or pathways are listed. Non-key findings should be relegated to supplemental figures or even tables. The authors should take time to consider how their results are displayed and do a better job at distilling the key findings down to fewer, more interpretable figures.**
- Page 14
 - Proteomic Characteristics of AEG Tumors

- **Please see the comment in the opening paragraph about repurposing the normal data and utilizing it to ask specific questions as opposed to showing numerous differences between normal and tumor in each omics type.**
 - Proteomics-Based Subtyping of AEG Tumors
 - **The authors need to do a better job at characterizing these subtypes. Yes, they do the bookkeeping in the results and write the expected things about differentially expressed proteins and pathways, but what is the overarching story? What biology defines these subtypes from one another? Why do the authors think the S1 subtype has the worst survival? Did the immune infiltration later shed any light into this? What is the role of FBXO44, if any, in these subtypes?**
 - '... whereas the "KRAS signaling up" hallmark was significantly down-regulated ($P = 1.1E-3$) in tumor samples (Figure 2D).'
 - **There are not many KRAS mutations and KRAS does not appear to be a major driver in this cancer. Why is this showing up? Is it a red herring?**
 - 'These observations implicate that flutamide might be effective in treating AHR-high AEG patients.'
 - **The authors should do a better job at supporting this claim. Just because this and other proteins are differentially expressed does not mean that they would make good drug targets. Not all of these will be drivers/critical to the survival of the tumor. Is there publicly available drug screening data that could be integrated here? Could a network analysis of proteomics findings using protein-protein interaction data identify bottlenecks/hub proteins that would make better drug targets?**
- Page 15
 - 'Of these, 12 signature proteins were targetable by FDA-approved drugs or candidate drugs currently in clinical trials (Figure 3G).'
 - **Same comment as above. The reviewer would like this idea developed more and have more supporting evidence shown (computational evidence would be acceptable).**
 - **Since the authors bring both up, can they quantify what would be better based on their data: drug targets based on normal to tumor analysis, or drug targets based on proteomic subtypes?**
 - 'For example, the activity of the "G2M checkpoint" hallmark in the S-III subtype was significantly higher than those in the other two subtypes ($P = 1.7E-3$ compared to S-I subtype, $P = 1.2E-4$ compared to S-II subtype) (Figure 3E), while "pancreas beta cells" showed remarkably lower levels in the S-III subtype ($P = 1.7E-2$ compared to S-I subtype, $P = 4.3E-2$ compared to S-II subtype) (Figure 3F).'
 - **The reviewer assumes that the authors mean the large degree of change of expression/abundance of "G2M checkpoint" proteins when they refer to activity. This is confusing because 1) there are activity-based protein profiling (ABPP) proteomic assays that measure activity directly and 2) the authors have phosphoproteomics data. Please reword this section and change the axes of figures that mention activity. Should the figures say fold change instead?**
 - **What kind of pathway enrichment is being performed? The reviewer does not remember "pancreas beta cells" being one of the cancer hallmarks. If the type of pathway enrichment or the resource used for pathway enrichment is changing, then the authors should clarify. All pathway results, including non-significant ones, should be included in supplemental tables.**
 - **Why were these particular pathway findings called out? Were these the most differentially expressed by p-value/FDR? Given the large number of changes, the authors should do a better job at prioritizing what they show and make it clear why they are showing it. Is there an empirical cut point at say the top 5 or 10 pathways before the p-values drop off substantially? That would be one way to prioritize all of these significant findings.**
 - **Do AEG tumors share similarities to pancreatic tissue or beta cells?**

- 'Patients in these three subtypes showed significantly distinct overall survival time ($P = 1.1E-3$)...'
- **What test is this p-value associated with? What is the hazard ratio? There appears to be no survival analysis details in the methods. Has this been adjusted for gender, smoking history, and stage?**
- Page 16
- 'FBXO44 Promotes AEG Tumor Progression and Metastasis '
- **Why was FBXO44 chosen out of all of the possible choices? Given the sheer volume of data, why were there no better choices based on combining and integrating the findings? This protein does not look like it mentioned in the results until this page.**
- **Were these data generated independently of the multi-omic analyses being presented? These findings seem like they were shoehorned between several sections of global omics characterization as an afterthought to give the results some more clinical relevance.**
- **Please convince the reviewer that this was the best target and similar or better results could not have been obtained looking at the highest fold changes and pathways different across normal/tumor/tumor subtypes.**
- **Is the biological function of FBXO44 known? If so what pathway is it a part of? How does it relate to the immune findings further down? Is its expression highly correlated with anything interesting across the omics types? If little is known about this protein, then surely some hints about mechanism could be generated with the abundance of data. Even a simple co-expression analysis could shed some light on upstream/downstream targets.**
- **What are the exact fold changes from normal to tumor and across the proteomic subtypes? These values do not look like logged intensities. Why is relative abundance used instead of log2 intensities? Abundances relative to what? Is it not implied that these are relative abundances since the authors did not do absolute quantification?**
- **See above survival analysis comments. What is test is this p-value associated with? What is the hazard ratio? There appears to be no survival analysis details in the methods. Has this been adjusted for gender, smoking history, and stage?**
- **Is this simply a protein that is correlated with or involved in cell proliferation therefore explaining the survival curve and mouse model results?**
- **How targetable is this protein/pathway given the authors used shRNA as opposed to a drug?**
- 'The up-regulation of FBXO44 protein in tumor samples was further validated in an independent clinical cohort ($P = 1.55E-4$) (Figure 4B). '
- **What cohort? AEG patients? How many samples? How much up-regulation? The overall pattern of blue/brown is easy to see in the figure, but the images are too small to see individual features. Please enlarge these images significantly, or if they do not fit in the main figure please include in supplemental.**
- '...FBXO44 KD remarkably suppressed cell proliferation...'
- **Please quantify the amount of suppression instead of using words like remarkable, and please use more precise wording throughout to quantify results when they are mentioned.**
- Page 19
- 'Phosphoproteomic Characterization of AEG Tumors'
- **How important is the phosphoproteomics data in relation to the multi-omics subtypes? Is it a defining feature? Since the phosphorylation data gives a measurement of signaling events in the tumors, is it wise to have these measurements weighted the same as the other omics types (e.g. RNA-seq) even though the phospho-data may be more relevant for activated signaling pathways**

and drug targets? Do any of these phosphoproteins or kinases relate back to FBXO44?

Minor comments

- Page 6
 - 'Tandem mass spectra were searched against the Homo_sapiens_9606 database (20,366 entries) concatenated with a reverse decoy database. '
 - **Is this from Uniprot or another database? Please provide a citation as well as the date accessed.**
 - 'The *limma* package was also adopted to compute the difference of protein and phosphorylation abundances between tumor and paired NAT samples. Specifically, the difference was statistically evaluated by employing a simple linear model and moderated t-statistics by the empirical Bayes shrinkage method. '
 - **Do the authors think that limma might be too conservative here? The shrinkage method is certainly appropriate for large RNA-seq or microarray datasets, but will meaningful signal be lost applying here to proteomics data?**
 - **Please cite the limma manuscript and provide the version of the package and the version of R used.**
- Page 11
 - 'The xCell scores (relative abundances) were calculated in each sample and were compared between different groups by using Student's t-test. '
 - **In the reviewer's hands, xCell scores tend to not follow a normal distribution. Do the findings from xCell still hold if a non-parametric Wilcoxon rank sum test is used?**
- Page 13
 - 'In particular, proteomics and phosphoproteomics profiling were performed on 206 samples (Figure 1B).'
 - **The circos plot is difficult to read since most of the plot has the same pattern/is not different. This does not seem to convey much meaningful information Please consider a simple table or a different figure to convey the total amount of differentially expressed analytes.**
- Page 13
 - 'In the present AEG cohort, the most frequently mutated cancer-related genes (derived from COSMIC v95)37 were TP53 (62%), MUC16 (31%), FAT4 (22%), LRP1B (18%), ARID1A (16%), and FAT3 (16%) (Figure 1C).'
 - Please add additional tick marks to the TMB barplot at the top of the heatmap to make it easier to read.
 - 'Overall, significantly larger number of proteins ($P = 3.8E-15$), phosphorylation sites ($P = 1.6E-4$), and genes ($P < 2.2E-16$) were detected in AEG tumors than those in NAT samples (Supplemental Figure S2). '
 - **What test was used here? T-test? Fisher's exact test? Please make sure to provide the name of the test along with the p-value, or alternatively make a list of the tests performed for the different data types in the methods.**
- Page 14
 - 'We next investigated the disturbance of proteins in AEG tumors. Differential protein analysis revealed 2,300 up-regulated and 1,667 down-regulated proteins in AEG tumor samples compared to paired NAT samples (Figure 2A and Supplemental Table S3). '
 - **For 2A and all figures, the authors should use HUGO gene symbols instead of Uniprot accessions. Readers will be much more familiar with gene symbols and it will make results easier to interpret. Uniprot provides mapping tables for this purpose**
- Page 18

- 'Tumors in the S-II subtype showed the least number of changed cell types, while the S-III subtype exhibited the most altered cell types, especially the increase of lymphoid and myeloid cells.'
- **Altered from what? Please reword. Is this not inherent representation of immune cells in a given proteomic subtype? Is this referring to "differential expression" of the xCell scores?**
- 'The xCell algorithm was employed to infer the relative cell abundance of 41 different cell types (see Methods).'
- **Why only 41? The reviewer believes xCell can estimate more than this. The table of xCell scores and p-values should be provided as supplemental.**
- Figure 2
 - A
 - **The reviewer had a hard time reading these volcano plots. The shading makes them almost uninterpretable. These do not need the sample frequency since presumably the authors did some type of filtering for missingness before the data was presented.**
 - **Gene symbols should be used instead of Uniprot accessions.**
 - **The reviewer was going to comment that the non-significant findings should be colored black or dark grey with the significant findings colored red/blue, but it now appears like this has already been done. Something needs to be done to help with the interpretability of these figures. Maybe only coloring the top fold-change/FDR hits? A much more stringent cutoff?**
 - C
 -
- Figure 6
 - A, D
 - **The dots/bubbles are very small and hard to see/interpret. Please increase their size to aid in interpretability. There does not need to be so much empty space in between them.**
 - **Do these have -logp and -logfdr abbreviated with the "o" taken out? Please write them out.**
 - **Justify using p-values for one and FDR for the other. Why not be consistent? Do the results still hold with FDR? That would be acceptable as long as the results are clearly stated.**
- Figure 7
 - A
 - **Similar comments above about this volcano plot.**
 - B
 - **If the pathway enrichment is shown in C, then are these nearly identical heatmaps really needed?**
 - C
 - **The text direction is the opposite of other figures. Please be consistent.**
 - **There are too many genes listed.**
 - D-F
 - **What are readers supposed to gain from these? There is too much going on here. Please consider hiding the nodes that are not relevant or highlighting the key nodes.**
- Supplemental Figure S1
 - **Was the phosphoproteomics data normalized the same as the protein expression? The distribution of the boxplots look different.**
- Supplemental Figure S4
 - **Have all of the survival analyses been adjusted for clinical variables mentioned above? What about false discovery? Hazard ratios should always be reported.**
- Supplemental Figure S5
 - C, D

- **These images are too small to see. Please make them larger and include scale bars on any other histology/ICC/IHC images throughout.**

Reviewer #3:

Remarks to the Author:

Considerable effort is appreciated.

There are the following major issues:

AEG subtype (Siewert type I, II, and III) have different biology and cannot be combined as such. Overall samples size is rather small.

According to the Table s1. there are no Siewert type I patients in the cohort studied. '

Again, there are only 4 patients with Siewert type II (gastroesophageal junction). These should be removed. Therefore, what is left in the cohort are Siewert type III and some gastric cancer patients. Essentially, not a study of 3 types of upper GI tumors.

The two cell lines studied (OE3 and Sk-GT-4) are Siewert type I cell lines and not relevant in this study.

All tumors (almost) are of high localized stage and with varied survival. The overall, survival analysis fails to correlate molecular subtypes with phenotypes/histotypes.

The manuscript claims that multiomics analysis has not been done, which is not true. TCGA STAD included 4 times more patients and was much more comprehensive. Similarly, the Samsung paper not quoted. The authors have not acknowledged TCGA subtypes and validated their findings.

Figure 1. Remove AEG I and II (as there are no AEG 1 tumors in this study and there are only 4 AEG II and they should be removed from the analysis as they do not provide useful data).

In the introduction, "surgical resection is most effective" cannot be generalized. It is acknowledged that surgery is essential for cure but multimodality is commonly practiced. Surgery first may be a Chinese approach and should be qualified.

In the introduction, there should be mention of novel studies with IO

The normal tissue is seemingly appropriate for some comparisons but it is expected that once some proteins are differentially expressed in tumor/normal, repetitive analysis of tumor v normal (Figures 1D, 1E, and 1F are not very informative).

Similarly, Figure 2A distracts from what we can learn about tumors. Same for Figures 2D and 2C.

Proteomics did not provide the location of these proteins (cell surface, nuclear, cytoplasmic, or total).

Figure 3 is interesting. 3 types (S-I, S-II, and S-III) are not correlated with phenotypes/histologies. Types S-I and S-II are similar in prognosis. It is not clear what may be promoting better survival in S-III when one reviews Figure 3D (many oncogenes are up-MYC and cell cycle). Angiogenesis is down can make sense. OxPhos down can make sense but need better interpretation from the authors. and correlate with clinical variables.

Figure 4G. why include normals here??? Why normals in different subtypes are different? Were they not obtained from a distant gastric location? If so, are the differences related to cancer? Very confusing.

Figure 5A. again, inclusion of normals does not seem to add much here. Confusing for S-I.

the finding that FBXO44 is associated with poor outcome in multiple cancer patients (their ref 38) is not novel. In ref 38, those authors have produced significant high quality data and the current manuscript provides no novelty. It would appear that it would be difficult to target FBXO44 but it could serve as a marker to use IO. these authors could have considered those studies.

Integration of various platform remains elusive. Need better description and plan. Integration with clinical variables would be more meaningful.

Subtypes I, II, and III were derived by proteomics data and by integrated analysis. The significance remains unclear. Subtypes not integrated with clinical variables.

there is useful information on TME analysis. but again not correlated with clinical phenotypes. Not integrated.

Genomics of subtypes is noted but not integrated to the extent it can be done.

A lot of analyses are descriptive and correlative. Not highly informative.

Discussion has many misstatements and unfocused emphasis.

It is unclear if these data provide a step forward as prior studies were not placed in context.

Point-by-point Response (NCOMMS-22-09290)

Reviewer #1 (Remarks to the Author): Expert in gastric cancer

Li et al. Integrative Characterization of Adenocarcinoma of Esophagogastric Junction
Li and colleagues perform a multi-omics analysis of junctional cancer from more than 100 cases that includes whole exome, RNAseq, and mass spectrometry of proteins and the phosphorylation sites. They show that the cases can be roughly divided into three subclasses (S-1, SII, and SIII) based on the aggregate molecular parameters they identify, and that there is clinical significance for each with respect to survival times, potential for certain therapeutic regimens, and perhaps immunotherapy. This work adds important dimensions to the earlier studies out of the Broad and Sanger that were largely focused on genomics and expression profiles of EAC, and suggests that much will be learned from expanding and correlating the multiple datasets.

Response: Thanks very much for the overall positive comment of our manuscript. We appreciate very much the valuable comments and suggestions raised by the Reviewer. We have carefully revised the manuscript according to these comments, which greatly improved our manuscript. Please see the detailed point-to-point response as follows:

Comments:

Q1: In the introduction there needs to be a clearer presentation of "gastric cancer" as distinct from the junctional cancer. Earlier genomic studies of upper GI cancer showed what many expected that so-called "intestinal gastric cancer", the H. pylori-associated adenocarcinoma that initiates with GIM, is in fact closely related to EAC or junctional cancer. In contrast, diffuse gastric cancer is very different. As intestinal gastric cancer is so prevalent, especially in Asia, it needs to be highlighted in the introduction.

Response: Thanks very much for the professional comment and nice advice. In the revised manuscript, we described the relationship and differences between AEG and gastric cancer from aspects of epidemiology, etiology, pathological features, and the role of *Helicobacter pylori* infection. The detailed description is as follows:

Line 7-19, Page 3: " AEG is obviously different from gastric cancer in epidemiology, etiology, and pathological characteristics. The incidence rate of AEG has increased year by year, while that of gastric antral carcinoma has decreased significantly^{1,2}. According to the Lauren classification, the intestinal type was most common in AEG, and intestinal metaplasia led by gastroesophageal reflux disease (GERD) is the main risk factor for AEG^{3,4}. However, there are more diffuse type cases of gastric antrum carcinoma, and chronic atrophic gastritis is an important precancerous lesion of gastric antrum carcinoma⁴. In addition, *Helicobacter pylori* (*H. pylori*) infection is a recognized carcinogenic factor of gastric antrum cancer. Cytotoxigenic associated gene A (CagA) in *H. pylori* may significantly increase the risk of atrophic gastritis and gastric antrum cancer, but its role in AEG is controversial⁵. Some studies have shown that *H. pylori* infection can prevent GERD, Barrett's esophagus and other reflux diseases, thus reducing the incidence of AEG to a certain extent⁶."

Note: Related references were cited in the revised manuscript.

Q2: From an informatics standpoint, one is concerned for the "pairwise" analysis used here though these concerns may be unfounded. Regardless, the source of "normal" tissue in the figures is said to be the gastric mucosa. However, the gastric mucosa is topologically (regionally) diverse suggesting that the reference point in these studies could be variable for each case. Has the team considered developing an amalgam 'normal' from all normals to be generically compared with each tumor sample to limit such variability?

Response: Thanks very much for the Reviewer's comment. The NAT site marked in the original Figure 1A was misleading. We apologize for the confusion caused by the inaccurate schematic diagram, which should have been avoided. In our study, all NAT samples were collected from regions within ~2 cm around the corresponding AEG tumor sites. We revised the schematic diagram and added text of "~2cm around tumor" in revised Figure 1A (**Figure R1-1**). This has also been described in the revised manuscript (**Line 21-22, Page 5**). Pairwise comparisons of tumor and NAT around tumor sites are common in many multi-omics studies in gastric or colon cancer⁷⁻⁹.

Figure R1-1. Schematic diagram of anatomical sites where AEG and NAT samples were collected.

Q3: With the intense interest in immunotherapy for these difficult tumors, it was intriguing to see that the SIII subclass, with the longest survival time, was also the class that, by cell type deconvolution analysis, had the most lymphoid and myeloid cells and the fewest fibroblasts. Given the thoughts that fibroblasts, and particularly myofibroblasts, may limit access by immune cells, was this correlation evident in the H&E analysis across the subtypes.

Response: Thanks very much for the professional comment. In this revision, we performed H&E analysis across these three subtypes. Compared to those in the S-I and S-II subtype, we observed decrease of fibroblasts and increase of lymphoid and myeloid cells in the S-III subtype (**Figure R1-2, revised Figure 6E**). This has also been described in the revised manuscript (**Line 16-18, Page 23**).

Figure R1-2 (revised Figure 6E). H&E analysis of tumor cells, lymphoid cells, myeloid cells and fibroblasts across three AEG subtypes.

Q4: The focus on FOXO44 is interesting, and yet it seems to be a property of S-II class which apparently has a more complex survival profile than S-I and S-III. May have missed this, but given the link proposed with metastasis, is S-II cases, especially those with high FOXO44, more metastatic?

Response: Thanks very much for the comment and nice advice. In the AEG cohort of 103 patients, 23 patients were classified into S-II subtype. To examine the association between FOXO44 and the metastasis of AEG, we verified FOXO44 in a larger AEG cohort of 251 patients. The expression of FOXO44 was assessed using the H-score system. The formula for the H-score was as follows:

$$\text{H-score} = \sum (\text{IS} \times \text{AP}),$$

where IS represents the staining intensity and AP represents the percentage of positively stained tumor cells. The H-score ranged between 0 and 12. An IS between 0 and 3 was assigned for the intensity of tumor cell staining (0 for no staining; 1 for weak staining; 2 for intermediate staining; 3 for strong staining). AP depended on the percentage of positive-stained cells as follows: 0 (0%), 1 (1-25%), 2 (26-50%), 3 (51-75%), and 4 (76-100%). The score was assigned using the estimated proportion of positively stained tumor cells. A score ≥ 6 is positive and < 6 is negative. Our analysis found that FOXO44 was significantly associated with distant metastasis ($\chi^2 = 6.19$, $P = 0.013$) and advanced TNM stage ($\chi^2 = 8.95$, $P = 0.030$) of AEG tumor (**Figure R1-3**, revised Supplemental **Figure S10**). Furthermore, we also assessed the association between FOXO44 protein level and all other clinicopathological features of AEG patients (**Table R1-1**, revised Supplemental **Table S8**). In addition to distant metastasis and advanced TNM stage, FOXO44 was found to be highly associated with older age ($\chi^2 = 5.507$, $P = 0.019$) and high AFP level ($\chi^2 = 14.489$, $P < 2.00E-16$). These have also been described in the revised manuscript (**Line 17-25, Page 20**).

Figure R1-3 (revised Supplemental Figure S10). Validation of FBXO44 in an independent cohort of 251 AEG patients. (A) The number of samples with different H-score ranges (0, 1-5, and 6-12). **(B)** FBXO44 protein was significantly enriched in AEG patients with M1 tumor stage. **(C)** FBXO44 protein was significantly enriched in AEG patients with advanced tumor stages.

Table R1-1 (revised Supplemental Table S8). The associations between FBXO44 protein level and clinicopathological features of AEG patients.

Variables	FBXO44 expression		#Total	Positive Rate	χ^2	p-value
	#Positive	#Negative				
Age (year)						
≥ 65	43	94	137	31.39%	5.507	0.019*
< 65	21	93	114	18.42%		
Sex						
Female	18	36	54	33.33%	2.224	0.136
Male	46	151	197	23.35%		
Family history (gastric cancer)						
Yes	3	15	18	16.67%	0.000	1
No	19	83	102	18.62%		
Smoking						
Yes	4	31	35	11.43%	0.990	0.320
No	18	67	85	21.18		
Drinking						
Yes	4	18	22	18.18%	0.000	1
No	18	80	98	18.37%		
Borrmann type						
I/ II	9	34	43	20.93%	0.172	0.679
III/IV	13	60	73	17.81%		
Lauran type						
Intestinal	12	56	68	17.65%	0.360	0.835

Diffuse	6	28	34	17.65%		
Mixed	3	9	12	0.25		
Tumor size (cm)						
≥5cm	43	109	152	28.29%	1.264	0.261
<5cm	21	75	96	21.86%		
Grade of differentiation						
Well/ Moderate	10	47	57	17.54%	0.446	0.504
Poor/not	12	41	53	22.64%		
T stage						
T1/2	5	14	19	26.32%	0.001	0.975
T3/4	59	168	227	25.99%		
N stage						
N0/1	23	64	87	26.44%	0.048	0.826
N2/3	40	119	159	25.16%		
M stage						
M0	56	175	231	24.23%	6.193	0.013*
M1	8	7	15	53.33%		
TNM stage						
I	4	9	15	30.77%		
II	13	22	35	37.14%	8.953	0.030*
III	40	143	183	21.86%		
IV	8	7	15	53.33%		
AFP (ng/ml)						
>8.1	4	0	4	100%	14.489	<2.00E-16*
≤8.1	15	92	107	13.51%		
CEA (ng/ml)						
>5	9	25	34	26.47%	3.132	0.077
≤5	10	68	78	12.82%		
CA199 (U/ml)						
>37	9	29	38	23.68%	2.471	0.116
≤37	9	65	74	12.16%		
CA724 (U/ml)						
>6.9	6	20	26	23.08%	1.013	0.314
≤6.9	12	70	82	14.63%		
CA125 (U/ml)						
>35	2	5	7	28.57%	0.080	0.778
≤35	15	75	90	16.67%		
CA50 (U/ml)						
>25	5	14	19	26.32%	1.382	0.240

≤ 25	12	68	80	15.00%		
HER2						
Positive	2	20	22	9.09%	0.599	0.439
Negative	18	78	96	18.75%		
PD-L1						
Positive	4	18	22	18.18%	0.608	0.436
Negative	28	68	96	29.16%		

*Statistically significant ($P < 0.05$).

Reviewer #2 (Remarks to the Author): Expert in multi-omics

Li et al. present a multi-omic characterization of adenocarcinoma of the esophagogastric junction (AEG). The results presented are impressive and represent a comprehensive overview of these tumors from a variety of omics standpoints. Unlike many multi-omic survey manuscripts, the authors included follow up experiments and mouse model work to help validate their findings. The reviewer believes these results are important, but some work is needed before these findings are suitable for publication. In particular, the results describing FBXO44 seem out of place. Next, the authors need to go beyond simply listing feature and pathway differences between their multi-omic subtypes. Each results section reads like its own long list of differences, seemingly without much thought given to how these differences relate to the other results. This work would be greatly strengthened if the authors defined the key phenotypic features of these multi-omic subtypes and how they may be clinically relevant. Further, too much time is spent on comparisons with normal samples. Not surprisingly the normal samples are very different from tumors, and nearly every comparison made by the authors resulted in many hits. The corresponding results have too many genes and/or pathways to easily interpret. A better use of the normal data may be to use it in a more targeted manner. For example, to help screen for cancer-specific targets that were enriched in a particular multi-omic subtype. Finally, the wording throughout the manuscript needs to be more specific, and the statistical tests utilized along with fold changes and p-values need to be provided. In addition to this, please find the major and minor comments to address below.

Response: We appreciate very much for the Reviewer's efforts on reviewing our manuscript. The Reviewer raised many professional comments and valuable suggestions. Our manuscript has been much improved after the revision following these valuable comments and suggestions. Please see detailed revisions in the point-to-point response as follows:

Major comments

Page 6 'Protein Database Searching'

Q1: All of the proteomic findings seem to refer to single protein accessions/names. What happened to the protein group assignments from MaxQuant? The reviewer expects at least some of these protein groups to contain more than one protein accession. Protein group abundances as rows and samples as columns should be included in supplemental.

Response: Thanks very much for the nice suggestion. We have provided a supplemental table, wherein protein group abundances as rows and samples as columns, as Supplemental **Table S2** in our revised submission. This has also been mentioned in the revised manuscript (**Line 4-5, Page 8**).

Q2: The reviewer appreciates the data being deposited into various repositories, but as much as possible the data used to generate the figures presented in the manuscript should be included as supplemental materials. This includes raw gene counts, protein group intensities, called mutations, multi-omic subtyping, etc. at the sample level. Presently, the supplemental data only seems to contain summarized results and fold changes across conditions.

Response: Thanks very much for the nice suggestion. In the revised submission, we provided protein group intensities in Supplemental **Table S2**, called mutations with annotation information in Supplemental **Table S3**, raw gene counts in Supplemental **Table S4**, and subtyping information in Supplemental **Table S9**.

Page 7 'Whole Exome Sequencing' ... 'mRNA Sequencing'

Q3: The description of computational methods for WES and RNA-seq processing are severely deficient. Please provide detailed processing of the raw sequencing data including the alignment, reference genome, QC steps, variant callers, read counting, etc. to allow for reproduction of the presented results. Sample level RNA-seq counts should be included in supplemental.

Response: Thanks for the nice advice. We have provided details of processing the raw WES and RNA-seq data as follows:

Line 6-17, Page 9 (processing WES data): "Adaptors and low-quality reads (q quality score < 20) were removed from raw WES reads by using *fastp* software (version 0.21.0)¹⁰. Then, *BWA* software (version 0.7.17)¹¹ was utilized to align filtered reads to the human reference genome (GRCh38). Alignments were subjected to Picard tools (<http://broadinstitute.github.io/picard/>) to identify and mark duplicate reads. Next, local realignment was performed to correct potential alignment errors around indels. Prior to variant calling, base quality score recalibration was performed to reduce systematic biases. Then, somatic SNVs and InDels were jointly called by Mutect2 (version 4.1.9.0)¹² and Strelka2 (version 2.9.10)¹³. Only variants that passed both quality filtering steps were used in the follow-up analysis. The Variant Effect Predictor (VEP) tool¹⁴ was

utilized to fetch biological information of the variant set. Called mutations with annotation information are supplied in Supplemental **Table S3**."

Line 28-29, Page 9; Line 1-7, Page 10 (processing RNA-seq): " Raw sequencing RNA reads were first trimmed to remove low-quality bases and reads by using *Trimmomatic* software (version 0.39)¹⁵ with default parameters. The filtered reads were then aligned to the human reference genome (GRCh38) by using the splice-aware aligner *HISAT2* (version 2.2.1)¹⁶. Alignment results were subjected to gene quantification with gene annotation from GENCODE (version 35)¹⁷ by adopting *StringTie* software (version 2.14)¹⁸. Gene expression levels were normalized in TPM (transcripts per million mapped reads). Genes with expression levels higher than 0.1 TPM in at least one sample remained for downstream analysis. Raw gene counts are provided in Supplemental **Table S4**."

In addition, the sample-level RNA-seq counts has been provided in Supplemental **Table S4**.

Note: Related references were cited in the revised manuscript.

Page 13 'From the RNA-seq data, 23,131 genes were found to be expressed in 166 AEG tumor and NAT samples...'

Q4: See the above comment on the RNA-seq processing. Include the expression/missingness thresholds used to filter the data (if any). Were there a large number of genes expressed in only the normal/only the tumor?

Response: Thanks very much for the suggestion. We have provided details of processing RNA-seq data in the revised manuscript (please see response to **Q3**). Genes with expression levels higher than 0.1 TPM in at least one sample were remained for downstream analysis. In total, there are 1,500 (0.33% of all detected genes in normal samples) and 6,528 (14.32% of all detected genes in tumor samples) genes expressed in only normal and tumor samples, respectively. It seems that AEG tumor samples express more specific genes.

'...including proteomics profiling, phosphoproteomics profiling, WES, and RNA-seq (Figure 1A).'

Q5: The reviewer had a hard time with the figures. The text in most of the figures is very small, and many, many protein names and/or pathways are listed. Non-key findings should be relegated to supplemental figures or even tables. The authors should take time to consider how their results

are displayed and do a better job at distilling the key findings down to fewer, more interpretable figures.

Response: Thanks very much for the comment and advice. We have enlarged the text font across all figures in the revision. In this revision, we only kept representative or top genes/proteins/pathways in figures. We have carefully revised the manuscript according to the Reviewer's following suggestions. We appreciate very much the Reviewer's valuable comments and suggestions, which have greatly improved our manuscript. Please see the detailed revisions in the following point-to-point response.

Page 14 Proteomic Characteristics of AEG Tumors

Q6: Please see the comment in the opening paragraph about repurposing the normal data and utilizing it to ask specific questions as opposed to showing numerous differences between normal and tumor in each omics type.

Response: Thanks very much for the Reviewer's comment. Our original description may not be clear. It is known that molecular alterations occurred frequently in tumor samples, but the specific alterations of proteome in AEG tumor have not yet systematically investigated. By comparing to the normal samples, we identified differentially expressed proteins and altered biological processes in AEG tumor. Our analysis presented a comprehensive view of proteomic alterations in AEG tumors, and further investigation on their functions and molecular mechanisms in AEG may provide promising drug targets for this disease.

The normal samples were also used to identify subtype-specific alterations. In our study, all NAT samples were collected from regions within ~2 cm around the corresponding AEG tumor sites. Paired tumor-NAT samples were derived from the same patients. To reduce the effect of inter-patient heterogeneity and identify subtype-specific tumor differences, we separately compared tumor with NAT samples in each AEG subtype. These have also been discussed in the revised manuscript (**Line 10-21, Page 26**).

Proteomics-Based Subtyping of AEG Tumors

Q7: The authors need to do a better job at characterizing these subtypes. Yes, they do the bookkeeping in the results and write the expected things about differentially expressed proteins

and pathways, but what is the overarching story? What biology defines these subtypes from one another? Why do the authors think the S1 subtype has the worst survival? Did the immune infiltration later shed any light into this? What is the role of FBXO44, if any, in these subtypes?

Response: Thanks very much for the Reviewer's comment. The molecular alterations of AEG, especially those in proteome, and its molecular subtypes have been obscure. In this study, we presented a comprehensive molecular atlas of AEG, characterizing multi-layer alterations in tumor samples. We identified three proteomic AEG subtypes with significant differences in clinical features and molecular alterations. AEG patients in the S-III subtype had better prognosis than those in the S-I and S-II subtype. We then dissected multi-layer differences between the three subtypes by comparing the genomics, immune infiltration, and phosphoproteomics. In genomics, The SBS1 signature was specifically identified in the S-I subtype, which showed spontaneous or enzymatic deamination of 5-methylcytosine. The S-II subtype exclusively exhibited the mutation signature of APOBEC cytidine deaminase (the SBS2 signature). The mutation signature of "deficiency in base excision repair due to inactivating mutations in NTHL1" (the SBS30 signature) was specifically detected in the S-III subtype. In the aspect of immune infiltration, the abundance of fibroblasts was significantly decreased in the S-III subtype ($P = 2.2E-5$, Student's t test) but showed no obvious changes in tumor samples from the S-I and S-II subtypes. Compared to samples in the S-I and S-II subtypes, our H&E analysis also revealed a decrease in fibroblast abundance of the S-III subtype. Given that fibroblasts may limit the immune cell infiltration to exert the immunosuppressive role in cancer¹⁹, this observation may partly explain that AEG patients in the S-I and S-II subtype had worse prognosis than those in the S-III subtype. In phosphoproteomics, The S-I subtype specifically showed enrichment of IKBKB and PRKDC. HIPK2 kinase was exclusively enriched in the S-II subtype, while CHEK2 and AURKB were specifically enriched in the S-III AEG subtype.

The comparisons of cell abundances between tumor and NAT samples in each subtype revealed pervasive changes in cell abundances across various cell types. Compared to the corresponding NAT samples, tumors in the S-II subtype had the least number of cell types, while the S-III subtype had the most cell types that showed alterations in cell abundance, especially the increase in lymphoid and myeloid cells. Some types of cells exhibited dysregulated abundances in all AEG subtypes. For example, the abundance of activated dendritic cells (aDCs) showed a significant

increase in tumor samples of all three AEG subtypes. **The abundance of fibroblasts was significantly decreased in the S-III subtype ($P = 2.2E-5$, Student's t test) but showed no obvious changes in tumor samples from the S-I and S-II subtypes. Compared to samples in the S-I and S-II subtypes, our H&E analysis also revealed a decrease in fibroblast abundance of the S-III subtype. Given that fibroblasts may limit the immune cell infiltration to exert the immunosuppressive role in cancer¹⁹, this observation may partly explain that AEG patients in the S-I and S-II subtype had worse prognosis than those in the S-III subtype.** These has also been described in the revised manuscript (**Line 6-20, Page 23**).

The FBXO44 was identified as a signature protein of the S-II subtype, which showed specific high expression in the S-II subtype. FBXO44 was demonstrated to promote AEG tumor progression and metastasis *in vitro* and *in vivo*. It needs much more investigation to determine whether FBXO44 play roles in defining the AEG subtype.

We have carefully revised our manuscript according to the valuable comments and suggestions raised by the Reviewer, which has greatly strengthened our manuscript. Please see the detailed revisions in point-to-point response.

'... whereas the "KRAS signaling up" hallmark was significantly down-regulated ($P = 1.1E-3$) in tumor samples (Figure 2D).'

Q8: There are not many KRAS mutations and KRAS does not appear to be a major driver in this cancer. Why is this showing up? Is it a red herring?

Response: Thanks for the comment. The Reviewer is right that there are not many KRAS mutations in this cancer. Only 6 patients were observed with KRAS mutation in our AEG cohort (revised **Figure 1B**). Our original unclear description caused confusions. The "KRAS signaling up" hallmark is the gene set comprises genes up-regulated by KRAS activation²⁰.

'These observations implicate that flutamide might be effective in treating AHR-high AEG patients.'

Q9: The authors should do a better job at supporting this claim. Just because this and other proteins are differentially expressed does not mean that they would make good drug targets. Not all of these will be drivers/critical to the survival of the tumor. Is there publicly available drug screening data

that could be integrated here? Could a network analysis of proteomics findings using protein-protein interaction data identify bottlenecks/hub proteins that would make better drug targets?

Response: Thanks very much for the Reviewer's professional comment and suggestion. Our original description may not be clear. The Reviewer is right that differentially expressed proteins (DEPs) doesn't mean good drug targets. To examine whether these DEPs were targeted by FDA-approved drugs or candidate anti-cancer compounds in clinical trials, we screened datasets of the GDSC²¹, CTRP²², and Broad Institute Drug Repurposing²³ project. In total, 252 DEPs were found to be targets of 195 anti-cancer compounds, which was provided in revised **Supplemental Table S6**. The Reviewer is correct that not all DEPs are drivers/critical to the survival of the tumor. Further investigation on these DEPs that are targeted by known anti-cancer compounds may provide promising drug targets for AEG. These have also been re-worded and described to make the statements clearer in the revised manuscript (**Line 2-8, Page 18**).

Furthermore, in this revision, we retrieved the human protein-protein interactions (PPIs) from the STRING database (v11.5)²⁴. DEPs were then mapped to these PPI relations to generate the DEP PPI network in AEG. Single nodes were removed from the network. We obtained a PPI network of 3,923 nodes and 79,088 edges (**Figure R1-4A**, Supplemental Figure S5A). To identify the hub proteins of this PPI network, the network topology was then analyzed to calculate the degree, closeness, and betweenness of each node (**Figure R1-4B-D**, Supplemental Figure S5B-D, Supplemental Table S7). In this PPI network, the top 10 degree proteins are TP53, HSP90AA1, FN1, HDAC1, CD4, EP300, DHX15, CDK1, FBL, and STAT3 (**Figure R1-4B**), the top 10 closeness proteins are TP53, HSP90AA1, FN1, EP300, CD4, HDAC1, STAT3, CDK1, SIRT1, and CDH1 (**Figure R1-4C**), and the top 10 betweenness proteins are TP53, HSP90AA1, FN1, CD4, CDH1, EP300, STAT3, APP, HDAC1, and SIRT1 (**Figure R1-4D**). Hub proteins of the PPI network may be drug target candidates that are worth of further investigation. These have also been described in the revised Supplemental methods and manuscript (**Line 11-14, Page 18**).

A

Figure R1-4. Network analysis of differentially expressed proteins (DEPs). (A) Protein-protein interactions (PPI) network of DEPs in AEG. The circle size indicates node degree. Red circle represents up-regulated protein, blue circle represents down-regulated protein, and grey circle represents non-change protein. (B) The distribution of network degree of each node. Top 10 proteins with the largest degrees were highlighted. (C) The distribution of network closeness of each node. Top 10 proteins with the largest closeness were highlighted. (D) The distribution of network betweenness of each node. Top 10 proteins with the largest betweenness were highlighted.

To further optimize the candidates that may play crucial roles in AEG, and are potential drug targets, we mapped the top 50 DEPs with the top 50 proteins with the largest degree, closeness, or betweenness. In total, 43 proteins were included in at least two sets of the top 50 DEPs, top 50

proteins with the largest degree, closeness, or betweenness (**Figure R1-5A**). Some of these hub DEPs were found to be targeted by known compounds, such as HDAC1, HSP90AA1, and TP53 (**Figure R1-5B**). This has also been described in the revised manuscript (**Line 14-19, Page xx**).

Figure R1-5. Overlaps of the top 50 DEPs and top 50 proteins with the largest degree, closeness, and betweenness. (A) Venny plot showing the overlaps between the top 50 DEPs and the top 50 proteins with the largest degree, closeness, or betweenness. (B) Heatmap showing the difference in the proteins that are included in at least two sets of top 50 DEPs, top 50 proteins with the largest degree, closeness, or betweenness. Bubble plot on the right shows the degree, closeness, or betweenness of the corresponding proteins in the PPI network.

Note: Related references were cited in the revised manuscript.

Page 15 'Of these, 12 signature proteins were targetable by FDA-approved drugs or candidate drugs currently in clinical trials (Figure 3G).'

Q10: Same comment as above. The reviewer would like this idea developed more and have more supporting evidence shown (computational evidence would be acceptable).

Response: Thanks very much for the Reviewer's comment. Our original statement was not accurate nor clear. We apologize for the misstatement that caused confusions that should have been avoided. These 12 signature proteins showed significant association with patient overall

survival in the univariate Cox regression analysis. The original statement that 12 signature proteins were targetable by FDA-approved drugs or candidate drugs currently in clinical trials was not correct. According to datasets of the GDSC²¹, CTRP²², and Broad Institute Drug Repurposing²³ project, only 1 of 12 proteins was targetable by known anti-tumor compounds. In particular, PKD2 is a target of SKF-96365, which is currently investigated in preclinical phase. Therefore, these proteins are more likely used as prognosis or diagnosis markers for AEG patients in the near future. These proteins can also be target candidates for the development of effective anti-cancer drugs for AEG patients. This statement has also been corrected in the revised manuscript (**Line 2-8, Page 18**).

Note: Related references were cited in the revised manuscript.

Q11: Since the authors bring both up, can they quantify what would be better based on their data: drug targets based on normal to tumor analysis, or drug targets based on proteomic subtypes?

Response: Thanks very much for the comment. The drug targets based on tumor-to-normal analysis may target a wider population, while drug targets based on proteomic subtypes may be more specific and personalized. Due to the heterogeneity of tumors, precision therapy is the direction and trend of research. More and more studies have systematically described the molecular spectrum of tumors through multi-omics technology, and carried out accurate diagnosis and treatment research such as accurate classification of tumors, screening of drug targets and prognosis prediction according to molecular characteristics²⁵⁻²⁹. In our study, all NAT samples were collected from regions within ~2 cm around the corresponding AEG tumor sites. Paired tumor-NAT samples were derived from the same patients. To reduce the effect of inter-patient heterogeneity and identify subtype-specific tumor differences, we separately compared tumor with NAT samples in each AEG subtype. In total, 389, 731, and 630 DEPs in the S-I, S-II, and S-III subtype, respectively were not detected in the analysis of all samples. These results demonstrated that subtype analysis could reveal many subtype-specific candidates that may help personalized therapy of AEG patients.

Figure R1-6. DEPs in different comparisons. (A) Venny plot shows the overlaps among DEPs identified in all AEG, S-I subtype, S-II subtype, S-III subtype samples. (B) Upset plot shows the statistics of DEPs in different comparisons.

Note: Related references were cited in the revised manuscript.

'For example, the activity of the "G2M checkpoint" hallmark in the S-III subtype was significantly higher than those in the other two subtypes ($P = 1.7E-3$ compared to S-I subtype, $P = 1.2E-4$ compared to S-II subtype) (Figure 3E), while "pancreas beta cells" showed remarkably lower levels in the S-III subtype ($P = 1.7E-2$ compared to S-I subtype, $P = 4.3E-2$ compared to S-II subtype) (Figure 3F).'

Q12: The reviewer assumes that the authors mean the large degree of change of expression/abundance of "G2M checkpoint" proteins when they refer to activity. This is confusing because 1) there are activity-based protein profiling (ABPP) proteomic assays that measure activity directly and 2) the authors have phosphoproteomics data. Please reword this section and change the axes of figures that mention activity. Should the figures say fold change instead?

Response: Thanks very much for the professional advice. The original description was not appropriate or unclear. By saying "activity", we meant the integrated abundance of "G2M checkpoint" or "pancreas beta cells" proteins. The integrated abundance of hallmarks was calculated by utilizing the GSVA R package. These have also described in the revised manuscript as follows:

Line 8-13, Page 10: "The hallmark gene sets were retrieved from the Molecular Signatures Database (MSigDB)²⁰. These fifty gene sets were refined from a wide range of biological processes by reducing both variation and redundancy. The integrated abundance of proteins in these hallmarks were then calculated in each sample by utilizing the GSVA R package (version 1.38.2)³⁰. A normalized protein expression matrix was used in the calculation."

In addition, the axes of figures and text that mentioned "activity" have been changed to "integrated abundance" in the revised manuscript.

Note: Related references were cited in the revised manuscript.

Q13: What kind of pathway enrichment is being performed? The reviewer does not remember "pancreas beta cells" being one of the cancer hallmarks. If the type of pathway enrichment or the resource used for pathway enrichment is changing, then the authors should clarify. All pathway results, including non-significant ones, should be included in supplemental tables.

Response: Thanks very much for the comment and suggestion. In this part, we used hallmark gene sets that were downloaded from the Molecular Signature Database²⁰ (MSigDB, <https://www.gsea-msigdb.org/gsea/msigdb/genesets.jsp?collection=H>), which includes 50 gene sets. In each sample, we calculated the integrated abundance of all proteins for each hallmark. Then we compared the difference between tumor and normal samples in each subtype. The results of 50 gene sets were shown in Figure 3D (**Figure R1-7**). We have clarified this in the revised manuscript as follows:

Line 8-13, Page 10: "The hallmark gene sets were retrieved from the Molecular Signatures Database (MSigDB)²⁰. These fifty gene sets were refined from a wide range of biological processes by reducing both variation and redundancy. The integrated abundance of proteins in these hallmarks were then calculated in each sample by utilizing the GSVA R package (version 1.38.2)³⁰. A normalized protein expression matrix was used in the calculation."

Figure R1-7 (revised Figure 3D). The differences of integrated protein abundances of hallmarks comparing tumor and NAT samples in each subtype.

Note: Related references were cited in the revised manuscript.

Q14: Why were these particular pathway findings called out? Were these the most differentially expressed by p-value/FDR? Given the large number of changes, the authors should do a better job at prioritizing what they show and make it clear why they are showing it. Is there an empirical cut point at say the top 5 or 10 pathways before the p-values drop off substantially? That would be one way to prioritize all of these significant findings.

Response: Thanks very much for the comment and nice suggestion. we used hallmark gene sets that were downloaded from the Molecular Signature Database²⁰ (MSigDB, <https://www.gsea-msigdb.org/gsea/msigdb/genesets.jsp?collection=H>), which includes 50 gene sets. We showed results of all these 50 hallmarks in revised **Figure 3D**. These have also been described in the revised manuscript (**Line 8-13, Page 10**).

Note: Related references were cited in the revised manuscript.

Q15: Do AEG tumors share similarities to pancreatic tissue or beta cells?

Response: Thanks for the question. The gastrointestinal (GI) epithelium is a highly regenerative tissue with the potential to provide a renewable source of insulin⁺ cells after undergoing cellular reprogramming. The stomach and intestine are unique among endodermal organs in that they harbor large numbers of adult stem/progenitor cells that constantly produce epithelial cells, including hormone-secreting enteroendocrine cells^{31,32}. Both organs are developmentally related to the pancreas, arising in adjacent embryonic domains³³. Native antral endocrine cells share a surprising degree of transcriptional similarity with pancreatic β cells, and expression of β cell reprogramming factors *in vivo* converts antral cells efficiently into insulin⁺ cells with close

molecular and functional similarity to β cells. Reprogramming of antral stomach cells assembled into bioengineered mini-organs *in vitro* yielded transplantable units that also suppressed hyperglycemia in diabetic mice, highlighting the potential for development of engineered stomach tissues as a renewable source of functional β cells for glycemic control³⁴. This has also been discussed in the revised manuscript (**Line 22-29, Page 26; Line 1-6, Page 27**).

Note: Related references were cited in the revised manuscript.

'Patients in these three subtypes showed significantly distinct overall survival time ($P = 1.1E3$)...'

Q16: What test is this p-value associated with? What is the hazard ratio? There appears to be no survival analysis details in the methods. Has this been adjusted for gender, smoking history, and stage?

Response: Thanks very much for the professional comment. The p value is generated from log-rank test. The hazard ratio with 95% confidence interval is 0.27 (0.12-0.58), which is also provided in the revised Figure 3B (**Figure R1-8**). Clinical variables, including age, sex, smoking history, alcohol history, Siewert type, and tumor stage, were considered in the analysis. We also provided details in the revised manuscript as follows:

Line 5-15, Page 11: "The overall survival time was compared between different groups by using the log-rank test implemented in the *survival* package (version 3.2.3, <https://CRAN.R-project.org/package=survival>). The survival curves were generated by using the Kaplan-Meier method in the R package *survminer* (version 0.4.9, <https://CRAN.R-project.org/package=survminer>). Except for the analysis of subtypes, tumor patients were divided into high- and low-abundance groups by using the median abundances of individual proteins, phosphorylation sites or genes. Hazard ratios with 95% confidence intervals were calculated from the Cox proportional hazards regression analysis. Clinical variables, including age, sex, smoking history, alcohol history, Siewert type, and tumor stage, were used in the Cox regression multivariate analysis."

Figure R1-8. Kaplan-Meier survival curve comparing patients in different subtypes.

Page 16 'FBXO44 Promotes AEG Tumor Progression and Metastasis '

Q17: Why was FBXO44 chosen out of all of the possible choices? Given the sheer volume of data, why were there no better choices based on combining and integrating the findings? This protein does not look like it mentioned in the results until this page.

Response: Thanks very much for the comment. In the original manuscript, we didn't state this clearly. We would like to take this opportunity to clarify why FBXO44 was chosen out. In total, we identified 100 signature proteins in these three AEG subtypes. Of these, 12 signature proteins showed significant association with patient survival time in the univariate Cox regression analysis (as listed in revised Figure 3G). To further prioritize the risk factors independent of others, we performed multivariate Cox regression analysis of these 12 proteins. In the multivariate Cox regression analysis, FBXO44 showed a significantly high unfavorable risk score, while PKD2 and CD3D exhibited remarkably favorable scores (**Figure R1-9**). These results suggested that FBXO44 was a robust risk factor that indicated unfavorable prognosis of AEG patients. Therefore, we chose FBXO44 to further explore its functions in promoting AEG tumor. These has also been described in the revised manuscript (**Line 1-6, Page 20**).

Figure R1-9. Forest plot shows the HR and p values of 12 signature proteins in multivariate Cox regression analysis.

Q18: Were these data generated independently of the multi-omic analyses being presented? These findings seem like they were shoehorned between several sections of global omics characterization as an afterthought to give the results some more clinical relevance.

Response: Thanks very much for the Reviewer's comment. The original manuscript may not state this clearly. We would like to take this opportunity to clarify the idea that we experimentally investigated the tumor promoting role of FBXO44. Following the identification of the three AEG subtypes, we detected signature proteins that showed specific high expression in separate subtypes, which aims to distinguish the three subtypes. We totally identify 100 signature proteins in the three subtypes, of which 12 proteins showed significant association with the overall survival. To further optimize the candidates that may play crucial roles in AEG tumor, we performed multivariate Cox regression analysis. In the multivariate Cox regression analysis, FBXO44 is the only protein that showed significantly unfavorable risk score, which indicated that FBXO44 was worthy of further investigation for its potential role in promoting AEG tumor.

Q19: Please convince the reviewer that this was the best target and similar or better results could not have been obtained looking at the highest fold changes and pathways different across normal/tumor/tumor subtypes.

Response: Thanks very much for the Reviewer's comment. In the original manuscript, we didn't state this clearly. We would like to take this opportunity to clarify why FBXO44 was chosen out. In total, we identified 100 signature proteins in these three AEG subtypes. Of these, 12 signature proteins showed significant association with patient survival time in the univariate Cox regression analysis (as listed in Figure 3G). To further prioritize the risk factors independent of others, we performed multivariate Cox regression analysis of these 12 proteins. In the multivariate Cox regression analysis, FBXO44 showed a significantly high unfavorable risk score, while PKD2 and CD3D exhibited remarkably favorable scores (**Figure R1-9**). These results suggested that FBXO44 was a robust risk factor that indicated unfavorable prognosis of AEG patients. Therefore, we chose FBXO44 to further explore its functions in promoting AEG tumor. These has also been described in the revised manuscript (**Line 1-6, Page 20**).

Figure R1-9. Forest plot shows the HR and p values of 12 signature proteins in multivariate Cox regression analysis.

Q20: Is the biological function of FBXO44 known? If so what pathway is it a part of? How does it relate to the immune findings further down? Is its expression highly correlated with anything interesting across the omics types? If little is known about this protein, then surely some hints about mechanism could be generated with the abundance of data. Even a simple co-expression analysis could shed some light on upstream/downstream targets.

Response: Thanks very much for the professional comment. FBXO44 is a member of the ubiquitin ligase subunit family and contain a conserved G domain that mediates substrate binding³⁵. Lu *et al.* found that SCF(FBXO44) is an E3 ubiquitin ligase responsible for BRCA1 degradation, and FBXO44 expression pattern in breast carcinomas suggests that SCF(FBXO44)-mediated BRCA1 degradation might contribute to sporadic breast tumor development³⁶. Sjögren B, *et al.* identified a novel E3 ligase complex containing cullin 4B (CUL4B), DNA damage binding protein 1 (DDB1) and F-box protein 44 (FBXO44) that mediates RGS2 protein degradation³⁷. Shen *et al.* Found that FBXO44/SUV39H1 are crucial repressors of repetitive elements transcription, and their inhibition selectively induces DNA replication stress and viral mimicry in cancer cells³⁸. It can be seen that FBXO44 may play different roles in different tumors, which is worthy of further study in AEG. We evaluated the associations between FBXO44 and immune cells or checkpoints (**Figure R1-26**). The high expression of FBXO44 was found associated with the low infiltration of Th2 cells and CD4⁺ Tem cells, and also correlated with the high expression of immune checkpoints TNFRSF14, TNFRSF25, CD40, and VTCN1.

Q21: What are the exact fold changes from normal to tumor and across the proteomic subtypes? These values do not look like logged intensities. Why is relative abundance used instead of log2 intensities? Abundances relative to what? Is it not implied that these are relative abundances since the authors did not do absolute quantification?

Response: Thanks very much for the Reviewer's comment. Our original Y-axis name of the protein abundance was not appropriate. All the values were log2-transformation of the normalized iBAQ intensities (the normalization process was described in the Methods). We replaced "relative abundance" by "log2 (normalized iBAQ intensity)" in all revised figures. We also added the exact fold changes (FC) in the revised Figure 4A (**Figure R1-10**).

Figure R1-10. Comparison of FBXO44 protein abundance between tumor and NAT samples in each subtype, and tumors from different subtypes. FC indicates fold change.

Q22: See above survival analysis comments. What is test is this p-value associated with? What is the hazard ratio? There appears to be no survival analysis details in the methods. Has this been adjusted for gender, smoking history, and stage?

Response: Thanks for the comment. The p value is generated from log-rank test. The hazard ratio with 95% confidence interval is 0.48 (0.26-0.88), which is also provided in the revised Figure 4C (Figure R1-11). Clinical variables, including age, sex, smoking history, alcohol history, Sievert type, and tumor stage, were considered in the analysis. We also provided details in the revised manuscript (Line 5-15, Page 11).

Figure R1-11. Kaplan-Meier survival curve comparing FBXO44-high and -low abundance patients.

Q23: Is this simply a protein that is correlated with or involved in cell proliferation therefore explaining the survival curve and mouse model results?

Response: Thanks very much for the Reviewer's professional comment. FBXO44 is the only protein that showed significantly unfavorable risk score in the multivariate Cox regression analysis, which led our further investigation of its tumor promoting role in AEG. FBXO44 is a member of the ubiquitin ligase subunit family and contain a conserved G domain that mediates substrate binding³⁵. Lu *et al.* found that SCF(FBXO44) is an E3 ubiquitin ligase responsible for BRCA1 degradation, and FBXO44 expression pattern in breast carcinomas suggests that SCF(FBXO44)-mediated BRCA1 degradation might contribute to sporadic breast tumor development³⁶. Sjögren B, *et al.* identified a novel E3 ligase complex containing cullin 4B (CUL4B), DNA damage binding protein 1 (DDB1) and F-box protein 44 (FBXO44) that mediates RGS2 protein degradation³⁷. Shen *et al.* Found that FBXO44/SUV39H1 are crucial repressors of repetitive elements transcription, and their inhibition selectively induces DNA replication stress and viral mimicry in cancer cells³⁸. It can be seen that FBXO44 may play different roles in different tumors, which is worthy of further study in AEG.

Note: Related references were cited in the revised manuscript.

Q24: How targetable is this protein/pathway given the authors used shRNA as opposed to a drug?

Response: Thanks very much for the question. In the current study, we would like to investigate the biological function of FBXO44 in the occurrence and development of AEG. Therefore, we used shRNA to observe the changes in the growth, invasion and metastasis ability of cancer cells before and after knocking down FBXO44. We are also trying to optimize one or several drugs that effectively target FBXO44, which may take quite a long time.

'The up-regulation of FBXO44 protein in tumor samples was further validated in an independent clinical cohort (P = 1.55E-4) (Figure 4B). '

Q25: What cohort? AEG patients? How many samples? How much up-regulation? The overall pattern of blue/brown is easy to see in the figure, but the images are too small to see individual features. Please enlarge these images significantly, or if they do not fit in the main figure please include in supplemental.

Response: Thanks very much for the comment and advice. We validated the expression and clinical significance in another AEG cohort of 251 patients. Our analysis found that FBXO44 was highly expressed ($P = 1.55E-4$) in AEG tumor tissues (Positive rate: 24.30%; 61/251), compared with corresponding NAT tissues (Positive rate: 10.36%; 26/251) (**Figure R1-12**). We enlarged these images and put them in the revised figures. These have also been described in the revised manuscript (**Line 17-25, Page 20**).

Figure R1-12. The distribution of FBXO44 in an independent cohort of tumor and NAT samples from 251 AEG patients.

'...*FBXO44* KD remarkably suppressed cell proliferation...'

Q26: Please quantify the amount of suppression instead of using words like remarkable, and please use more precise wording throughout to quantify results when they are mentioned.

Response: Thank very much for the Reviewer's nice suggestion. In this revision, we used quantitative words to replace the words like "remarkable". Please see the revision as follows:

Line 1-14, Page 21: " In OE19 and SK-GT-4 cells, FBXO44 OE promoted cell proliferation by 1.79-fold ($P = 0.031$) and 1.48-fold ($P = 0.029$) (**Figure 4E** and Supplemental **Figure S9C**), increased cell invasion by 1.68-fold ($P = 0.032$) and 2.18-fold ($P = 0.035$) (**Figure 4F** and Supplemental **Figure S9D**), and enhanced cell migration by 2.13-fold ($P = 0.004$) and 1.18-fold ($P = 0.018$) (**Figure 4G** and Supplemental **Figure S9E**), respectively, compared to control cells. In contrast, FBXO44 KD inhibited cell proliferation by 68.1% ($P = 0.002$) and by 49.1% ($P = 0.005$) (**Figure 4E** and Supplemental **Figure S9C**), decreased cell invasion by 79.3% ($P = 0.008$) and 70.9% ($P = 0.001$) (**Figure 4F** and Supplemental **Figure S9D**), and reduced cell migration by 71.8% ($P = 0.005$) and 54.7% ($P = 0.003$) (**Figure 4G** and Supplemental **Figure S9E**) in OE19 and SK-GT-4, respectively. The oncogenic role of FBXO44 in AEG was further validated in the

OE19 xenograft mouse model. We observed that FBXO44 OE increased the growth of AEG xenograft tumors by 2.54-fold (P = 0.004), whereas FBXO44 KD suppressed tumor growth by 67.17% (P = 0.029) *in vivo* (Figure 4H and Supplemental Figure S9F-H)."

Page 19 o 'Phosphoproteomic Characterization of AEG Tumors'

Q27: How important is the phosphoproteomics data in relation to the multi-omics subtypes? Is it a defining feature? Since the phosphorylation data gives a measurement of signaling events in the tumors, is it wise to have these measurements weighted the same as the other omics types (e.g. RNA-seq) even though the phospho-data may be more relevant for activated signaling pathways and drug targets? Do any of these phosphoproteins or kinases relate back to FBXO44?

Response: Thanks very much for the Reviewer's comment. The original description may not be clear. The Reviewer is correct that protein phosphorylation gives a measurement of signalling events in tumor. In our study, the phosphorylation data was not used to define AEG subtype. Actually, the phosphorylation data is not used as a defining feature of tumor subtyping in many proteomics-based tumor subtyping studies^{27,28,39}. Phosphoproteomics, a large-scale analysis of protein phosphorylation sites, has emerged as a powerful tool to identify aberrant phosphorylation-mediated signalling networks that play crucial roles in cancer⁴⁰. In this study, we identified differentially phosphorylated proteins and dysregulated kinase-phosphosubstrate relationships in each AEG subtype, revealing subtype-specific protein phosphorylation. Our analysis revealed differences in kinase-phosphosubstrate regulatory networks between different subtypes and suggested potential personalized responses to clinical therapeutics for AEG patients.

As for the FBXO44, we didn't get any phosphorylated signal of FBXO44 protein in our phosphoproteomics data.

Note: Related references were cited in the revised manuscript.

Minor comments

Page 6 o 'Tandem mass spectra were searched against the Homo_sapiens_9606 database (20,366 entries) concatenated with a reverse decoy database. '

Q28: Is this from Uniprot or another database? Please provide a citation as well as the date accessed.

Response: Yes, it's from the UniProt database⁴¹. The original description was not clear, we have revised this as follows:

Line 25-26, Page 7: "Tandem mass spectra were searched against the human UniProt database (20,366 entries, downloaded on May 9th, 2020)⁴¹ concatenated with a reverse decoy database"

Note: Related references were cited in the revised manuscript.

'The limma package was also adopted to compute the difference of protein and phosphorylation abundances between tumor and paired NAT samples. Specifically, the difference was statistically evaluated by employing a simple linear model and moderated t-statistics by the empirical Bayes shrinkage method.'

Q29: Do the authors think that limma might be too conservative here? The shrinkage method is certainly appropriate for large RNA-seq or microarray datasets, but will meaningful signal be lost applying here to proteomics data?

Response: Thanks for the comment. The Reviewer is right that the shrinkage method is certainly appropriate for large RNA-seq or microarray datasets. Recently, it has been also appropriately used in the analysis of proteomics data²⁹.

Q30: Please cite the limma manuscript and provide the version of the package and the version of R used.

Response: Thanks for the suggestion. In the revised manuscript, we cited the limma manuscript (Ritchie et al., *Nucleic Acids Res.*, 2015)⁴² and provided the version of limma package (version 3.46.0) and R version (4.0.2) (**Line 13, Page 8**).

Page 11 'The xCell scores (relative abundances) were calculated in each sample and were compared between different groups by using Student's t-test.'

Q31: In the reviewer's hands, xCell scores tend to not follow a normal distribution. Do the findings from xCell still hold if a non-parametric Wilcoxon rank sum test is used?

Response: Thanks very much for the Reviewer's comment. We re-computed the differences in relative abundance of infiltrating cells in the three AEG types by using Wilcoxon rank sum test. The vast majority of statistical results from these two tests were consistent (**Figure R1-13**).

Findings in this part still hold in Wilcoxon rank sum test, such as the decreased abundance in fibroblast and the increased abundance in myeloid and lymphoid cells. In addition, Student's t test has also been applied to compare xCell scores in previous studies^{43,44}.

Figure R1-13. The difference in the relative abundance of different infiltrating cells in the three AEG subtypes calculated by Student's t test or Wilcoxon rank sum test.

Page 13 o 'In particular, proteomics and phosphoproteomics profiling were performed on 206 samples (Figure 1B).'

Q32: The circos plot is difficult to read since most of the plot has the same pattern/is not different. This does not seem to convey much meaningful information Please consider a simple table or a different figure to convey the total amount of differentially expressed analytes.

Response: Thanks for the comment and nice advice. We removed the circos plot (original Figure 1B), and provided the corresponding information in revised Supplemental **Table S1**.

Page 13 o 'In the present AEG cohort, the most frequently mutated cancer-related genes (derived from COSMIC v95)³⁷ were TP53 (62%), MUC16 (31%), FAT4 (22%), LRP1B (18%), ARID1A (16%), and FAT3 (16%) (Figure 1C).'

Q33: Please add additional tick marks to the TMB barplot at the top of the heatmap to make it easier to read.

Response: Thanks for the suggestion. We add more tick marks to the TBM bar plot in the revised Figure 1B (**Figure R1-14**).

Figure R1-14. Barplot shows the mutation burden (TMB) in each patient.

'Overall, significantly larger number of proteins ($P = 3.8E-15$), phosphorylation sites ($P = 1.6E4$), and genes ($P < 2.2E-16$) were detected in AEG tumors than those in NAT samples (Supplemental Figure S2).'

Q34: What test was used here? T-test? Fisher's exact test? Please make sure to provide the name of the test along with the p-value, or alternatively make a list of the tests performed for the different data types in the methods.

Response: Thanks very much for the nice suggestion. These p values were generated from the Wilcoxon rank sum test. We have provided the test name along with the p value in the revised manuscript.

Line 8-11, Page 17: "Overall, significantly more proteins ($P = 3.8E-15$, Wilcoxon rank sum test), phosphorylation sites ($P = 1.6E-4$, Wilcoxon rank sum test), and genes ($P < 2.2E-16$, Wilcoxon rank sum test) were detected in AEG tumors than in NAT samples"

Page 14 'We next investigated the disturbance of proteins in AEG tumors. Differential protein analysis revealed 2,300 up-regulated and 1,667 down-regulated proteins in AEG tumor samples compared to paired NAT samples (Figure 2A and Supplemental Table S3).'

Q35: For 2A and all figures, the authors should use HUGO gene symbols instead of Uniprot accessions. Readers will be much more familiar with gene symbols and it will make results easier to interpret. Uniprot provides mapping tables for this purpose

Response: Thanks very much for the Reviewer's nice suggestion. In the revised submission, we have replaced the Uniprot accessions by HUGO gene symbols in Figure 2A and all other figures.

Page 18 'Tumors in the S-II subtype showed the least number of changed cell types, while the S-III subtype exhibited the most altered cell types, especially the increase of lymphoid and myeloid cells.'

Q36: Altered from what? Please reword. Is this not inherent representation of immune cells in a given proteomic subtype? Is this referring to "differential expression" of the xCell scores?

Response: Thanks for the comment. The original description was not clear. By saying "changed cell types" or "altered cell types", we meant immune cell types that showed altered xCell scores (relative cell infiltrating abundance) in tumor samples compared to NAT samples. These have also been re-worded in the revised manuscript as follows:

Line 8-11, Page 23: "Compared to the corresponding NAT samples, tumors in the S-II subtype had the least number of cell types, while the S-III subtype had the most cell types that showed alterations in cell abundance, especially the increase in lymphoid and myeloid cells"

'The xCell algorithm was employed to infer the relative cell abundance of 41 different cell types (see Methods).'

Q37: Why only 41? The reviewer believes xCell can estimate more than this. The table of xCell scores and p-values should be provided as supplemental.

Response: Thanks for the comment and suggestion. The Reviewer is correct that xCell has more than 41 cell types. Our original description was not clear. The xCell method curated gene signatures of 64 different cell types. In our analysis, we removed those cell types that were not relevant in AEG tissue, such as hepatocytes, keratinocytes, and osteoblast. Cell types that had a xCell score of 0 across all samples were also removed. In total of 41 cell types were involved in subsequent analysis. The xCell raw scores, transformed scores, and p-values of these 41 cell types are provided in Supplemental Table S11. This has also been described in the revised manuscript as follows:

Line 8-11, Page 14: "In the 64 cell types curated by the xCell method, we removed those that were not relevant in AEG tissues, such as hepatocytes, keratinocytes, and osteoblast. We then removed those cell types that had a xCell score of 0 across all samples. A total of 41 cell types were involved in subsequent analysis, ..."

Figure 2 A

Q38: The reviewer had a hard time reading these volcano plots. The shading makes them almost uninterpretable. These do not need the sample frequency since presumably the authors did some type of filtering for missingness before the data was presented.

Response: Thanks for the suggestion. We removed the circle size that represents the sample frequency in the revised Figure 2A (**Figure R1-15**). To make the figure more interpretable, we also set a more stringent cutoff ($FDR < 0.01$ and $|\log_2(\text{fold change})| > 1$) for coloring, and replaced Uniport accessions by gene symbols.

Figure R1-15. Volcano plot shows the difference of proteins between AEG tumor and paired NAT samples. Red circles represent up-regulated proteins ($FDR < 0.01$ and $\log_2(\text{fold change}) > 1$) and blue circles indicate down-regulated proteins ($FDR < 0.01$ and $\log_2(\text{fold change}) < -1$).

Q39: Gene symbols should be used instead of Uniport accessions.

Response: Thanks for the nice advice. We have replaced the Uniport accessions by gene symbols in the revised Figure 2A (**Figure R1-15**).

Q40: The reviewer was going to comment that the non-significant findings should be colored black or dark grey with the significant findings colored red/blue, but it now appears like this has already been done. Something needs to be done to help with the interpretability of these figures. Maybe only coloring the top foldchange/FDR hits? A much more stringent cutoff?

Response: Thanks very much for the nice advice. The original Figure 2A was hard to interpret. We made substantial revision according to the Reviewer's suggestion. We removed the circle size that represents the sample frequency, set a more stringent cutoff ($FDR < 0.01$ and $|\log_2(\text{fold change})| > 1$) for coloring, and replaced Uniport accessions by gene symbols (**Figure R1-15**). The figure legend has also been revised accordingly.

Figure R1-15. Volcano plot shows the difference of proteins between AEG tumor and paired NAT samples. Red circles represent up-regulated proteins ($FDR < 0.01$ and $\log_2(\text{fold change}) > 1$) and blue circles indicate down-regulated proteins ($FDR < 0.01$ and $\log_2(\text{fold change}) < -1$).

Figure 6 A, D

Q41: The dots/bubbles are very small and hard to see/interpret. Please increase their size to aid in interpretability. There does not need to be so much empty space in between them.

Response: Thanks very much for the comment. In the revised Figure 6, we increased the bubble size and narrow down the empty space between them (**Figure R1-16**, revised Figure 6B and 6F).

Figure R1-16. (A) The difference of relative abundances of different infiltrating cells in three AEG subtypes. (B) The differential significance of protein expression of immune checkpoints across three AEG subtypes.

Q42: Do these have -logp and -logfdr abbreviated with the "o" taken out? Please write them out.

Response: Thanks for the comment. We corrected those to "-logP" and "-logFDR" in the revised figures.

Q43: Justify using p-values for one and FDR for the other. Why not be consistent? Do the results still hold with FDR? That would be acceptable as long as the results are clearly stated.

Response: Thanks for the Reviewer's comment. The FDR values were not calculated in the original figure. In this revision, we calculated the FDR values. Most of the significance still hold when set the threshold of FDR as 0.05 (**Figure R1-17**). This result has been updated in the revised Figure 6B-D.

Figure R1-17 (revised Figure 6B). The difference in the relative abundance of different infiltrating cells in the three AEG subtype.

Figure 7A

Q44: Similar comments above about this volcano plot.

Response: Thanks very much for the advice. To make Figure 7A more interpretable, we removed the circle size that represents the sample frequency, set a more stringent cutoff ($FDR < 0.01$ and $|\log_2(\text{fold change})| > 1$) for coloring, and replaced Uniport accessions by gene symbols (**Figure R1-18**). The figure legend has also been revised accordingly.

Figure R1-18. Volcano plot shows the differential significance of phosphorylation sites. Red circles represent up-regulated phosphorylation sites ($FDR < 0.01$ and $\log_2(\text{fold change}) > 1$) and blue circles indicate down-regulated phosphorylation sites ($FDR < 0.01$ and $\log_2(\text{fold change}) < -1$).

Figure 7B

Q45: If the pathway enrichment is shown in C, then are these nearly identical heatmaps really needed?

Response: Thanks for the Reviewer's comment. These heatmaps conveyed no additional information, we removed them in the revised Figure 7B.

Figure 7C

Q46: The text direction is the opposite of other figures. Please be consistent.

Response: Thanks for the comment. The text direction of Figure 7C has been reversed to keep consistent with those in other figures, and only kept genes that had a p value < 0.05 (Figure R1-19, revised Figure 7C). We also enlarged the text of genes.

Figure R1-19 (revised Figure 7C). Kinase enrichment of differential phosphosites in each AEG tumor subtype.

Q47: There are too many genes listed.

Response: Thanks for the comment. We only kept genes that had a p value < 0.05, and reversed the text direction to keep consistent with those in other figures (Figure R1-19, revised Figure 7C). We also enlarged the text of genes.

Figure R1-19 (revised Figure 7C). Kinase enrichment of differential phosphosites in each AEG tumor subtype.

Figure 7D-F

Q48: What are readers supposed to gain from these? There is too much going on here. Please consider hiding the nodes that are not relevant or highlighting the key nodes.

Response: Thanks very much for the comment and nice advice. There were too many text in the original Figure 7D-E, which prevented a better interpretation. We only kept the text of significant nodes, and highlighted the nodes by black border (Figure R1-20, revised Figure 7D-F). Also, we replaced the protein accession by gene symbol. The full list of known kinase-phosphosubstrate correlations were provided in the revised Supplemental Table S11.

Figure R1-20. Kinase-phosphosubstrate regulatory networks in tumors of the S-I (A), S-II (B), and S-III (C) subtype.

Supplemental Figure S1

Q49: Was the phosphoproteomics data normalized the same as the protein expression? The distribution of the boxplots look different.

Response: Yes, the phospho-proteomic data was normalized the same as the protein expression by using the method described in a previous study²⁹. The slight difference of distribution may be caused by the high variability of phosphorylation across samples. In our datasets, we identified 3,967 differentially expressed proteins and 8,078 differentially phosphorylated sites.

Note: Related references were cited in the revised manuscript.

Supplemental Figure S4

Q50: Have all of the survival analyses been adjusted for clinical variables mentioned above? What about false discovery? Hazard ratios should always be reported.

Response: Thanks for the comment. The p value is generated from log-rank test. Clinical variables, including age, sex, smoking history, alcohol history, Siewert type, and tumor stage, were considered in the analysis. We provided details in the revised manuscript (**Line 5-15, Page 11**). We also provide hazard ratios with 95% CI values in the revised figure (**Figure R1-21**, revised Supplemental Figure 8).

Figure R1-21. Kaplan-Meier survival curves of 11 druggable signature proteins between corresponding high- and low-abundance patient groups.

Supplemental Figure S5 o C, D

Q51: These images are too small to see. Please make them larger and include scale bars on any other histology/ICC/IHC images throughout.

Response: Thanks for the comment. The original Supplemental Figure S5C and S5D were too small. In the revised submission, we adjusted the space of Supplemental Figure S5 to enlarge S5C and S5D (**Figure R1-22**, revised Supplemental Figure S9).

Figure R1-22 (revised Supplemental Figure S9). FBXO44 promotes AEG tumor progression and metastasis.

Reviewer #3 (Remarks to the Author): Expert in AEG subtypes

Considerable effort is appreciated. There are the following major issues:

Response: We appreciate very much for reviewing efforts of the Reviewer on our manuscript. We carefully revised the manuscript according to the valuable comments and suggestions raised by the Reviewer, which has largely improved our manuscript. Please see detailed revisions in the following point-to-point response.

Q1: AEG subtype (Siewert type I, II, and III) have different biology and cannot be combined as such. Overall samples size is rather small. According to the Table s1. there are no Siewert type I patients in the cohort studied. 'Again, there are only 4 patients with Siewert type II (gastroesophageal junction). These should be removed. Therefore, what is left in the cohort are Siewert type III and some gastric cancer patients. Essentially, not a study of 3 types of upper GI tumors.

Response: Thanks very much for Reviewer's professional comment. In the original Supplemental Table S1, we didn't accurately describe the distance/location from the tumor center to the esophagogastric junction. We apologize for this mistake that caused confusions that should have been avoided. In this revision, we provided the exact distance/location and corresponding Siewert types in revised Supplemental Table S1. Please see the location and Siewert type information of each patient in **Table R1-2**. We also collected and supplied the corresponding gastroscopy and pathology report of each patient for review. Please see one example report of each Siewert type in **Figure R1-23**. In total, we included 27 Siewert type I, 31 Siewert type II, and 45 Siewert type III patients in this study.

Table R1-2. The location of primary tumor and Siewert type information of 103 AEG patients.

Patient ID	Primary tumor location	Distance from the tumor center to the esophagogastric junction (cm)	Siewert type
AEG001	Cardia, gastric fundus	1.5	II
AEG002	Cardia, gastric body	3.5	III
AEG003	Cardia, lower esophageal	1.8	I
AEG004	Cardia, gastric body	2.3	III

AEG005	Cardia, lower esophageal	2.5	I
AEG006	Cardia, gastric body	4.5	III
AEG007	Cardia, gastric body	5	III
AEG008	Cardia, lower esophageal	1.5	I
AEG009	Cardia, lower esophageal	1.8	I
AEG010	Cardia, lower esophageal	1.5	I
AEG011	Cardia, gastric body	2.1	III
AEG012	Cardia, lower esophageal	1.2	I
AEG013	Cardia, gastric body	3	III
AEG014	Cardia	1.5	II
AEG015	Cardia, gastric fundus	2.5	III
AEG016	Cardia, gastric fundus	1.5	II
AEG017	Cardia, gastric fundus, partly gastric body	2.5	III
AEG018	Cardia, gastric body	3	III
AEG019	Cardia, gastric body	3	III
AEG020	Cardia, lower esophageal	1.2	I
AEG021	Cardia, lower esophageal	2	I
AEG022	Cardia, gastric fundus	1.5	II
AEG023	Cardia, gastric fundus	1.5	II
AEG024	Cardia, gastric fundus	2.5	III
AEG025	Cardia, gastric fundus	1	II
AEG026	Cardia, gastric fundus	2.2	III
AEG027	Cardia, gastric fundus	1.5	II
AEG028	Cardia, gastric fundus	1	II
AEG029	Cardia, lower esophageal	1.8	I
AEG030	Cardia, gastric fundus	1.5	II
AEG031	Cardia, gastric body	3	III
AEG032	Cardia, gastric fundus	2.2	III
AEG033	Cardia, gastric body	2.5	III
AEG034	Cardia, lower esophageal	1.1	I
AEG035	Cardia, gastric body	3	III
AEG036	Cardia, gastric body	4	III
AEG037	Cardia, lower esophageal	1.4	I
AEG038	Cardia, lower esophageal	2	I
AEG039	Cardia, gastric fundus	1	II
AEG040	Cardia, gastric body	3	III
AEG041	Cardia, gastric body	3	III
AEG042	Cardia, gastric fundus	2.5	III
AEG043	Cardia, gastric body	3	III

AEG044	Cardia, gastric body	2.5	III
AEG045	Cardia, lower esophageal	2	I
AEG046	Cardia, lower esophageal	1.6	I
AEG047	Cardia, gastric body	5	III
AEG048	Cardia, lower esophageal	2	I
AEG049	Cardia, gastric body	2.5	III
AEG050	Cardia, lower esophageal	2.5	I
AEG051	Cardia, gastric fundus	3	III
AEG052	Cardia, gastric body	3	III
AEG053	Cardia, lower esophageal	1.8	I
AEG054	Cardia, gastric body	3	III
AEG055	Cardia, gastric body	3.5	III
AEG056	Cardia, lower esophageal	2.5	I
AEG057	Cardia, gastric body, gastric fundus	3	III
AEG058	Cardia, gastric body	2.5	III
AEG059	Cardia, lesser curvature of stomach	1.5	II
AEG060	Cardia, gastric body	4.2	III
AEG061	Cardia, gastric fundus	3.5	III
AEG062	Cardia, lower esophageal	1.3	I
AEG063	Cardia, gastric body	3	III
AEG064	Cardia, gastric body	2.3	III
AEG065	Cardia, gastric fundus	2.5	III
AEG066	Cardia	1	II
AEG067	Cardia, lesser curvature of stomach	1.5	II
AEG068	Cardia	1	II
AEG069	Cardia, gastric body	2.5	III
AEG070	Cardia, lower esophageal	1.2	I
AEG071	Cardia, gastric fundus	1.5	II
AEG072	Cardia lesser curvature of gastric body	2.5	III
AEG073	Cardia, lesser curvature of stomach	1.8	II
AEG074	Cardia, gastric fundus	2.3	III
AEG075	Cardia, gastric fundus	1.7	II
AEG076	Cardia, gastric fundus, lesser curvature of stomach	2.8	III

AEG077	Cardia, gastric fundus	1.8	II
AEG078	Cardia, lesser curvature of stomach	1.4	II
AEG079	Cardia	1.5	II
AEG080	Cardia, gastric fundus	2.2	III
AEG081	Cardia, lower esophageal	1.7	I
AEG082	Cardia, lesser curvature of stomach	1.7	II
AEG083	Cardia, lesser curvature of stomach, gastric body	1.5	II
AEG084	Cardia, gastric fundus	1.3	II
AEG085	Cardia, gastric fundus	3	III
AEG086	Cardia, lower esophageal	1.2	I
AEG087	Cardia, gastric body	4	III
AEG088	Cardia, gastric body	2.8	III
AEG089	Esophagogastric junction	1.8	II
AEG090	Cardia, lower esophageal	1.3	I
AEG091	Cardia, lower esophageal	2.5	I
AEG092	Cardia, gastric fundus	1	II
AEG093	Cardia, lower esophageal	2	I
AEG094	Cardia, gastric fundus	1.5	II
AEG095	Esophagogastric junction	1.5	II
AEG096	Cardia, gastric body	2.2	III
AEG097	Cardia, gastric fundus	1.2	II
AEG098	Cardia, gastric body	3	III
AEG099	Cardia, lower esophageal	1.8	I
AEG100	Esophagogastric junction	1.6	II
AEG101	Esophagogastric junction	1	II
AEG102	Esophagogastric junction	1.5	II
AEG103	Cardia, lower esophageal	1.6	I

Figure R1-23. The gastroscopic and pathological report of the Siewert type I, II, and III patient.

Q2: The two cell lines studied (OE19 and Sk-GT-4) are Siewert type I cell lines and not relevant in this study.

Response: Thanks very much for the comment. The original description about Siewert type in our AEG cohort was not accurate nor clear. We apologize for this mistake that caused confusions that should have been avoided. Actually, we have 27 Siewert type I, 31 Siewert type II, and 45 Siewert type III AEG patients in our cohort (also see response to Q1). Currently, AEG cell lines mainly include OE19, SK-GT-4, OACP4, and OACM5.1C. Only OE19 and SK-GT-4 are available in China now. Obtaining other cell lines are now difficult because of COVID-19 epidemic prevention policies in different countries. Therefore, we used the OE19 (OE19 was established in 1993 from a 72-year-old male patient with gastric cardia adenocarcinoma⁴⁵) and SK-GT-4 (SK-GT-4 was established in 1989 from the primary tumor of an 89-year-old Caucasian male with an adenocarcinoma of the distal esophagus^{46,47}) cell line for validation, and obtained expected results that FBXO44 promotes AEG tumor progression and metastasis.

Note: Related references were cited in the revised manuscript.

Q3: All tumors (almost) are of high localized stage and with varied survival. The overall, survival analysis fails to correlate molecular subtypes with phenotypes/histotypes.

Response: Thanks very much for the Reviewer's comment. Yes, the Reviewer is correct that most of the tumors in our study are of high localized stage. Briefly, we included 28 AEG patients with TNM stage I/II, and 75 AEG patients with TNM stage III/IV. In China, the early diagnosis rate of esophageal cancer, gastric cancer, and AEG is less than 20%⁴⁸⁻⁵⁰. Patients in these three proteomics-based subtypes showed significantly distinct overall survival time (revised **Figure 3B**, P = 0.0011, log-rank test). Clinical variables, including age, sex, smoking history, alcohol history, Siewert type, and tumor stage, were considered in the overall survival analysis.

Note: Related references were cited in the revised manuscript.

Q4: The manuscript claims that multiomics analysis has not been done, which is not true. TCGA STAD included 4 times more patients and was much more comprehensive. Similarly, the Samsung paper not quoted. The authors have not acknowledged TCGA subtypes and validated their findings.

Response: Thanks very much for the Reviewer's comment. The original statement was not clear nor accurate. **By saying "multi-omics analysis of AEG has not been done", we meant the proteomics-based multi-omics analysis of AEG.** This has been corrected and discussed in the revised manuscript (**Line 2-13, Page 3; Line 15, Page 25**). In our study, we included proteomics, phosphoproteomics, genomics, and transcriptomics. Other studies that performed multi-omics analysis of AEG focused on genomics and transcriptomics. The TCGA Research Network analyzed 295 primary gastric adenocarcinomas using six molecular platforms, including array-based somatic copy number analysis, whole-exome sequencing, array-based DNA methylation profiling, messenger RNA sequencing, microRNA (miRNA) sequencing, and reverse-phase protein array (RPPAR)⁵¹. They classified gastric cancer into four subtypes: tumors positive for Epstein-Barr virus; microsatellite unstable tumors; genomically stable tumors; tumors with chromosomal instability, which was mainly dependent on genomics data.

Cristescu *et al.* (the Samsung paper) used transcriptomics data to describe four molecular subtypes of gastric cancer, including the mesenchymal-like type, microsatellite-unstable type, and the tumor protein 53 (TP53)-active and TP53-inactive types⁵². The subtyping was primarily based on gene expression signatures.

Other studies related to AEG subtyping based on omics data mainly including genomics and transcriptomics data^{7,51,53-56}. These studies included no proteomics data, so we didn't validate our findings in these datasets.

Note: Related references were cited in the revised manuscript.

Q5: Figure 1. Remove AEG I and II (as there are no AEG I tumors in this study and there are only 4 AEG II and they should be removed from the analysis as they do not provide useful data).

Response: Thanks very much for the comment. In the original Supplemental Table S1, we didn't accurately describe the distance/location from the tumor center to the esophagogastric junction. We apologize for this mistake that caused confusions that should have been avoided. In this revision, we provided the exact distance/location and corresponding Siewert types in revised Supplemental Table S1. Please see the location and Siewert type information of each patient in **Table R1-2** (please see response to Q1). We also collected and supplied the corresponding gastroscopy and pathology report of each patient for review. Please see one example report of each Siewert type in **Figure R1-23** (please see response to Q1). In total, we included 27 Siewert type I, 31 Siewert type II, and 45 Siewert type III patients in this study.

Q6: In the introduction, "surgical resection is most effective" cannot be generalized. It is acknowledged that surgery is essential for cure but multimodality is commonly practiced. Surgery first may be a Chinese approach and should be qualified.

Response: Thanks very much for the professional comment. Our original statement was not accurate. We corrected this statement in the revised manuscript as follows:

Line 19-20, Page 3: "Currently, comprehensive treatment, including surgical resection, chemotherapy, and immunotherapy, is the most effective treatment for AEG"

Q7: In the introduction, there should be mention of novel studies with IO

Response: Thanks very much for the Reviewer's professional advice. We introduced the current state of immunotherapy for AEG patients in the revised manuscript as follows:

Line 23-26, Page 3: " With the use of PD1/PD-L1 inhibitors, the immunotherapy of AEG has made significant progress. However, due to the heterogeneity and complexity of immune microenvironment, immunotherapy still has many challenges, such as hyperprogression⁵⁷."

Note: Related references were cited in the revised manuscript.

Q8: The normal tissue is seemingly appropriate for some comparisons but it is expected that once some proteins are differentially expressed in tumor/normal, repetitive analysis of tumor v normal (Figures 1D, 1E, and 1F are not very informative).

Response: Thanks very much for the Reviewer's comment. The original description may not be clear enough. These figures presented the number distributions of detected proteins, phosphorylation sites, and genes in paired tumor and NAT samples. On average, 8,885 proteins (revised **Figure 1C**) and 8,445 phosphorylation sites (revised **Figure 1D**) were identified from the 206 proteomes and phosphoproteomes of 103 AEG patients. From the RNA-seq data, 23,131 genes were found to be expressed in 166 AEG tumor and NAT samples on average (revised **Figure 1E**). Overall, significantly more proteins ($P = 3.8E-15$, Wilcoxon rank sum test), phosphorylation sites ($P = 1.6E-4$, Wilcoxon rank sum test), and genes ($P < 2.2E-16$, Wilcoxon rank sum test) were detected in AEG tumors than in NAT samples (revised Supplemental **Figure S3**). This observation indicates that compared with NATs, AEG tumors might show abnormally higher molecular activity. These have been described in the revised manuscript (**Line 5-12, Page 17**).

Q9: Similarly, Figure 2A distracts from what we can learn about tumors. Same for Figures 2D and 2C.

Response: Thanks very much for the Reviewer's comment. The original Figure 2A was hard to interpret. Figure 2A showed the results of differential protein analysis, which revealed 2,300 up-regulated and 1,667 down-regulated proteins in AEG tumor samples compared to paired NAT samples (Figure 2A and Supplemental Table S3). To make it more interpretable, we made substantial revision of Figure 2A. We removed the circle size that represents the sample frequency, set a more stringent cutoff ($FDR < 0.01$ and $|\log_2(\text{fold change})| > 1$) for coloring, and replaced Uniport accessions by gene symbols (**Figure R1-15**, revised Figure 2A). The figure legend has also been revised accordingly. To further examine the changes of key biological processes in AEG tumor, the overall protein-level integrated abundances of fifty hallmark biological processes were evaluated in each sample (see Methods). Figure 2C, 2D, and 2E showed the results and representative examples of the alterations and significance of hallmarks in AEG tumor. Most of the hallmarks (36 out of 50, 72%) showed significantly distinct integrated abundance between paired tumor and NAT samples (Figure 2C). For example, the "apical junction" hallmark gene set

was remarkably up-regulated ($P = 2.40E-16$), whereas the "KRAS signaling up" hallmark gene set was significantly down-regulated ($P = 1.1E-3$) in tumor samples (Figure 2D). Higher integrated abundances of the "apical junction" hallmark gene set indicate a worse prognosis ($P = 1.6E-2$), while the higher integrated abundance of "KRAS signaling up" indicated a longer overall survival time in AEG patients ($P = 3.3E-3$) (Figure 2E). These results revealed extensive dysregulation of hallmark biological processes in AEG tumors, which also showed clinical significance.

Figure R1-15 (revised Figure 2A). Volcano plot shows the difference of proteins between AEG tumor and paired NAT samples. Red circles represent up-regulated proteins ($FDR < 0.01$ and $\log_2(\text{fold change}) > 1$) and blue circles indicate down-regulated proteins ($FDR < 0.01$ and $\log_2(\text{fold change}) < -1$).

Q10: Proteomics did not provide the location of these proteins (cell surface, nuclear, cytoplasmic, or total).

Response: Thanks very much for the Reviewer's comment. In our study, total proteins were extracted from each sample to generate proteomics data. This has also been described in the revised manuscript (Line 4, Page 6).

Q11: Figure 3 is interesting. 3 types (S-I, S-II, and S-III) are not correlated with phenotypes/histologies. Types S-I and S-II are similar in prognosis. It is not clear what may be promoting better survival in S-III when one reviews Figure 3D (many oncogenes are up-MYC and

cell cycle). Angiogenesis is down can make sense. OxPhos down can make sense but need better interpretation from the authors. and correlate with clinical variables.

Response: Thanks very much for the Reviewer's professional comment and nice suggestion. In this revision, we added clinicopathological characteristics, including age, sex, smoking, alcohol, Siewert type and tumor stage, to each AEG patient of the three subtypes (**Figure R1-24**, revised Figure 3A). The S-I subtype was significantly associated with older age (75% \geq 65 years old, $P = 0.0093$, Fisher's exact test). The Siewert type II patients were more enriched in the S-I subtype, while the S-III subtype had many more Siewert type III patients ($P = 0.011$, Fisher's exact test). The three AEG subtypes showed no differences in the other clinicopathological features. This has also been described in the revised manuscript (**Line 28-29, Page 18; Line 1-4, Page 19**).

Patients in the S-III AEG subtype had the longest overall survival than those in the S-I and S-II subtype. The MYC-regulated and cell cycle-related genes were observed upregulation in all subtypes, which was not supposed to explain the survival differences. The exclusive downregulation of cancer-associated pathways, such as WNT/ β -catenin signaling and Hedgehog signaling, may explain the better survival of AEG patients in the S-III subtype. In addition, the abundance of fibroblasts was significantly decreased in the S-III subtype ($P = 2.2E-5$, Student's t test) but showed no obvious changes in tumor samples from the S-I and S-II subtypes (revised Figure 6D). Compared to samples in the S-I and S-II subtypes, our H&E analysis also revealed a decrease in fibroblast abundance of the S-III subtype (revised Figure 6E). Given that fibroblasts may limit the immune cell infiltration to exert the immunosuppressive role in cancer¹⁹, this observation may partly explain that AEG patients in the S-I and S-II subtype had worse prognosis than those in the S-III subtype.

Figure R1-24 (revised Figure 3A). Heatmap showing the differentially expressed proteins among the three subtypes. Tiling bars above the heatmap show the distribution of different clinicopathological characteristics among the three subtypes.

Q12: Figure 4G. why include normals here??? Why normals in different subtypes are different? Were they not obtained from a distant gastric location? If so, are the differences related to cancer? Very confusing.

Response: Thanks very much for the comment. Our original description may not be clear. In our study, all NAT samples were collected from regions within ~2 cm around the corresponding AEG tumor sites. Paired tumor-NAT samples were derived from the same patients. To reduce the effect of inter-patient heterogeneity and identify subtype-specific tumor differences, we separately compared tumor with NAT samples in each AEG subtype. Only 27.2% of differentially expressed proteins that were identified in all AEG samples showed dysregulation in subtype comparisons (**Figure R1-25A**). In the subtype tumor-NAT comparison, 300, 636, and 523 differentially expressed proteins that showed no dysregulation in the comparison of all AEG samples were identified in the S-I, S-II, and S-III subtype, respectively (**Figure R1-25B**).

Furthermore, to identify the specific molecular alterations in our proteomic subtypes, we compared the protein abundances between tumor samples in individual subtypes with those in tumor and NAT samples of the other subtypes. In each subtype, a protein that showed remarkably higher abundances than all NAT samples and tumor samples in the other subtypes was considered a signature protein.

In light of above results, subtype tumor NAT comparison was applied to identify subtype-specific alterations that might be negligible in the comparison of all AEG samples.

Figure R1-25. Differential proteins in different comparisons. **(A)** Venny plot shows the overlaps among differential proteins in all AEG, S-I subtype, S-II subtype, and S-III subtype samples. **(B)** Upset plot shows the numbers of differential proteins in different comparisons.

Q13: Figure 5A. again, inclusion of normals does not seem to add much here. Confusing for S-I.

Response: Thanks very much for the Reviewer's comment. Our original description may not be clear. In our study, all NAT samples were collected from regions within ~2 cm around the corresponding AEG tumor sites. Paired tumor-NAT samples were derived from the same patients. To reduce the effect of inter-patient heterogeneity and identify subtype-specific tumor differences, we separately compared tumor with NAT samples in each AEG subtype. subtype tumor NAT comparison was applied to identify subtype-specific alterations that might be negligible in the comparison of all AEG samples (see more details in response to Q12). To identify signature proteins of each subtype, the tumor-NAT comparison was performed in each subtype. FBXO44

was identified as a signature protein of the S-II AEG subtype. In particular, the FBXO44 protein exhibited significantly higher abundance in S-II AEG tumor samples than in S-II normal samples, S-I tumor samples, and S-III tumor samples.

Q14: the finding that FBXO44 is associated with poor outcome in multiple cancer patients (their ref 38) is not novel. In ref 38, those authors have produced significant high quality data and the current manuscript provides no novelty. It would appear that it would be difficult to target FBXO44 but it could serve as a marker to use IO. these authors could have considered those studies.

Response: Thanks very much for the Reviewer's comment and advice. FBXO44 is a member of the ubiquitin ligase subunit family and contain a conserved G domain that mediates substrate binding³⁵. The Reviewer is right that FBXO44 have been reported in some cancers. Lu *et al.* found that SCF(FBXO44) is an E3 ubiquitin ligase responsible for BRCA1 degradation, and FBXO44 expression pattern in breast carcinomas suggests that SCF(FBXO44)-mediated BRCA1 degradation might contribute to sporadic breast tumor development³⁶. Sjögren B, *et al.* identified a novel E3 ligase complex containing cullin 4B (CUL4B), DNA damage binding protein 1 (DDB1) and F-box protein 44 (FBXO44) that mediates RGS2 protein degradation³⁷. Shen *et al.* Found that FBXO44/SUV39H1 are crucial repressors of repetitive elements transcription, and their inhibition selectively induces DNA replication stress and viral mimicry in cancer cells³⁸. It can be seen that FBXO44 may play different roles in different tumors, which is worthy of further study. **In the original ref 38, the research systematically studied the role of FBXO44 in the development of cancer, but the article mainly verified the role of FBXO44 in breast cancer, lung cancer, colon cancer and brain glioma. There was no data related to AEG.** In pan-cancer analysis, we found that the FBXO44 gene showed significant dysregulation in eight of 18 different tumor types wherein FBXO44 showed up-regulation in colon cancer but showed no significant expression change in stomach cancer (revised Supplemental **Figure S9A**). Considering the differences between AEG and other tumors, we verified the role of FXBO44 in the development of AEG.

In this revision, we further calculated the correlations between FBXO44 and different immune cells, and immune checkpoint genes. In our AEG data, FBXO44 was found to be correlated to plasma cells, central memory CD4⁺ T cells (CD4⁺ Tcm), T helper type 2 cells (Th2 cells), and effector memory CD4⁺ T cells (CD4⁺ Tem) (**Figure R1-26A**, revised Supplemental Figure S12A). In particular, the high abundance of plasma cells, Th2 cells, and CD4⁺ Tem was significantly

associated with low expression level of FBXO44 (**Figure R1-26B**). CD4⁺ Tcm was evaluated to have relative abundance > 0 in only 6 samples, so it was discarded. The immune checkpoint genes TNFRSF14, TNFRSF25, CD40, and VTCN1 were found to correlated with the expression of FBXO44 (**Figure R1-26C**). The high expression level of FBXO44 was significantly associated with the expression of TNFRSF14, TNFRSF25, CD40, and VTCN1 (**Figure R1-26D**). Collectively, these results suggested that FBXO44 could be a potential marker in the immunotherapy of AEG targeting or related to these immune cells or checkpoints.

Figure R1-26 (Supplemental Figure S12). Associations of FBXO44 with immune cells and immune checkpoints. (A) The correlations between FBXO44 and different immune cells. **(B)** Box plots showing the relative abundance of plasma cells, CD4⁺ Tcm, Th2 cells, and CD4⁺ Tem between FBXO44-low and -high samples. **(C)** The correlations between FBXO44 and immune checkpoint genes. **(D)** Box plots showing the expression level of TNFRSF14, TNFRSF25, CD40, and VTCN1 between FBXO44-low and -high samples.

Q15: Integration of various platform remains elusive. Need better description and plan. Integration with clinical variables would be more meaningful.

Response: Thanks very much for the Reviewer's comment. In this study, we presented a proteomic-based multi-omics profiling for AEG tumors, including genomics, transcriptomics, proteomics, and phosphoproteomics. We characterized the proteogenomic alterations in AEG tumors and classified AEG into three different subtypes based on proteomics data. The three subtypes showed significant differences in clinical features and molecular alterations. We identified signature proteins in each subtype, and experimentally validated the tumor promoting role of FBXO44 that showed highly unfavorable risk score in multivariate Cox regression analysis. We then dissected multi-layer differences between subtypes by comparing the genomics, immune infiltrations, and phosphoproteomics. In this revision, we integrated clinical variables in corresponding comparisons.

Q16: Subtypes I, II, and III were derived by proteomics data and by integrated analysis. The significance remains unclear. Subtypes not integrated with clinical variables.

Response: Thanks very much for the comment. In this submission, we performed integrated analysis of AEG subtypes and clinicopathologic characteristics, including age, sex, smoking, alcohol, Siewert type, and tumor stage (**Figure R1-27**, revised **Figure 3A**). These clinicopathologic characteristics exhibited no significant differences between these three AEG subtypes except for age and Siewert type. The S-I subtype had significantly more older patients (≥ 65 years old, 75%, $P = 0.0093$, Fisher's exact test). The Siewert type II patients were more enriched in the S-I subtype, while the S-III subtype had remarkably more Siewert type III patients ($P = 0.011$, Fisher's exact test). The proteomics-based subtyping remained an independent prognostic factor when adjusted for other clinicopathological characteristics in multivariate Cox regression analysis ($P = 0.002$). These were also described in the revised manuscript (**Line 28-29, Page 18; Line 1-4, Page 19**).

Figure R1-27. (A) Heatmap shows the differential proteins among three subtypes. Tiling bars above heatmap show the distribution of different clinicopathological characteristics among three subtypes. (B) Multivariate Cox regression analysis of clinicopathological characteristics and the proteomics-based subtyping.

Q17: there is useful information on TME analysis. but again not correlated with clinical phenotypes. Not integrated.

Response: Thanks very much for the professional comment. In addition to the infiltration differences in separate subtypes, we compared the infiltration of different cells between the three AEG subtypes in this revision (**Figure R1-28**, revised Figure 6A and Supplemental Figure S12). The infiltration of some cell types showed significant differences between the three AEG subtypes, such as regulatory T cells and fibroblasts, but none of them have associations with clinicopathological features of AEG patients. For example, the S-II AEG tumor samples showed lower abundance of gamma delta T cells, regulatory T cells, and plasmacytoid dendritic cells, whereas they had higher infiltration of fibroblasts, lymphatic endothelial cells, and microvascular endothelial cells, compared to those of the S-I and S-II subtype (**Figure R1-28B**).

Figure R1-28. (A) Heatmap shows the relative abundance of different cells across samples of the three AEG subtypes. The Kruskal-Wallis Rank Sum test was used to compare the differences between subtypes. (B) Box plots show the comparisons of different cell types between the three AEG subtypes.

Q18: Genomics of subtypes is noted but not integrated to the extent it can be done.

Response: Thanks very much for the Reviewer's comment. In this revision, we matched clinicopathological features of AEG patients with the genomic mutation (Figure R1-29A, revised Figure 1B). We then compared the differences in tumor mutation burdens (TMB) between patients with different clinicopathological features, including age, sex, smoking status, alcohol status, Siewert type, and tumor stage. AEG patients of older age were found to harbor higher TMB (P = 0.045, Wilcoxon rank sum test), while other clinicopathological features showed no obvious association with the TMB (Figure R1-29B, revised Supplemental Figure S1). In addition, the three subtypes showed specific mutation signatures. The SBS1 signature was specifically identified in the S-I subtype, which showed spontaneous or enzymatic deamination of 5-methylcytosine (revised Figure 5A). The S-II subtype exclusively exhibited the mutation signature of APOBEC cytidine deaminase (the SBS2 signature) (revised Figure 5B). The mutation signature of

"deficiency in base excision repair due to inactivating mutations in NTHL1" (the SBS30 signature) was specifically detected in the S-III subtype (revised **Figure 5C**).

Figure R1-29. Genomic mutation landscape of AEG patients. (A) The genomic profiles of AEG patients. The top panel shows the mutation burden in each patient. The top bars show the clinicopathological features of AEG patients. The middle panel is the oncoplot generated with *maftools* depicting the top 20 mutated genes in the present AEG cohort. The bottom panel shows the proportion of different types of nucleotide substitutions in each patient. The right panel represents mutation types and frequencies for each gene. **(B)** Box plots showing the differences of TMB between patients with different clinicopathological features.

Q19: A lot of analyses are descriptive and correlative. Not highly informative.

Response: Thanks very much for the Reviewer's comment. Some descriptions in the original manuscript may be not accurate nor clear. In this study, we aimed to portray the molecular landscape and identify the molecular subtypes of AEG. We conducted proteomics and

phosphoproteomics profiling of 103 AEG tumors with paired normal adjacent tissues (NATs), whole exome sequencing (WES) of 94 tumor-NAT pairs, and RNA sequencing (RNA-seq) in 83 tumor-NAT pairs. Our proteomic analysis revealed an extensively altered proteome and identified 252 potential druggable proteins in AEG tumors. We identified three proteomic subtypes with significant differences in clinical features and molecular alterations. One of the S-II subtype signature proteins, FBXO44, was demonstrated to promote AEG tumor progression and metastasis *in vitro* and *in vivo*. Our comparative analyses revealed distinct genomic features in AEG subtypes. Tumor microenvironment infiltration analysis revealed that the S-III subtype had a specific decrease of fibroblasts. Further phosphoproteomic comparisons revealed different kinase-phosphosubstrate regulatory networks among the three subtypes, such as Occludin S408 phosphorylation by CSNK2A1 in the S-II subtype. Our proteogenomics dataset provides a valuable resource for better understanding the molecular mechanisms of AEG and the development of precision treatment strategies for AEG patients. We have carefully revised the manuscript according to the valuable comments and suggestions raised by the Reviewer, which largely improved our manuscript.

Q20: Discussion has many misstatements and unfocused emphasis.

Response: Thanks very much for the Reviewer's comment. We corrected misstatements in the original discussion, such as the statement of "the first multi-omics profiling for AEG". We also added more discussion to clarify and discuss our findings, such as the pairwise tumor-NAT comparisons in subtypes and the hallmark gene set analysis.

Q21: It is unclear if these data provide a step forward as prior studies were not placed in context.

Response: Thanks very much for the Reviewer's perspective comment. In our study, we included proteomics, phosphoproteomics, genomics, and transcriptomics. **Other studies that performed multi-omics analysis of AEG focused on genomics and transcriptomics.** The TCGA Research Network analyzed 295 primary gastric adenocarcinomas using six molecular platforms, including array-based somatic copy number analysis, whole-exome sequencing, array-based DNA methylation profiling, messenger RNA sequencing, microRNA (miRNA) sequencing, and reverse-phase protein array (RPPAR)⁵¹. They classified gastric cancer into four subtypes: tumors positive for Epstein-Barr virus; microsatellite unstable tumors; genomically stable tumors; tumors with chromosomal instability, which was mainly dependent on genomics data. Cristescu *et al.* (the

Samsung paper) used transcriptomics data to describe four molecular subtypes of gastric cancer, including the mesenchymal-like type, microsatellite-unstable type, and the tumor protein 53 (TP53)-active and TP53-inactive types⁵². The subtyping was primarily based on gene expression signatures. Other studies related to AEG subtyping based on omics data **mainly including genomics and transcriptomics** data^{7,51,53–56}. In our study, **we determined three proteomic subtypes with significant differences in clinical features and molecular alterations**.

FBXO44 was identified as a signature protein in the S-II AEG subtype. Previous studies have demonstrated that FBXO44 may play different roles in different tumors^{36–38}. There was no data related to AEG. In pan-cancer analysis, we found that the FBXO44 gene showed significant dysregulation in eight of 18 different tumor types wherein FBXO44 showed up-regulation in colon cancer but showed no significant expression change in stomach cancer. Considering the differences between AEG and other tumors, we verified the role of FBXO44 in the development of AEG.

Tumor microenvironment infiltration analysis revealed that the abundance of fibroblasts was significantly decreased in the S-III subtype but showed no obvious changes in tumor samples from the S-I and S-II subtypes. Compared to samples in the S-I and S-II subtypes, our H&E analysis also revealed a decrease in fibroblast abundance of the S-III subtype. Given that fibroblasts may limit the immune cell infiltration to exert the immunosuppressive role in cancer¹⁹, this observation may partly explain that AEG patients in the S-I and S-II subtype had worse prognosis than those in the S-III subtype.

Protein kinases, which modulate the phosphorylation of proteins, have been developed as operable drug targets in the treatment of cancer^{58,59}. Phosphoproteomics, a large-scale analysis of protein phosphorylation sites, has emerged as a powerful tool to identify aberrant phosphorylation-mediated signaling networks that play crucial roles in cancer⁴⁰. Kinases and phosphorylation have not been systematically investigated in AEG. In this study, we identified differentially phosphorylated proteins and dysregulated kinase-phosphosubstrate relationships in each AEG subtype, revealing subtype-specific protein phosphorylation. Our analysis revealed differences in kinase-phosphosubstrate regulatory networks between different subtypes and suggested potential personalized responses to clinical therapeutics for AEG patients.

These results of our study provide a step forward as prior studies.

Note: Related references were cited in the revised manuscript.

References

1. Cao, F. *et al.* Current treatments and outlook in adenocarcinoma of the esophagogastric junction: a narrative review. *Ann. Transl. Med.* **10**, 377–377 (2022).
2. Saito, T. *et al.* Treatment response after palliative radiotherapy for bleeding gastric cancer: a multicenter prospective observational study (JROSG 17-3). *Gastric Cancer* **25**, 411–421 (2022).
3. Qiu, M. Z. *et al.* Clinicopathological characteristics and prognostic analysis of Lauren classification in gastric adenocarcinoma in China. *J. Transl. Med.* **11**, 1–7 (2013).
4. Abdi, E., Latifi-Navid, S., Zahri, S., Yazdanbod, A. & Pourfarzi, F. Risk factors predisposing to cardia gastric adenocarcinoma: Insights and new perspectives. *Cancer Med.* **8**, 6114–6126 (2019).
5. Tomb, J. F. *et al.* The complete genome sequence of the gastric pathogen *Helicobacter pylori*. *Nature* **389**, 412 (1997).
6. Backert, N. T. S. *Helicobacter pylori*-Induced Changes in Gastric Acid Secretion and Upper Gastrointestinal Disease. *Subcell Biochem* vol. 400 (2017).
7. Mun, D.-G. *et al.* Proteogenomic Characterization of Human Early-Onset Gastric Cancer. *Cancer Cell* **35**, 111–124 (2019).
8. Li, C. *et al.* Integrated Omics of Metastatic Colorectal Cancer. *Cancer Cell* **38**, 734–747.e9 (2020).
9. Ge, S. *et al.* A proteomic landscape of diffuse-type gastric cancer. *Nat. Commun.* **9**, 1–16 (2018).
10. Chen, S., Zhou, Y., Chen, Y. & Gu, J. Fastp: An ultra-fast all-in-one FASTQ preprocessor. *Bioinformatics* **34**, i884–i890 (2018).

11. Li, H. & Durbin, R. Fast and accurate short read alignment with Burrows-Wheeler transform. *Bioinformatics* **25**, 1754–1760 (2009).
12. Cibulskis, K. *et al.* Sensitive detection of somatic point mutations in impure and heterogeneous cancer samples. *Nat. Biotechnol.* **31**, 213–219 (2013).
13. Kim, S. *et al.* Strelka2: fast and accurate calling of germline and somatic variants. *Nat. Methods* **15**, 591–594 (2018).
14. McLaren, W. *et al.* The Ensembl Variant Effect Predictor. *Genome Biol.* **17**, 1–14 (2016).
15. Bolger, A. M., Lohse, M. & Usadel, B. Trimmomatic: a flexible trimmer for Illumina sequence data. *Bioinformatics* **30**, 2114–2120 (2014).
16. Kim, D., Langmead, B. & Salzberg, S. L. HISAT: a fast spliced aligner with low memory requirements. *Nat. Methods* **12**, 357–60 (2015).
17. Frankish, A. *et al.* GENCODE reference annotation for the human and mouse genomes. *Nucleic Acids Res.* **47**, D766–D773 (2019).
18. Pertea, M. *et al.* StringTie enables improved reconstruction of a transcriptome from RNA-seq reads. *Nat. Biotechnol.* **33**, 290–295 (2015).
19. Barrett, R. L. & Pure, E. Cancer-associated fibroblasts and their influence on tumor immunity and immunotherapy. *Elife* **9**, 1–20 (2020).
20. Liberzon, A. *et al.* The Molecular Signatures Database Hallmark Gene Set Collection. *Cell Syst.* **1**, 417–425 (2015).
21. Iorio, F. *et al.* A Landscape of Pharmacogenomic Interactions in Cancer. *Cell* **166**, 740–754 (2016).
22. Basu, A. *et al.* An Interactive Resource to Identify Cancer Genetic and Lineage Dependencies Targeted by Small Molecules. *Cell* **154**, 1151–1161 (2013).
23. Corsello, S. M. *et al.* The Drug Repurposing Hub: A next-generation drug library and information resource. *Nat. Med.* **23**, 405–408 (2017).

24. Szklarczyk, D. *et al.* The STRING database in 2021: Customizable protein-protein networks, and functional characterization of user-uploaded gene/measurement sets. *Nucleic Acids Res.* **49**, D605–D612 (2021).
25. Krug, K. *et al.* Proteogenomic Landscape of Breast Cancer Tumorigenesis and Targeted Therapy. *Cell* 1436–1456 (2020).
26. Xu, J. Y. *et al.* Integrative Proteomic Characterization of Human Lung Adenocarcinoma. *Cell* vol. 182 245-261.e17 (2020).
27. Gillette, M. A. *et al.* Proteogenomic Characterization Reveals Therapeutic Vulnerabilities in Lung Adenocarcinoma. *Cell* **182**, 200-225.e35 (2020).
28. Chen, Y. J. *et al.* Proteogenomics of Non-smoking Lung Cancer in East Asia Delineates Molecular Signatures of Pathogenesis and Progression. *Cell* **182**, 226-244.e17 (2020).
29. Jiang, Y. *et al.* Proteomics identifies new therapeutic targets of early-stage hepatocellular carcinoma. *Nature* **567**, 257–261 (2019).
30. Hänzelmann, S., Castelo, R. & Guinney, J. GSVA: Gene set variation analysis for microarray and RNA-Seq data. *BMC Bioinformatics* **14**, 7 (2013).
31. Barker, N. *et al.* Lgr5+ve Stem Cells Drive Self-Renewal in the Stomach and Build Long-Lived Gastric Units In Vitro. *Cell Stem Cell* **6**, 25–36 (2010).
32. May, C. L. & Kaestner, K. H. Gut endocrine cell development. *Mol. Cell. Endocrinol.* **323**, 70–75 (2010).
33. Levine, M. S. PDX-1 is required for pancreatic outgrowth and differentiation of the rostral duodenum. *Development* **122**, 983–985 (1996).
34. Ariyachet, C. *et al.* Reprogrammed Stomach Tissue as a Renewable Source of Functional β Cells for Blood Glucose Regulation. *Cell Stem Cell* **18**, 410–421 (2016).
35. Glenn, K. A., Nelson, R. F., Wen, H. M., Mallinger, A. J. & Paulson, H. L. Diversity in tissue expression, substrate binding, and SCF complex formation for a lectin family of ubiquitin ligases. *J. Biol. Chem.* **283**, 12717–12729 (2008).

36. Lu, Y. *et al.* The F-box Protein FBXO44 Mediates BRCA1 ubiquitination and degradation. *J. Biol. Chem.* **287**, 41014–41022 (2012).
37. Sjögren, B., Swaney, S. & Neubig, R. R. FBXO44-mediated degradation of RGS2 protein uniquely depends on a cullin 4B/DDB1 complex. *PLoS One* **10**, 1–18 (2015).
38. Shen, J. Z. *et al.* FBXO44 promotes DNA replication-coupled repetitive element silencing in cancer cells. *Cell* **184**, 352–369.e23 (2021).
39. Integrated Proteogenomic Characterization across Major Histological Types of Pediatric Brain Cancer. *Cell* **183**, 1962–1985 (2020).
40. Gerritsen, J. S. & White, F. M. Phosphoproteomics: a valuable tool for uncovering molecular signaling in cancer cells. *Expert Rev. Proteomics* **18**, 661–674 (2021).
41. Bateman, A. *et al.* UniProt: the universal protein knowledgebase in 2021. *Nucleic Acids Res.* **49**, D480–D489 (2021).
42. Ritchie, M. E. *et al.* limma powers differential expression analyses for RNA-sequencing and microarray studies. *Nucleic Acids Res.* **43**, e47 (2015).
43. Gustave Roussy, C. C. G. P. & Hoffmann-La, R. Molecular correlates of clinical response and resistance to atezolizumab in combination with bevacizumab in advanced hepatocellular carcinoma. *Nat. Med.* (2022) doi:10.1038/s41591-022-01868-2.
44. Lehmann, B. D. *et al.* Multi-omics analysis identifies therapeutic vulnerabilities in triple-negative breast cancer subtypes. *Nat. Commun.* **12**, 1–18 (2021).
45. Yin, X. *et al.* Diallyl disulfide inhibits the metastasis of type II esophageal-gastric junction adenocarcinoma cells via NF- κ B and PI3K/AKT signaling pathways in vitro. *Oncol. Rep.* **39**, 784–794 (2018).
46. Boonstra, J. J. *et al.* Verification and unmasking of widely used human esophageal adenocarcinoma cell lines. *J. Natl. Cancer Inst.* **102**, 271–274 (2010).
47. De Both, N. J., Wijnhoven, B. P. L., Sleddens, H. F. B. M., Tilanus, H. W. & Dinjens, W. N. M. Establishment of cell lines from adenocarcinomas of the esophagus and gastric cardia growing in vivo and in vitro. *Virchows Arch.* **438**, 451–456 (2001).

48. Cao, M., Li, H., Sun, D. & Chen, W. Cancer burden of major cancers in China: A need for sustainable actions. *Cancer Commun.* **40**, 205–210 (2020).
49. Zhang, T. *et al.* Changing trends of disease burden of gastric cancer in China from 1990 to 2019 and its predictions: Findings from Global Burden of Disease Study. *Chinese J. Cancer Res.* **33**, 11–26 (2021).
50. Zhou, J. *et al.* Gastric and esophageal cancer in China 2000 to 2030: Recent trends and short-term predictions of the future burden. *Cancer Med.* **11**, 1902–1912 (2022).
51. Bass, A. J. *et al.* Comprehensive molecular characterization of gastric adenocarcinoma. *Nature* **513**, 202–209 (2014).
52. Cristescu, R. *et al.* Molecular analysis of gastric cancer identifies subtypes associated with distinct clinical outcomes. *Nat. Med.* **21**, 449–456 (2015).
53. Wang, K. *et al.* Whole-genome sequencing and comprehensive molecular profiling identify new driver mutations in gastric cancer. *Nat. Genet.* **46**, 573–582 (2014).
54. Lin, Y. *et al.* Genomic and transcriptomic alterations associated with drug vulnerabilities and prognosis in adenocarcinoma at the gastroesophageal junction. *Nat. Commun.* 1–14 (2020).
55. Suh, Y. S. *et al.* Comprehensive Molecular Characterization of Adenocarcinoma of the Gastroesophageal Junction between Esophageal and Gastric Adenocarcinomas. *Ann. Surg.* **275**, 706–717 (2022).
56. Hao, D. *et al.* Integrated genomic profiling and modelling for risk stratification in patients with advanced oesophagogastric adenocarcinoma. *Gut* 1–11 (2020).
57. Janjigian, Y. Y. *et al.* First-line nivolumab plus chemotherapy versus chemotherapy alone for advanced gastric, gastro-oesophageal junction, and oesophageal adenocarcinoma (CheckMate 649): a randomised, open-label, phase 3 trial. *Lancet* **398**, 27–40 (2021).
58. Islam, S., Wang, S., Bowden, N., Martin, J. & Head, R. Repurposing existing therapeutics, its importance in oncology drug development: Kinases as a potential target. *Br. J. Clin. Pharmacol.* 1–11 (2021).

59. Verbaanderd, C., Meheus, L., Huys, I. & Pantziarka, P. Repurposing Drugs in Oncology: Next Steps. *Trends in Cancer* **3**, 543–546 (2017).

Reviewers' Comments:

Reviewer #1:

Remarks to the Author:

I was originally impressed with this manuscript. Their Response to Reviewers is one of the most complete and compelling I have seen and addresses both the conceptual and informatics issues regarding the gastroesophageal cancers. The depth of their responses reflects a control of the information that is simply beyond those presented in the field to date. I believe this work will set a new and enviable standard for the application of multiomics to tease out the complexities and perhaps vulnerabilities of this diverse set of cancers that so far are beyond our ability to treat.

Reviewer #2:

Remarks to the Author:

The authors have done a great job addressing my concerns. The manuscript and figures are much improved. I recommend this for publication.

Reviewer #3:

Remarks to the Author:

Thank you for all your responses.

I have the following comments and queries:

Please review the AEG TCGA paper published in Nature in 2017 and review all the different subgroup identified and then compare (can validate your findings in those data or their findings in your data). I did not mention this last time because you had said you had very few or no AEG I and II.

Essentially, through proteomics you identified S-I, S-II, and S-III. Then you make some correlations with WES. It is not a true integromics.

Figure 3B (granted you have very small number of patients) S-I and S-II are about the same but S-III is surviving longer.

Fig 3C. Mutation frequency per se may have no meaning at all.

Figure 3G. remove normals to see what the heat map looks like

FBXO44 is a known oncogene and Figure 4C is consistent with it but not really novel. Figure 4A. Hard to find distinction in S subtypes. No explanation

Figures 5 ABC also don't explain why S-III are surviving longer

DDRd should confer longer survival (but it is all over the place). Also APOBEC should be with longer survivors but it is not.

Figure 5G is also not very instructive. Mostly oncogenes but distributions are not striking.

Similarly, Figure 6D. not striking for S-I and S-III. FDR is very high

Figure 6 F also not giving any clues why S-III should live longer.

Point-by-point Response (NCOMMS-22-09290A)

Reviewer #1 (Remarks to the Author):

I was originally impressed with this manuscript. Their Response to Reviewers is one of the most complete and compelling I have seen and addresses both the conceptual and informatics issues regarding the gastroesophageal cancers. The depth of their responses reflects a control of the information that is simply beyond those presented in the field to date. I believe this work will set a new and enviable standard for the application of multiomics to tease out the complexities and perhaps vulnerabilities of this diverse set of cancers that so far are beyond our ability to treat.

Response: We are very delighted that our revision satisfied all the Reviewer's concerns. We greatly appreciate the Reviewer's recognition of our efforts in addressing all the comments. Thanks again for the valuable comments and suggestions that have helped much improved our original manuscript.

Reviewer #2 (Remarks to the Author):

The authors have done a great job addressing my concerns. The manuscript and figures are much improved. I recommend this for publication.

Response: We are pleased to hear that all the Reviewer's concerns have been addressed. Thanks again for the valuable comments and suggestions that have helped much strengthened our manuscript.

Reviewer #3 (Remarks to the Author):

Thank you for all your responses. I have the following comments and queries:

Response: Thanks again for the valuable comments and suggestions raised by the Reviewer in last revision, which has helped much improve our manuscript. We appreciate very much for the comments and advice raised in this version. We have carefully revised the manuscript according to these comments and suggestions, which strengthened our last version of manuscript. Please see the detailed point-by-point responses as follows:

Q1: Please review the AEG TCGA paper published in Nature in 2017 and review all the different subgroup identified and then compare (can validate your findings in those data or their findings in your data). I did not mention this last time because you had said you had very few or no AEG I and II.

Response: Thanks very much for the nice suggestion. In the *Nature* paper¹, the TCGA group analyzed the molecular profiling of 559 oesophageal and gastric carcinoma. The major subdivision of these samples was based on anatomic data, i.e., oesophageal, gastric or indeterminate origins. Tumors were mainly categorized into oesophageal squamous cell carcinoma (ESCC), oesophageal adenocarcinoma (EAC), adenocarcinomas of gastroesophageal junction (GEJ), and gastric carcinomas. **They compared the molecular differences between these subtypes, divided ESCC into three molecular subtypes based on multi-omics data, and related EAC to gastric cancer.** By reviewing the gastroesophageal locations of cancer, we retrieved 129 samples that were regarded as AEG. **The most frequent genomic alterations in the TCGA AEG cohort were captured in our cohort (Figure R2-1, revised Supplementary Fig. 1).** In particular, 15 of top 30 mutated genes in our cohort were also among the top 30 in the TCGA cohort (**Figure R2-1a**). Of note, 9 of top 10 mutated genes in our cohort were among the top mutated genes of the TCGA cohort. Genes with top 20 frequent CNVs in the TCGA cohort were also found to be frequently altered in our cohort (**Figure R2-1b**). Compared to other types, GEJ cancer is featured with TP53 mutations, ERBB2 and VEGFA amplification in the TCGA cohort¹. Mutated TP53, amplified ERBB2 and VEGFA were also frequent in our cohort. **Compared to the TCGA study, our study was**

more specific to molecular subtypes among AEG tumors. These have also been described in the revised manuscript (Line 130-136).

The TCGA study included whole-exome sequencing (WES), single-nucleotide polymorphism (SNP) array profiling to somatic copy-number alterations (SCNAs), DNA methylation profiling and mRNA and microRNA sequencing. We performed proteomics, phosphoproteomics, WES, and RNA sequencing in Chinese AEG cohort. **In addition to the genomic findings, our study provided proteomic insights into AEG molecular subtypes.**

Figure R2-1. Frequency genomic alterations in the TCGA and our AEG cohorts. a Top 30 mutated genes in the TCGA and our AEG cohorts. **b** Top 20 CNV genes in the TCGA AEG cohort and their frequency in our cohort.

Q2: Essentially, through proteomics you identified S-I, S-II, and S-III. Then you make some correlations with WES. It is not a true integromics.

Response: Thanks very much for the Reviewer's comment. In the last version of manuscript, we compared the genomic differences between the three proteomic subtypes. To further integrate the genomics and proteomics data, we examined how subtype-specific mutations influence proteins in this revision (**Figure R2-2**, revised Supplementary Fig. 9). The consequence of mutation on protein was evaluated by compare the T/N (tumor/normal) values between mutation and wild-type samples as described in a previous study². For each mutated

genes, we examined changes of all the possible proteins. We identified 65,184, 3,900, and 1,146 significant mutation-to-protein associations in the S-I subtype, S-II subtype, and S-III subtype, respectively (**Figure R2-2a**). In all three subtypes, over 60 percent are negative associations, i.e., most mutations directly or indirectly led to the decrease of protein levels. Here, we showed the top 5 mutation-protein associations of the top 5 mutated genes (**Figure R2-2b-d**). Please see all the significant results in the Supplementary Data 8. These have also been described in the revised manuscript (Line 211-220).

Figure R2-2. Significant effects of selected subtype-specific mutations on the proteins. **a** Pie charts show the percentages of up-regulated and down-regulated mutation-to-protein associations in the S-I subtype, S-II subtype, and S-III subtype, respectively. The top 5 mutation-protein associations of the top 5 mutated genes in the S-I subtype (**a**), S-II subtype (**b**), and S-III subtype (**c**).

Q3: Figure 3B (granted you have very small number of patients) S-I and S-II are about the same but S-III is surviving longer.

Response: Thanks very much for the Reviewer's comment. The Reviewer is correct that patients of the S-I and S-II subtype showed no significant difference in overall survival time. But AEG subtyping remained an independent prognostic factor after adjusting for multiple clinicopathological characteristics (**Figure R2-3**, revised Supplementary Fig. 7), including age, sex, smoking status, alcohol status, Siewert type and tumor stage. In our multifaceted analysis,

we revealed extensive molecular differences between the three AEG subtypes. We found 97, 143, and 29 specifically mutated genes in the S-I, S-II, and S-III subtypes, respectively (Fig. 3C and Supplementary Data 11). For example, *LEPR* mutation was most common in the S-I subtype (OR = 20.1, P = 2.8E-4, Fisher's exact test), *NCKAPI* mutation was most common in the S-II subtype (OR = 10.5, P = 5.8E-3, Fisher's exact test), and *WIZ* mutation was most common in the S-III subtype (OR = 10.0, P = 7.5E-3, Fisher's exact test) (revised Supplementary Fig. 7d). Our analysis also found 36, 54, and 10 signature proteins in the S-I, S-II, and S-III subtypes, respectively. These signature proteins could be used differential diagnosis as for AEG subtypes. Different AEG subtypes were enriched for distinct lists of kinases, and the same kinases showed different levels of activities in the S-I, S-II, or S-III subtypes (Fig. 7c). CDK2 and CDK7 were highly enriched in all three subtypes. The S-I subtype specifically showed enrichment of IKBKB and PRKDC. HIPK2 kinase was exclusively enriched in the S-II subtype, while CHEK2 and AURKB were specifically enriched in the S-III AEG subtype. In addition, the abundance of fibroblasts was significantly decreased in the S-III subtype (FDR = 2.6E-4, Student's t test) but showed no obvious changes in tumor samples from the S-I (FDR = 0.48, Student's t test) and S-II (FDR = 0.98, Student's t test) subtypes (Fig. 6d). **Although the S-I and S-II subtypes have no survival difference, they are significantly distinguished in molecular alterations that could be potential markers as differential diagnosis and precision therapeutics.**

In molecular subtype studies of other cancer types, it's common that patients of some subtypes show no difference in survival time. For example, Li *et al.* identified three proteomic subtypes in metastatic colorectal cancer, i.e., the CC1, CC2, and CC3 subtype³. Patients of the CC1 and CC2 subtype have no significant differences in survival time. In a proteomics study of hepatocellular carcinoma (HCC)⁴, Jiang *et al.* found three proteomic subtypes (the S-I, S-II, and S-III subtypes). HCC patients of the S-I and S-II subtypes showed no difference in survival time.

Figure R2-3. Multivariate Cox regression analysis of clinicopathological characteristics and the proteomics-based subtyping.

Q4: Fig 3C. Mutation frequency per se may have no meaning at all.

Response: Thanks for the comment. We removed the mutation frequency in Figure 3C. The corresponding figure legend has also been revised in the manuscript. Please see the revised Figure 3C (**Figure R2-4**, revised Fig. 3c) as follows:

Figure R2-4. Volcano plot showing the difference in subtype-specific mutated genes.

Q5: Figure 3G. remove normals to see what the heat map looks like

Response: Thanks very much for the comment. We removed the normal samples in the heatmap of Figure 3G (**Figure R2-5**). Signature proteins in each subtype showed the highest expression levels in tumor samples of the corresponding subtypes. In last revision, we may not describe clearly about the usage of normal samples in the analysis, which confused the Reviewer. We would like to take this opportunity to clarify this. To reduce the effect of inter-patient heterogeneity and identify subtype-specific tumor differences, we collected NAT samples from regions within ~2 cm around the corresponding AEG tumor sites. We separately compared tumor with NAT samples in each AEG subtype. A considerable portion of differentially expressed proteins (DEPs) were exclusively identified in the subtype tumor-NAT comparisons (**Figure R2-6**). In total, 389, 731, and 630 DEPs in the S-I, S-II, and S-III subtype, respectively, were not detected in the analysis of all samples. The result demonstrated that subtype analysis could reveal many subtype-specific candidates that may help personalized therapy of AEG patients. To identify the specific molecular alterations in our proteomic subtypes, we compared the protein abundances between tumor samples in individual subtypes with those in tumor and NAT samples of the other subtypes. In each subtype, a protein that showed remarkably higher abundances than all NAT samples and tumor samples in the other subtypes was considered a signature protein.

Figure R2-5. Heatmap showing the expression of AEG subtype signature proteins that are significantly associated with patient survival across tumor samples in all subtypes.

Figure R2-6. DEPs in different comparisons. **a** Venny plot shows the overlaps among DEPs identified between tumor and normal samples in all AEG, S-I subtype, S-II subtype, or S-III subtype. **b** Upset plot shows the statistics of DEPs in different comparisons.

Q6: FBXO44 is a known oncogene and Figure 4C is consistent with it but not really novel. Figure 4A. Hard to find distinction in S subtypes. No explanation

Response: Thanks very much for the comment. We agree with the Reviewer that the oncogenic role of FBXO44 has been reported in several studies. Shen *et al.* interrogated public cancer transcriptomic data, and found high FBXO44 expression correlated with poor patient outcome in lung, breast, gastric and ovarian cancer⁵. Lu *et al.* found that FBXO44 is an E3 ubiquitin ligase responsible for BRCA1 degradation, which might contribute to the development of sporadic breast tumor⁶. These studies demonstrated FBXO44 may play different roles in different cancer types, which is worthy of further investigation in other cancer types. However, there was no data related to AEG. In addition, our study verified the expression of FBXO44 in AEG and its relationship with prognosis for the first time in **an Asian population cohort**, rather than only in public databases. As shown in Fig. 4a, the expression of FBXO44 in cancer tissues was significantly higher than that in adjacent cancer tissues in S-II, but there was no difference in S-I and S-III.

Q7: Figures 5 ABC also don't explain why S-III are surviving longer

Response: Thanks very much for the Reviewer's comment. We may not describe clearly in the last version of manuscript, which caused confusion. We would like to take this opportunity to clarify this. In this section, we would like to compare the genomic alterations between different AEG subtypes. These three AEG subtypes showed shared and specific mutation signatures. In particular, S-I and S-II shared the SBS3 signature, which indicates defects in DNA double-strand break (DSB) repair by homologous recombination (HR). Both the S-II and S-III subtypes exhibited SBS6 mutation signatures that represent defective DNA mismatch repair. The SBS17b mutation signature was shared by the S-I and S-III subtypes, which displayed an exclusively high frequency of T>G nucleotide substitution. The SBS1 signature was specifically identified in the S-I subtype, which showed spontaneous or enzymatic deamination of 5-methylcytosine. The S-II subtype exclusively exhibited the mutation signature of APOBEC cytidine deaminase (the SBS2 signature). The mutation signature of "deficiency in base excision repair due to inactivating mutations in NTHL1" (the SBS30 signature) was specifically detected in the S-III subtype. These genomic differences may provide insights into the development of tumor heterogeneity of AEG. **Although not all molecular differences could interpret the patient survival, these subtype-specific molecular features might serve as potential markers of differential diagnosis and precision treatment for AEG patients.**

Q8: DDRd should confer longer survival (but it is all over the place). Also APOBEC should be with longer survivors but it is not.

Response: Thanks for the Reviewer's comment. As shown in Fig. 5a-c, all subtypes have defects in DNA-DSB repair (DDRd), but the corresponding single base substitution (SBS) was not exactly the same. DNA double strand breaks (DSBs) are potential lethal lesions, including various SBS^{7,8}. The different SBS signature have different biological significance: SBS1, cell-division/mitotic clock; SBS2, hyperactivity of AID/APOBEC enzymes; SBS3: defective homologous recombination-based DNA repair; SBS6, defective DNA mismatch repair and microsatellite unstable tumors; SBS17b: specific KRAS/NRAS and EGFR driver mutations; SBS30: deficiency in base excision repair due to inactivating mutations in NTHL1. Besides, different SBS signature also play different roles in different tumors. For example, SBS30 (deficiency in base excision repair due to inactivating mutations in NTHL1), among various

base excision repair genes, NTHL1 was overexpressed in non-small cell lung cancer (NSCLC)⁹. In a clinical study of urothelial cancer patients, high NTHL1 expression negatively correlated with disease-free survival characterized by local recurrence of resected tumor or metastasis¹⁰. However, the overall NTHL1 expression remained insignificant in prognosis of grade or overall survival. Moreover, some studies found that the decrease expression of NTHL1 was significantly associated with a poor prognosis in astrocytoma¹¹. It can be seen that different SBS have different biological meanings, and the relationship between an SBS and prognosis is not the same in different tumors.

The APOBEC-induced mutagenesis promotes divergence in the genome that often results in evolving many variants with drug resistance and immune-escape capacity¹². On the other hand, the APOBEC-signature recurrent mutations found outside of stem-loops were reported to be accumulated in many validated driver genes and may anticipate new driver genes in cancer¹³. Survival analysis on the TCGA cohort revealed that low APOBEC signature is associated with prolonged overall survival in all patients². Notably, high APOBEC signature was associated with a marginally significant prolonged progression-free survival for an advanced NSCLC cohort treated with combination immunotherapy (PD-1 and CTLA-4)¹⁴. It can be seen that APOBEC plays different roles in different stages of tumor. Therefore, these mutation signatures are more likely markers to distinguish AEG subtypes, rather than to interpret the survival differences between AEG subtypes.

Q9: Figure 5G is also not very instructive. Mostly oncogenes but distributions are not striking.

Response: Thanks very much for the comment. We may not describe clearly in the last version of manuscript, which caused confusion. We would like to take this opportunity to clarify this. We agree with the Reviewer that the most frequently mutated oncogenic pathways in all three AEG subtypes. However, the specific mutations are different in different AEG subtypes. For example, patients of the S-I subtype had both large number of mutated genes and mutation rate of the "TP53" pathway, but the S-II and S-III subtype had smaller number of mutated genes and high mutation rate of the "TP53" pathway. Furthermore, distributions of mutated genes between different AEG subtypes are quite different in the same oncogenic pathways. For example, although gene mutations in the "RTK-RAS" pathway were found in over half of the

samples for individual subtypes, remarkably different sets of genes were affected in distinct subtypes (Fig. 5h).

Q10: Similarly, Figure 6D. not striking for S-I and S-III. FDR is very high

Response: Thanks very much for the comment. The Reviewer may refer to S-I and S-II that have high values of FDR. Our original description may not be clear enough. We would like to take this opportunity to clarify this. Fig. 6d separately compared the fibroblast abundance between AEG tumor and NAT samples in the S-I, S-II, and S-III subtypes. **The abundance of fibroblasts was significantly decreased in the S-III subtype (FDR=2.6E-4, Student's t test) but showed no obvious changes in tumor samples from the S-I and S-II subtypes (FDR=0.48 in S-I, FDR=0.98 in S-II, Figure R2-7).** In our following H&E analysis, the fibroblast abundance also showed a decrease in the S-III subtypes, compared to those in the S-I and S-II subtype (Fig. 6e).

Figure R2-7. Comparisons of fibroblast abundance between AEG tumor and NAT samples in the S-I, S-II, and S-III subtypes.

Q11: Figure 6 F also not giving any clues why S-III should live longer.

Response: Thanks very much for the comment. We may not state clearly about Fig. 6f in the last version of manuscript. As shown in Fig. 6f, we examined the expression changes in immune checkpoint genes to screen potential immunotherapy targets of different AEG subtypes, which were not necessarily associated with prognosis. We observed that some of the markers may be related to the prognosis, indicating that patients of the S-III subtype may have

a better response rate and treatment effect to tumor immunotherapy. Specifically, the expression of CD27 in the S-III subtype was significantly higher than that in the other types, while the expression of VTCN1 in the S-III subtype was significantly lower than that in the other types. CD27 is a co-stimulatory immune checkpoint molecule in the tumor necrosis factor receptor superfamily and functions to generate and maintain T cell immunity. In addition, CD27 signaling can increase production of the T cell growth/survival factor IL-2^{15,16}, leading to either improved T cell function or dysfunction. VTCN1, also known as B7-H4, is an immune checkpoint molecule that negatively regulates immune responses and is known to be overexpressed in many human cancers¹⁷. VTCN1 negatively regulates T cell immune response and promotes immune escape by inhibiting the proliferation, cytokine secretion, and cell cycle of T cells¹⁸. However, further studies are needed to confirm the specific role of these markers in the immune microenvironment of AEG.

References

- 1 Cancer Genome Atlas Research, N. *et al.* Integrated genomic characterization of oesophageal carcinoma. *Nature* **541**, 169-175 (2017).
- 2 Chen, Y. J. *et al.* Proteogenomics of Non-smoking Lung Cancer in East Asia Delineates Molecular Signatures of Pathogenesis and Progression. *Cell* **182**, 226-244 e217 (2020).
- 3 Li, C. *et al.* Integrated Omics of Metastatic Colorectal Cancer. *Cancer Cell* **38**, 734-747 e739 (2020).
- 4 Jiang, Y. *et al.* Proteomics identifies new therapeutic targets of early-stage hepatocellular carcinoma. *Nature* **567**, 257-261 (2019).
- 5 Shen, J. Z. *et al.* FBXO44 promotes DNA replication-coupled repetitive element silencing in cancer cells. *Cell* **184**, 352-369 e323 (2021).
- 6 Lu, Y. *et al.* The F-box protein FBXO44 mediates BRCA1 ubiquitination and degradation. *J Biol Chem* **287**, 41014-41022 (2012).
- 7 Krenning, L., van den Berg, J. & Medema, R. H. Life or Death after a Break: What Determines the Choice? *Mol Cell* **76**, 346-358 (2019).
- 8 Sizemore, S. T. *et al.* Pyruvate kinase M2 regulates homologous recombination-mediated DNA double-strand break repair. *Cell Res* **28**, 1090-1102 (2018).
- 9 Limpose, K. L. *et al.* Overexpression of the base excision repair NTHL1 glycosylase causes genomic instability and early cellular hallmarks of cancer. *Nucleic Acids Res* **46**, 4515-4532 (2018).

- 10 Wang, L. A. *et al.* The correlation of BER protein, IRF3 with CD8+ T cell and their prognostic significance in upper tract urothelial carcinoma. *Onco Targets Ther* **12**, 7725-7735, doi:10.2147/OTT.S222422 (2019).
- 11 Jiang, Z. *et al.* Expression analyses of 27 DNA repair genes in astrocytoma by TaqMan low-density array. *Neurosci Lett* **409**, 112-117 (2006).
- 12 Revathidevi, S., Murugan, A. K., Nakaoka, H., Inoue, I. & Munirajan, A. K. APOBEC: A molecular driver in cervical cancer pathogenesis. *Cancer Lett* **496**, 104-116 (2021).
- 13 Buisson, R. *et al.* Passenger hotspot mutations in cancer driven by APOBEC3A and mesoscale genomic features. *Science* **364** (2019).
- 14 Hellmann, M. D. *et al.* Genomic Features of Response to Combination Immunotherapy in Patients with Advanced Non-Small-Cell Lung Cancer. *Cancer Cell* **33**, 843-852 e844 (2018).
- 15 Matter, M. *et al.* Virus-induced polyclonal B cell activation improves protective CTL memory via retained CD27 expression on memory CTL. *Eur J Immunol* **35**, 3229-3239 (2005).
- 16 Peperzak, V., Xiao, Y., Veraar, E. A. & Borst, J. CD27 sustains survival of CTLs in virus-infected nonlymphoid tissue in mice by inducing autocrine IL-2 production. *J Clin Invest* **120**, 168-178 (2010).
- 17 Iizuka, A. *et al.* A T-cell-engaging B7-H4/CD3-bispecific Fab-scFv Antibody Targets Human Breast Cancer. *Clin Cancer Res* **25**, 2925-2934 (2019).
- 18 Wang, J. Y. & Wang, W. P. B7-H4, a promising target for immunotherapy. *Cell Immunol* **347**, 104008 (2020).

Reviewers' Comments:

Reviewer #3:

Remarks to the Author:

Mainly based on the efforts the authors have made, I recommend to consider publication of this manuscript.

Point-by-point Response (NCOMMS-22-09290A)

Reviewer #3 (Remarks to the Author):

Mainly based on the efforts the authors have made, I recommend to consider publication of this manuscript.

Response: We are very delighted to hear that our revision satisfied all the Reviewer's concerns. We greatly appreciate the Reviewer's valuable comments and suggestions that have helped much improved our manuscript.